# Simplifying Deep Temporal Difference Learning

**Matteo Gallici**[*1]  **Mattie Fellows**[*2]  **Benjamin Ellis**[2]
**Bartomeu Pou**[1,3]  **Ivan Masmitja**[4]  **Jakob Nicolaus Foerster**[2]  **Mario Martin**[1]
[1]Universitat Politècnica de Catalunya  [2]University of Oxford
[3]Barcelona Supercomputing Center  [4] Institut de Ciències del Mar
{gallici,mmartin}@cs.upc.edu
{matthew.fellows,benjamin.ellis,jakob.foerster}@eng.ox.ac.uk
bartomeu.poumulet@bsc.es  masmitja@icm.csic.es

## Abstract

$Q$-learning played a foundational role in the field reinforcement learning (RL). However, TD algorithms with off-policy data, such as $Q$-learning, or nonlinear function approximation like deep neural networks require several additional tricks to stabilise training, primarily a large replay buffer and target networks. Unfortunately, the delayed updating of frozen network parameters in the target network harms the sample efficiency and, similarly, the large replay buffer introduces memory and implementation overheads. In this paper, we investigate whether it is possible to accelerate and simplify off-policy TD training while maintaining its stability. Our key *theoretical* result demonstrates for the first time that regularisation techniques such as LayerNorm can yield provably convergent TD algorithms without the need for a target network or replay buffer, even with off-policy data. *Empirically*, we find that online, parallelised sampling enabled by vectorised environments stabilises training without the need for a large replay buffer. Motivated by these findings, we propose PQN, our *simplified* deep online $Q$-Learning algorithm. Surprisingly, this simple algorithm is competitive with more complex methods like: Rainbow in Atari, PPO-RNN in Craftax, QMix in Smax, and can be up to 50x faster than traditional DQN without sacrificing sample efficiency. In an era where PPO has become the go-to RL algorithm, PQN reestablishes off-policy $Q$-learning as a viable alternative. We open-source our code at: https://github.com/mttga/purejaxql.

## 1 Introduction

In reinforcement learning (RL), the challenge of developing simple, efficient and stable algorithms remains open. Temporal difference (TD) methods have the potential to be simple and efficient, but are notoriously unstable when combined with either off-policy sampling or nonlinear function approximation (Tsitsiklis & Van Roy, 1997). Starting with the introduction of the seminal deep $Q$-network (DQN)(Mnih et al., 2013), many tricks have been developed to stabilise TD for use with deep neural network function approximators, most notably: the introduction of batched learning through a replay buffer (Mnih et al., 2013), target networks (Mnih et al., 2015), trust region based methods (Schulman et al., 2015), double Q-networks (Wang & Blei, 2017; Fujimoto et al., 2018), maximum entropy methods (Haarnoja et al., 2017; 2018) and ensembling (Chen et al., 2021). Out of this myriad of algorithmic combinations, proximal policy optimisation (PPO) (Schulman et al., 2017) has emerged as the de facto choice for RL practitioners, proving to be a strong and efficient baseline across popular RL domains. Unfortunately, PPO is far from stable and simple: PPO does not have provable convergence properties for nonlinear function approximation and requires extensive tuning and additional tricks to implement effectively (Huang et al., 2022a; Engstrom et al., 2020).

Recent empirical studies (Lyle et al., 2023; 2024; Bhatt et al., 2024) provide evidence that TD can be stabilised without target networks by introducing regularisation such as BatchNorm (Ioffe & Szegedy, 2015) and LayerNorm (Ba et al., 2016; Nauman et al., 2024) into the $Q$-function approximator. Little is known about why these techniques work or whether they have unintended side-effects. Motivated by these findings, we ask: are regularisation techniques such as BatchNorm and LayerNorm the key to unlocking simple, efficient and stable RL algorithms? To answer this question, we provide a

---

[*]Equal Contribution.

rigorous analysis of regularised TD. We summarise our core theoretical contributions as: **I)** Introduce a highly general and widely applicable analysis of TD stability; **II)** we show introducing LayerNorm *and $\ell^2$* regularisation into the $Q$-function approximator leads to provable convergence, stabilising nonlinear and/or off-policy TD *without the need for target networks or replay buffers*.

Many applications in RL allow for multiple actions to be taken in an environment at once, solving a parallel world problem. Guided by our theoretical insights, we develop a modern off-policy value-based TD method which we call a parallelised $Q$-network (PQN): for simplicity, we revisit the original $Q$-learning algorithm (Watkins, 1989), which updates a $Q$-function approximator without a target network. A recent breakthrough in RL has been running the environment and agent jointly on the GPU (Makoviychuk et al., 2021; Gu et al., 2023; Lu et al., 2022; Matthews et al., 2024b; Rutherford et al., 2023; Lange, 2022). However, the replay buffer's large memory footprint makes pure-GPU training impractical with traditional DQN. With the goal of enabling $Q$-learning in pure-GPU setting, we propose replacing a large replay buffer with a synchronous update across a large number of parallel environments, reducing memory requirements. For stability, we integrate our theoretical findings in the form of a regularised deep $Q$ network. We provide a schematic of our proposed PQN algorithm in Fig. 1d.

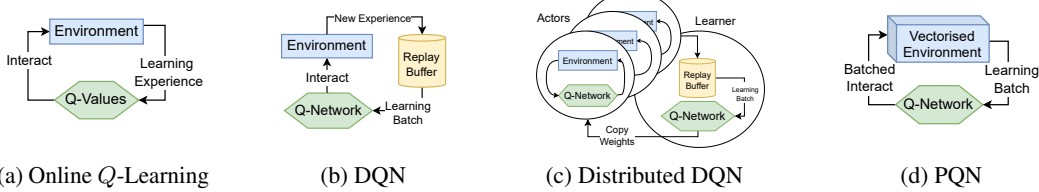

(a) Online $Q$-Learning     (b) DQN     (c) Distributed DQN     (d) PQN

Figure 1: Classical $Q$-Learning directly interacts with the environment and updates the learned $Q$-values at each transition. In contrast, DQN stores experiences in a replay buffer and trains a $Q$-network using minibatches sampled from this buffer. Distributed DQN enhances this approach by collecting experiences in parallel threads, while a separate process continually trains the network (i.e. a learner module and multiple actors modules run concurrently and independently). Similar to online $Q$-Learning, PQN trains a $Q$-network with the experiences as they are collected in the same process, but conducts interactions and learning in batches.

To validate our theoretical results, we evaluated PQN in Baird's counterexample, a challenging domain that is provably divergent for off-policy methods (Baird, 1995). Our results show that PQN can converge where non-regularised variants fails. We provide an extensive empirical evaluation to test the performance of PQN in single-agent and multi-agent settings. Despite its simplicity, our algorithm is competitive in a range of tasks; notably, PQN achieves high performances in just a few hours in many games of the Arcade Learning Environment (ALE) (Bellemare et al., 2013), competes effectively with PPO on the open-ended Craftax task (Matthews et al., 2024a), and stands alongside state-of-the-art Multi-Agent RL (MARL) algorithms, such as MAPPO in Overcooked (Carroll et al., 2019) and Hanabi (Bard et al., 2020) and Qmix in Smax (Rutherford et al., 2023). Despite not sampling from a large buffer of historic data, the faster convergence of PQN demonstrates that the sample efficiency loss can be minimal. This positions PQN as a strong method for efficient and stable RL in the age of deep vectorised Reinforcement Learning (DVRL).

We summarise our empirical contributions: **I)** we propose PQN, a simplified, parallelised, and normalised version of DQN which eliminates the use of both large replay buffers and the target network; **II)** we demonstrate that PQN is fast, stable, simple to implement, uses few hyperparameters, and is compatible with pure-GPU training and temporal-based networks such as RNNs, and **III)** our extensive empirical study demonstrates PQN achieves competitive results *in significantly less wall-clock time than existing state-of-the-art methods*.

## 2 PRELIMINARIES

Let $\|\cdot\|$ denote the $\ell^2$-norm and $\mathcal{P}(\mathcal{X})$ the set of all probability distributions over a set $\mathcal{X}$.

### 2.1 REINFORCEMENT LEARNING

In this paper, we consider the infinite horizon discounted RL setting, formalised as a Markov Decision Process (MDP) (Bellman, 1957; Puterman, 2014): $\mathcal{M} \coloneqq \langle \mathcal{S}, \mathcal{A}, P_S, P_0, P_R, \gamma \rangle$ with bounded state space $\mathcal{S}$, bounded action space $\mathcal{A}$, transition distribution $P_S : \mathcal{S} \times \mathcal{A} \to \mathcal{P}(\mathcal{S})$, initial state distribution $P_0 \in \mathcal{P}(\mathcal{S})$, bounded stochastic reward distribution $P_R : \mathcal{S} \times \mathcal{A} \to \mathcal{P}([-r_{max}, r_{max}])$ where $r_{max} \in \mathbb{R} < \infty$ and scalar discount factor $\gamma \in [0, 1)$. An agent in state $s_t \in \mathcal{S}$ taking action $a_t \in \mathcal{A}$ observes a reward $r_t \sim P_R(s_t, a_t)$. The agent's behaviour is determined by a policy that

maps a state to a distribution over actions: $\pi : \mathcal{S} \to \mathcal{P}(\mathcal{A})$ and the agent transitions to a new state $s_{t+1} \sim P_S(s_t, a_t)$. As the agent interacts with the environment through a policy $\pi$, it follows a trajectory $\tau_t := (s_0, a_0, r_0, s_1, a_1, r_1, \ldots s_{t-1}, a_{t-1}, r_{t-1}, s_t)$ with distribution $P_t^\pi$. For simplicity, we denote state-action pair $x_t := (s_t, a_t) \in \mathcal{X}$ where $\mathcal{X} := \mathcal{S} \times \mathcal{A}$. The state-action pair transitions under policy $\pi$ according to the distribution $P_X^\pi(x) : \mathcal{X} \to \mathcal{P}(\mathcal{X})$.

The agent's goal is to learn an optimal policy of behaviour $\pi^\star \in \Pi^\star$ by optimising the expected discounted sum of rewards over all possible trajectories $J^\pi$, where: $\Pi^\star := \arg\max_\pi J^\pi$ is the set of optimal policies for the objective $J^\pi := \mathbb{E}_{\tau_\infty \sim P_\infty^\pi} [\sum_{t=0}^\infty \gamma^t r_t]$. The expected discounted reward for an agent in state $s_t$ for taking action $a_t$ is characterised by a $Q$-function, which is defined recursively through the Bellman equation: $Q^\pi(x_t) = \mathcal{B}^\pi[Q^\pi](x_t)$, where the Bellman operator $\mathcal{B}^\pi$ projects functions forwards by one step through the dynamics of the MDP: $\mathcal{B}^\pi[Q^\pi](x_t) := \mathbb{E}_{x_{t+1} \sim P_X^\pi(x_t), r_t \sim P_R(x_t)} [r_t + \gamma Q^\pi(x_{t+1})]$. Of special interest is the $Q$-function for an optimal policy $\pi^\star$, which we denote as $Q^\star(x_t) := Q^{\pi^\star}(x_t)$. The optimal $Q$-function satisfies the optimal Bellman equation $Q^\star(x_t) = \mathcal{B}^\star[Q^\star](x_t)$, where $\mathcal{B}^\star$ is the optimal Bellman operator: $\mathcal{B}^\star[Q^\star](x_t) := \mathbb{E}_{s_{t+1} \sim P_S(x_t), r_t \sim P_R(x_t)} [r_t + \gamma \max_{a'} Q^\star(s_{t+1}, a')]$.

## 2.2 TEMPORAL DIFFERENCE METHODS

Many RL algorithms employ TD learning for policy evaluation, which combines bootstrapping, state samples and sampled rewards to estimate the expectation in the Bellman operator (Sutton, 1988). We introduce a $Q$-function approximation $Q_\phi : \mathcal{X} \to \mathbb{R}$ parametrised by $\phi \in \Phi$ to represent the space of $Q$-functions. We assume that $Q_\phi$ is initialised from a distribution $\phi_0 \sim P_\Phi$. In their simplest form, TD methods estimate the application of a Bellman operator by updating the $Q$-function approximator parameters according to:

$$\phi_{i+1} = \phi_i + \alpha_i \left(r + \gamma Q_{\phi_i}(x') - Q_{\phi_i}(x)\right) \nabla_\phi Q_{\phi_i}(x), \tag{1}$$

where $x \sim d^\mu, r \sim P_R(x), x' \sim P_X^\pi(x)$ and $\alpha_i$ is a sequence of stepsizes satisfying the standard Robbins-Munro conditions (Robbins & Monro, 1951):

**Assumption 1** (RM Conditions). *We assume $\alpha_i > 0$ with $\sum_{i=0}^\infty \alpha_i = \infty$ and $\sum_{i=0}^\infty \alpha_i^2 < \infty$.*

Here $d^\mu \in \mathcal{P}(\mathcal{X})$ is a sampling distribution, and $\mu$ is a sampling policy that may be different from the target policy $\pi$. Methods for which the sampling policy differs from the target policy are known as *off-policy* methods. In this paper, we will study the $Q$-learning (Watkins, 1989; Dayan, 1992) TD update: $\phi_{i+1} = \phi_i + \alpha_i(r + \gamma \sup_{a'} Q_{\phi_i}(s', a') - Q_{\phi_i}(x)) \nabla_\phi Q_{\phi_i}(x)$, which aims to learn an optimal $Q$-function by estimating the optimal Bellman operator. As data in $Q$-learning is gathered from an exploratory policy $\mu$ that is not optimal, $Q$-learning is an inherently off-policy algorithm. For simplicity of notation we define the tuple $\varsigma := (x, r, x')$ with distribution $P_\varsigma$ and the TD-error vector as:

$$\delta(\phi, \varsigma) := \left(r + \gamma Q_\phi(x') - Q_\phi(x)\right) \nabla_\phi Q_\phi(x), \tag{2}$$

allowing us to write the TD parameter update as: $\phi_{i+1} = \phi_i + \alpha_i \delta(\phi_i, \varsigma)$. Typically, $d^\mu$ is the stationary state-action distribution of an ergodic Markov chain but may be another offline distribution such as a distribution induced by a replay buffer. We introduce the following mild regularity assumptions for our analysis.

**Assumption 2** (Regularity Assumptions). *Assume that $\Phi \subset \mathbb{R}^d$ is compact and convex and $\delta(\phi, \varsigma)$ is Lipschitz in $\phi, \varsigma$. When updating TD, $x \sim d^\mu$ is either sampled i.i.d. from a distribution with support over $\mathcal{X}$ or is sampled from a geometrically ergodic Markov chain with stationary distribution $d^\mu$.*

The condition of $\Phi \subset \mathbb{R}^d$ being compact is ubiquitous in TD theory and stochastic approximation (Papavassiliou & Russell, 1999; Nemirovski et al., 2009; Maei et al., 2010; Kushner, 2010; Lacoste-Julien et al., 2012; Bhandari et al., 2018; Wang et al., 2020; Yang et al., 2019; Zhang et al., 2021) and can be achieved by projecting any $\phi \notin \Phi$ back into $\Phi$ using the projection $P_\Phi(\phi') := \arg\min_{\phi \in \Phi} \|\phi - \phi'\|$. Projection is a mathematical formality and should not be required in practice as $\Phi$ can be made large enough to contain all updates when TD is stable and a suitable stepsize regime is chosen. Finally, geometric ergodicity extends traditional notions of aperiodicity and irreducibility in discrete MDPs to the more general continuous state-action space formulations (see Roberts & Rosenthal (2004) for details). It is one of the weakest ergodicity assumptions.

We denote the expected TD-error vector as: $\bar{\delta}(\phi) := \mathbb{E}_{\varsigma \sim P_\varsigma}[\delta(\phi, \varsigma)]$, and define the set of TD fixed points as: $\phi^\star \in \{\phi | \bar{\delta}(\phi) = 0\}$. If a TD algorithm converges, it must converge to a TD fixed point as the expected parameter update is zero for all $\phi^\star$. We remark that convergence to a TD fixed point does not imply a value error of zero between the approximate and true $Q$-function (Kolter, 2011).

### 2.3 VECTORISED ENVIRONMENTS

Parallelising the interactions between an RL agent and a learning environment is a standard method for speeding up training. In classical frameworks like Gymnasium (Towers et al., 2023), this is achieved by processing multiple environments via multi-threading. In more recent GPU-based frameworks like IsaacGym (Makoviychuk et al., 2021), ManiSkill2 (Gu et al., 2023), Jumanji (Bonnet et al., 2024), Craftax (Matthews et al., 2024b), JaxMARL (Rutherford et al., 2023) the environments' operations are *vectorised*, meaning that they are performed together using batched tensors. This allows an agent to easily interact with thousands of environments, and it enables the compilation of end-to-end GPU learning pipelines, which can accelerate the training of on-policy agents like PPO and A2C by orders of magnitude (Makoviychuk et al., 2021; Weng et al., 2022; Gu et al., 2023; Lu et al., 2022). Unfortunately, end-to-end single-GPU training is not compatible with traditional off-policy methods like DQN for two reasons: firstly, maintaining a replay buffer in GPU is not feasible in complex environments, as it would occupy most of the GPU memory; and secondly, the convergence of off-policy methods demands a very high number of updates in relation to the sampled experiences (DQN traditionally performs one gradient step per environment step). Commonly, parallelisation of Q-Learning (like in Ape-X (Horgan et al., 2018), R2D2 (Kapturowski et al., 2018) and a recent method presented in Li et al. (2023)) is achieved by continuously training the Q-network in a separate process in order to keep up with the fast sampling (see Fig. 1c), a setup that is not feasible in a single pure-GPU setting. For this reason, all referenced frameworks primarily provide PPO or A2C baselines, i.e. *vectorised RL lacks a off-policy Q-learning baseline.*

### 2.4 RELATED WORK

Our paper makes several significant contributions across a range of interconnected threads in RL research. We provide an extensive discussion of all related work in Appendix A.

## 3 ANALYSIS OF REGULARISED TD

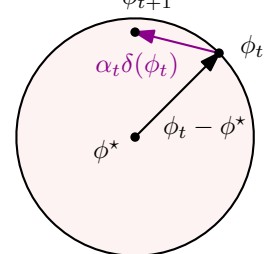

Figure 2: Geometric interpretation of TD stability criterion. Expected updates in the shaded ball ensure contraction mapping.

*Proofs for all theorems and corollaries can be found in Appendix B* Building on (Bhandari et al., 2018; Fellows et al., 2023), we now develop a powerful and general Jacobian analysis tool to characterise stability of TD approaches used in practice (Section 3.1). We then apply this analysis to regularised TD, confirming our theoretical hypothesis that careful application of LayerNorm and $\ell^2$ regularisation can stabilise TD (Section 3.2). Finally, we compare LayerNorm to BatchNorm regularisation techniques in Section 3.3, explaining our preference for LayerNorm. Recalling that $x = (s, a)$, we remark that our results can be derived for value functions by setting $x = s$ in our analysis.

### 3.1 STABILITY OF TD

As TD updates aren't a gradient of any objective, they fall under the more general class of algorithms known as stochastic approximation (Robbins & Monro, 1951; Borkar, 2008). Stability is not guaranteed in the general case and convergence of TD methods has been studied extensively (Watkins & Dayan, 1992; Tsitsiklis & Van Roy, 1997; Dalal et al., 2017; Bhandari et al., 2018; Srikant & Ying, 2019). We now extend the methods of Fellows et al. (2023) to study general nonlinear TD in a Markov chain, meaning our analysis applies exactly to TD methods used in practice. Key to determining stability of the TD updates is establishing that the Jacobian is *negative definite*:

> **TD Stability Criterion:** Define the TD Jacobian as $J(\phi) \coloneqq \nabla_\phi \delta(\phi)$. The TD stability criterion holds if the Jacobian is negative definite, that is: $v^\top J(\phi) v < 0$ for any test vector $v \neq 0$ and $\phi \in \Phi$, except possibly on a set of measure 0.

Intuitively, the Jacobian replaces the Hessian from classical optimisation theory (Boyd & Vandenberghe, 2004), which measures curvature of the underlying objective, thereby ensuring convexity. As TD methods are not a gradient of any objective, the TD stability condition instead implies $\delta(\phi_t)^\top (\phi_t - \phi^\star) < 0$ for all $\phi_t$, ensuring the expected update vector will always move the parameters closer to a fixed point with a sufficiently small stepsize. We sketch a geometric interpretation in Fig. 2. Mathematically, if the TD stability criterion holds, then as stepsizes approach zero in the limit $\lim_{i \to \infty} \alpha_i = 0$, there exists some $t$ such that for every $i > t$ each update is a contraction mapping: $\mathbb{E}[\|\phi_{i+1} - \phi^\star\|] < \mathbb{E}[\|\phi_i - \phi^\star\|]$. This key condition allows us to prove convergence of TD:

**Theorem 1** (TD Stability). *Let Assumptions 1 and 2 hold. If the TD criterion holds then the TD updates in Eq.* (1) *converge with:* $\lim_{i \to \infty} \mathbb{E}\left[\|\phi_i - \phi^\star\|^2\right] = 0$.

We can split the TD Jacobian condition into two separate off-policy and nonlinear components: $v^\top J(\phi)v = \mathcal{C}_{\text{OffPolicy}}(Q_\phi, d^\mu) + \mathcal{C}_{\text{Nonlinear}}(Q_\phi)$, whose negativity ensure the overall TD stability criterion is satisfied (see Appendix B.1). This naturally yields two forms of TD instability:

**Off-policy Instability:** The TD stability criterion can be violated if:

$$\mathcal{C}_{\text{OffPolicy}}(Q_\phi, d^\mu) := \gamma \mathbb{E}_{\varsigma \sim P_\varsigma}\left[v^\top \nabla_\phi Q_\phi(x')v^\top \nabla_\phi Q_\phi(x)\right] - \mathbb{E}_{x \sim d^\mu}\left[\left(v^\top \nabla_\phi Q_\phi(x)\right)^2\right] < 0, \quad (3)$$

does not hold for any test vector $v$. To better understand the off-policy component, we invoke the Cauchy-Schwarz inequality to show $\mathbb{E}_{\varsigma \sim P_\varsigma}\left[\left(v^\top \nabla_\phi Q_\phi(x')\right)^2\right] \leq \mathbb{E}_{x \sim d^\mu}\left[\left(v^\top \nabla_\phi Q_\phi(x)\right)^2\right]$ is key to proving $\mathcal{C}_{\text{OffPolicy}}(Q_\phi, d^\mu) < 0$ (see Appendix B.1 for a derivation). Unfortunately, ergodic theory reveals this condition only holds in the *on-policy* sampling regime, i.e. when $d^\mu = d^\pi$, for both i.i.d. or Markov chain sampling. For off-policy sampling, the distributional shift between the target policy $\pi$ and the sampling policy $\mu$ can cause the expectation $\mathbb{E}_{\varsigma \sim P_\varsigma}\left[\left(v^\top \nabla_\phi Q_\phi(x')\right)^2\right]$ to be arbitrarily large. We conclude that $\mathcal{C}_{\text{OffPolicy}}(Q_\phi, d^\mu)$ characterises the degree of distributional shift that TD can tolerate before becoming unstable and *off-policy sampling* is a key source of instability in TD, especially in algorithms such as $Q$-learning.

**Nonlinear Instability:** The TD stability criterion can be violated if:

$$\mathcal{C}_{\text{Nonlinear}}(Q_\phi) := \mathbb{E}_{\varsigma \sim P_\varsigma}\left[(r + \gamma Q_\phi(x') - Q_\phi(x))v^\top \nabla_\phi^2 Q_\phi(x)v\right] < 0, \quad (4)$$

does not hold for any test vector $v$. This condition does not apply in the linear case as second order derivatives are zero: $\nabla_\phi^2 Q_\phi(x) = 0$. In the nonlinear case, the left hand side of the inequality can be arbitrarily positive depending upon the specific MDP and choice of function approximator. Hence *nonlinearity* is a key source of instability in TD which is characterised by $\mathcal{C}_{\text{Nonlinear}}(Q_\phi)$. Together both off-policy and nonlinear instability formalise the *deadly triad* (Sutton & Barto, 2018a; van Hasselt et al., 2018) and TD can be unstable if *either* Conditions 3 or 4 are not satisfied. We now investigate how LayerNorm with $\ell^2$ regularisation can tackle these sources of instability.

## 3.2 STABILISING TD WITH LAYERNORM + $\ell^2$ REGULARISATION

To understand how LayerNorm with $\ell^2$ regularisation stabilises TD, we study the following $Q$-function approximator:

$$Q_\phi^k(x) = w^\top \sigma_{\text{Post}} \circ \text{LayerNorm}^k[\sigma_{\text{Pre}} \circ Mx]. \quad (5)$$

Here $\phi = [w^\top, \text{Vec}(M)^\top]$ is the parameter vector where $M \in \mathbb{R}^{k \times d}$ is a $k \times d$ matrix where each row $\|m_i\|$ is bounded, $w \in \mathbb{R}^k$ is a vector of final layer weights where $\|w\|$ is bounded and $\sigma_{\text{Pre}}$ and $\sigma_{\text{Post}}$ are element-wise $C^2$ continuous activations with bounded 2nd order derivatives. We assume the final activation $\sigma_{\text{Post}}$ is $L_{\text{Post}}$-Lipschitz with $\sigma_{\text{Post}}(0) = 0$ (e.g. tanh, identity, GELU, ELU...). LayerNorm (Ba et al., 2016) is defined element-wise as:

$$\text{LayerNorm}_i^k[f(x)] := \frac{1}{\sqrt{k}} \cdot \frac{f_i(x) - \frac{1}{k}\sum_{j=0}^{k-1} f_j(x)}{\sqrt{\frac{1}{k}\sum_{i=0}^{k-1}(f_i(x) - \frac{1}{k}\sum_{j=0}^{k-1} f_j(x))^2 + \epsilon}}, \quad (6)$$

where $\epsilon > 0$ is a small constant introduced for numerical stability. Deeper networks with more LayerNorm layers may be used in practice, however our analysis reveals that only the final layer weights affect the stability of TD with wide LayerNorm neural networks. We observe that adding LayerNorm does not affect the representational capacity of the network as it merely rescales the input according to a standard Gaussian. The output is then rescaled due to the final linear layer. As $k$ increases, the empirical mean and standard deviations in Eq. (6) approach their true expectations, thereby increasing the degree of normalisation provided. Using the LayerNorm $Q$-function, we can bound the off-policy and nonlinear components of the TD stability condition:

**Lemma 2.** *Let Assumption 2 apply. Let $v_w$ be the first $k$ components of the test vector $v = [v_w^\top, v_M^\top]^\top$, associated with final layer parameters $w$, and $v_M$ be the remaining components, associated with the matrix parameters $\text{Vec}(M)$. Using the* LayerNorm *$Q$-function defined in Eq.* (5):

$$\text{Off-Policy Bound:} \qquad \mathcal{C}_{\text{OffPolicy}}(Q_\phi^k, d^\mu) \leq \|v_w \cdot \gamma L_{\text{Post}}/2\|^2 + \mathcal{O}\left(\|v_M\|^2/k\right), \quad (7)$$

$$\text{Nonlinear Bound:} \qquad \mathcal{C}_{\text{Nonlinear}}(Q_\phi^k) = \mathcal{O}\left(\|v\|^2/\sqrt{k}\right), \quad (8)$$

*almost surely for any test vector $v$ and any state-action transition pair $x, x' \in \mathcal{X}$.*

Analysis in Eq. (7) and Eq. (8) of Lemma 2 reveals that as the degree of regularisation increases, that is in the limit $k \to \infty$, all nonlinear instability can be mitigated: $\lim_{k \to \infty} \mathcal{C}_{\text{Nonlinear}}(Q_\phi^k) = 0$ and a residual term is left in the off-policy bound: $\lim_{k \to \infty} \mathcal{C}_{\text{OffPolicy}}(Q_\phi^k, d^\mu) \leq \left\| v_w \cdot \gamma^{L_{\text{Post}}/2} \right\|^2$. The nonlinear bound in Eq. (8) can be explained using established theory of wide neural networks; as layer width increases, second order derivative terms tend to zero (Liu et al., 2020). Our proof extends this theory, showing that LayerNorm preserves this property.

As linear function approximators still stuffer from off-policy instability due the distributional shift between $\pi$ and $\mu$, linearisation of wide networks cannot explain the bound in Eq. (7). Instead, our proof for Lemma 2 reveals this bound is due to the normalising property of LayerNorm, which upper bounds the expected norm: $\mathbb{E}_{x \sim d^\mu} \left[ \left\| \text{LayerNorm}^k[Mx] \right\| \right] \leq 1$ *regardless of the sampling distribution* $d^\mu$ or *magnitude of* $M$. This yields a bound with a residual term of $\left\| v_w \cdot \gamma^{L_{\text{Post}}/2} \right\|^2$ that is independent of $\pi$ and $\mu$, overcoming the distributional shift issue responsible for off-policy instability. We tackle it by targeting $\phi$ with $\ell^2$ regularisation using the following TD update vector:

$$\delta_{\text{reg}}^k(\phi, \varsigma) := \delta^k(\phi, \varsigma) - \left( \eta \left( \gamma^{L_{\text{Post}}/2} \right)^2 \begin{bmatrix} w \\ 0 \end{bmatrix} + (\eta - 1) \begin{bmatrix} 0 \\ \text{Vec}(M) \end{bmatrix} \right), \tag{9}$$

for any $\eta > 1$ where $\delta^k(\phi, \varsigma)$ is the TD update vector from Eq. (2) using the LayerNorm critic from Eq. (5) respectively. Eq. (9) yields a bound:

$$\mathcal{C}_{\text{OffPolicy}}(Q_\phi^k, d^\mu) \leq (1 - \eta) \left( \left\| v_w \cdot \gamma^{L_{\text{Post}}/2} \right\|^2 + \left\| v_M \right\|^2 \right) + \mathcal{O}\left( 1/k \right),$$

which implies $\mathcal{C}_{\text{OffPolicy}}(Q_\phi^k, d^\mu) < 0$ with sufficiently large $k$, meaning the TD stability criterion will be satisfied. We now formally confirm now this intuition:

**Theorem 2.** *Let Assumption 2 apply. Using the* LayerNorm *regularised TD update* $\delta_{\text{reg}}^k(\phi, \varsigma)$ *in Eq.* (9)*, there exists some finite* $k'$ *such that the TD stability criterion holds for all* $k > k'$.

In Section 5.1 we test our theoretical claim in Theorem 2 empirically, demonstrating that LayerNorm $+ \ell^2$ regularisation can stabilise Baird's counterexample, an MDP intentionally designed to cause TD to diverge (Baird, 1995). We remark that whilst adding an $\ell^2$ regularisation term $-\eta\phi$ to all parameters can stabilise TD alone, large $\eta$ recovers a quadratic optimisation problem with minimum at $\phi = 0$, pulling the TD fixed points towards 0. Hence, we suggest $\ell^2$-*regularisation should be used sparingly*; only when LayerNorm alone cannot stabilise the environment and initially only over the final layer weights. Aside from Baird's counterexample, we find LayerNorm without $\ell^2$ regularisation can stabilise all environments in our extensive empirical evaluation in Section 5.

### 3.3 LAYERNORM AND BATCHNORM TD

We have seen from Theorem 2 that LayerNorm $+ \ell^2$ regularised TD can stabilise TD by mitigating the effects of nonlinearity and off-policy sampling. Empirical evidence suggests that BatchNorm (Ioffe & Szegedy, 2015) regularisation, which is essential for stabilising algorithms such as CrossQ (Bhatt et al., 2024), may also possess similar properties to LayerNorm. It is natural to ask: 'what are the potential benefits of LayerNorm over BatchNorm methods?'

Naïvely applying BatchNorm as presented by Ioffe & Szegedy (2015) does not stabilise TD as CrossQ does not succeed without applying several modifying 'tricks' such as double Q-learning, batch renormalisation using running statistics and calculating the batch statistics from a mixture of datasets (Bhatt et al., 2024). In contrast, LayerNorm $+ \ell^2$ regularisation benefits from the strong theoretical guarantees in Theorem 2 without burdening practioners with additional tricks and their associated hyperparameter tuning. Additionally, compared to BatchNorm, LayerNorm does not require memory or estimation of the running batch averages.

Our empirical analysis in Section 5 shows that BatchNorm can degrade performance in some cases, while in others it can improve results if applied early in the network. Therefore, we don't dismiss BatchNorm outright, but a thorough theoretical analysis is needed to fully understand its practical effects. Nonetheless, we recommend starting with LayerNorm and $\ell^2$ regularisation as a strong, simple baseline for stabilising TD algorithms before experimenting with alternatives like BatchNorm.

## 4 PARALLELISED $Q$-LEARNING

Guided by our analysis in Section 3, we develop a simplified version of deep $Q$-learning to exploit the power of parallelised sampling with minimal memory requirements and without target networks. The Q-Network is regularised with network normalisation (preferably LayerNorm) and $\ell^2$ regularisation

as required (see Eq. (9)). As we are developing an online algorithm, it is straightforward to exploit $n$-step returns. In Algorithm 1 we present PQN with $\lambda$-returns, which is a parallelised variant of the approach of Daley & Amato (2019). An exploration policy $\pi_{\text{Explore}}$ ($\epsilon$-greedy for this paper) is rolled out for a small trajectory of size $T$: $(s_i, a_i, r_i, s_{i+1} \ldots s_{i+T})$. Starting with $R^\lambda_{i+T} = \max_{a'} Q_\phi(s_{i+T}, a')$ the targets are computed recursively back in time from $R^\lambda_{i+T-1}$ to $R^\lambda_i$ using: $R^\lambda_t = r_t + \gamma \left[ \lambda R^\lambda_{t+1} + (1-\lambda) \max_{a'} Q_\phi(s_{t+1}, a') \right]$ or $R^\lambda_t = r_t$ if $s_t$ is a terminal state. We provide a derivation of our approach in Appendix B.4. Due to the use of $\lambda$-returns and minibatches, we require a small buffer of size $I \cdot T$ containing interactions from the *current exploration policy*.

The special case $\lambda = 0$ with $T = 1$ is equivalent to a vectorised variant of Watkins (1989)'s original $Q$-learning algorithm with LayerNorm + $\ell^2$ regularisation where $I$ separate interactions occur in parallel with the environment.

PQN with $\lambda$-returns is simpler than existing state-of-the-art $\lambda$-based algorithms such as Retrace (Munos et al., 2016) which adopt computationally intensive techniques to handle the computation of $\lambda$-targets. Similarly, an implementation of PQN using RNNs only requires sampling trajectories for multiple time-steps and then back-propagating the gradient through time in the learning phase. In contract existing approaches like R2D2 (Kapturowski et al., 2018) that integrate RNNs with replay buffers must handle hidden states of trajectories collected with old policies during replay. A basic multi-agent version of PQN for coordination problems can be obtained by adopting Value Network Decomposition Networks (VDN) (Sunehag et al., 2017b), i.e. opti-

---

**Algorithm 1** PQN with $\lambda$-returns

---

1: $\phi \leftarrow$ initialise regularised $Q$-network parameters
2: $s_0 \sim P_0, t \leftarrow 0$
3: **for** *each episode* **do**
4:   **for** *each $i \in \{0, 1, \ldots I - 1\}$* *(in parallel)* **do**
5:     $a^i_t \sim \pi_{\text{Explore}}(s^i_t)$, (e.g. $\epsilon$-greedy)
6:     $r^i_t \sim P_R(s^i_t, a^i_t) \; s^i_{t+1} \sim P_S(s^i_t, a^i_t)$,
7:     $t \leftarrow t + 1$,
8:   **end for**
9:   **if** $t \mod T = 0$ **then**
10:     calculate $R^{\lambda,i}_{t-1}$ to $R^{\lambda,i}_{t-T}$,
11:     **for** *number of epochs* **do**
12:       **for** *number of minibatches* **do**
13:         draw minibatch $B$ of size $b \leq I \cdot T$ from $\{t - T, \ldots t - 1\}$ and $\{0, \ldots I - 1\}$
14:         $\phi \leftarrow \phi + \frac{\alpha_t}{2b} \nabla_\phi \sum_{i,t \in B} (R^{\lambda,i}_t - Q_\phi(x^i_t))^2$
15:       **end for**
16:     **end for**
17:   **end if**
18: **end for**

---

mising the joined action-value function as a sum of the single agents action-values. Finally, similar to PPO, it is possible to increase PQN's sample efficiency by dividing the collected experiences into multiple minibatches and using them multiple times within epochs.

Table 1 summarises the advantages of PQN in comparison to popular methods. Compared to traditional DQN and distributed DQN, PQN enjoys ease of implementation, fast execution, very low memory requirements, and high compatibility with GPU-based training and RNNs. The only algorithm that shares these attributes is PPO. However, although PPO is in principle a simple algorithm, its success is determined by numerous interacting implementation details (Huang et al., 2022a; Engstrom et al., 2020), making the actual implementation challenging. Moreover, PQN uses few main hyperparameters, namely the number of parallel environments, the learning rate and epsilon with its decay, plus the value for $\lambda$ if $\lambda$-returns are used. We emphasise that, whilst PQN can be run using a single environment interaction at each timestep (i.e. with $I = 1, T = 1$), yielding a stable, regularised $Q$-learning algorithm without a replay buffer (see Fig. 10), PQN is also designed to exploit vectorisation to solve *parallel world problems*, i.e. applications trained in simulators where parallelisation is advantageous and possible.

Table 1: Advantages and Disadvantages of DQN, Distributed DQN, PPO and PQN.

| | DQN | Distr. DQN | PPO | PQN |
|---|---|---|---|---|
| Implementation | Easy | Difficult | Medium | **Very Easy** |
| Memory Requirement | High | Very High | **Low** | **Low** |
| Training Speed | Slow | **Fast** | **Fast** | **Fast** |
| Sample Efficient | **Yes** | No | **Yes** | **Yes** |
| Compatibility with RNNs | Medium | Medium | **High** | **High** |
| Compatibility w. end-to-end GPU Training | Low | Low | **High** | **High** |
| Amount of Hyper-Parameters | Medium | High | Medium | **Low** |
| Convergence | No | No | No | **Yes** |

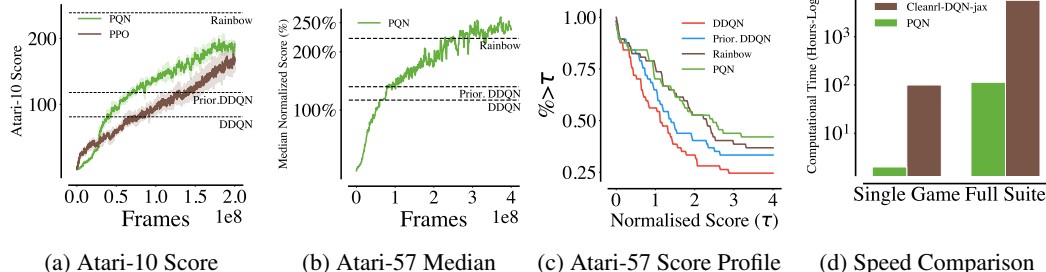

(a) Atari-10 Score     (b) Atari-57 Median     (c) Atari-57 Score Profile     (d) Speed Comparison

Figure 4: (a) Comparison between PPO and PQN in Atari-10. (b) Median score of PQN in the full Atari suite of 57 games. (c) Percentage of games with score higher than human score. (d) Computational time required to run a single game and the full ALE suite for PQN and DQN implementation of CleanRL. In (c) and (d) performances of PQN are relative to training for 400M frames.

### 4.1 BENEFITS OF ONLINE $Q$-LEARNING WITH VECTORISED ENVIRONMENTS

Vectorisation of the environment enables fast collection of many parallel transitions from independent trajectories. Denoting the stationary distribution at time $t$ of the MDP under policy $\pi_t$ as $d^{\pi_t}$, uniformly sampling from a replay buffer containing historic data estimates sampling from the average of all distributions across all timesteps: $\frac{1}{t'+1}\sum_{t=0}^{t'} d^{\pi_t}$. In contrast, vectorised sampling in PQN estimates sampling from the stationary distribution $d^{\pi_{t'}}$ at timestep $t'$. We sketch the difference in these sampling regimes in Fig. 3. Coloured lines represent different state-actions trajectories across the vectorised environment as a function of timestep $t$. Crosses represent samples drawn for each algorithm at timestep $t'$.

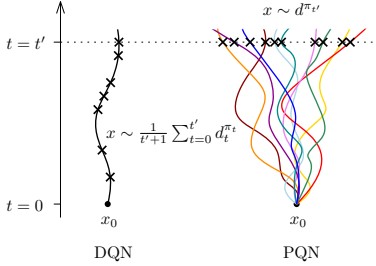

Figure 3: Sketch of Sampling Regimes in DQN and PQN

PQN's sampling further aids algorithmic stability by better approximating this regime in two ways: firstly, the parallelised nature can help exploration since the (potential) natural stochasticity in the dynamics means even a greedy policy will explore several different states in parallel. Secondly, by taking multiple actions in multiple states, PQN's sampling distribution is a good approximation of the true stationary distribution under the current policy: as time progresses, ergodic theory states that this sampling distribution converges to $d^{\pi_{t'}}$. In contrast, sampling from DQN's replay buffer involves sampling from an average of *older* stationary distributions under shifting policies from a single agent, which will be more offline and take longer to converge, as illustrated in Fig. 3. We emphasise that *PQN is still an off-policy approach* since it uses two different policies to optimise the Bellman equations: the $\epsilon$-greedy policy for the current timestep and the current policy for the next. Notice that at beginning of training PQN uses an $\epsilon = 1$, meaning that it approximates a value function from a completely random policy. This requires normalisation to mitigate off-policy instability identified in Section 3.

## 5 EXPERIMENTS

In contrast to prior work in $Q$-learning, which has focused heavily on evaluation in the Atari Learning Environment (ALE) (Bellemare et al., 2013), probably overfitting to this environment, we evaluate PQN on a range of single- and multi-agent environments, with PPO as the primary baseline. We summarise the memory and sample efficiency of PQN in Table 2. Due to our extensive evaluation, additional results are presented in Appendix D. All experimental results are shown as mean of 10 seeds, except in ALE where we followed a common practice of reporting 3 seeds.

### 5.1 CONFIRMING THEORETICAL RESULTS

Fig. 5a shows that together LayerNorm + $\ell^2$ can stabilise TD in Baird's counterexample (Baird, 1995), a challenging environment that is intentionally designed to be provably divergent, even for linear function approximators. Our results show that stabilisation is mostly attributed to the introduction of LayerNorm. Moreover the degree of $\ell^2$-regularisation needed is small - just enough to mitigate off-policy stability due to final layer weights according to Theorem 2 - and it makes relatively little difference when used in isolation.

### 5.2 ATARI

To save computational resources, we evaluate PQN against PPO in the Atari-10 suite of games from the ALE, which estimates the median across the full suite using a smaller sample of games. PQN

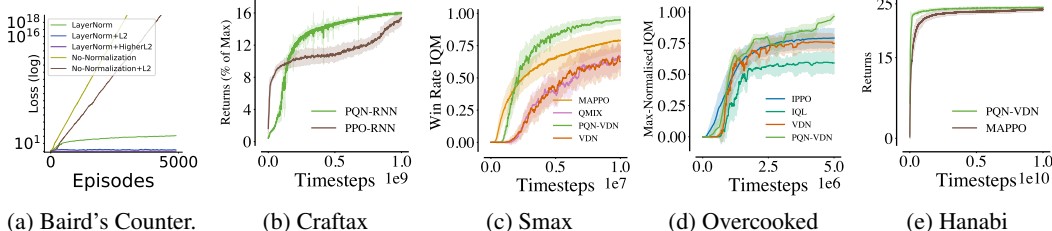

(a) Baird's Counter.  (b) Craftax  (c) Smax  (d) Overcooked  (e) Hanabi

Figure 5: Results in Baird's Counterexample, Craftax and Multi-Agent tasks. For Smax, we report the Interquartile Mean (IQM) of the Win Rate on the 9 most popular maps. For Overcooked, we report the IQM of the returns normalized by the maximum obtained score in the classic 4 layouts. In Hanabi, we report the returns of self-play in the 2-player game.

outperforms PPO in terms of sample efficiency, final score, and training time (1 hour compared to 2.5 hours for PPO), and also surpasses sample-efficient methods like Double-DQN and Prioritised DDQN in the same number of frames, despite these methods being trained for several days and using over 16 times more gradient updates (12.5M compared to 780k for PQN). To further test our method, we train PQN on the full suite of 57 Atari games. Fig. 4d shows that the time needed to train PQN on the full Atari suite is equivalent to the time required to train traditional DQN methods on a single game[1]. With an additional budget of 100M frames (30 minutes of training), PQN achieves the median score of Rainbow (Hessel et al., 2018), which is still a SOTA method in ALE for sample efficiency but requires around 3 days of training per game, meaning that PQN can be considered 50x faster. While Rainbow is slightly more sample efficient, it's important to note that Rainbow is a much more complex system, designed specifically for Atari. Moreover, parallelisation of $Q$-Learning has traditionally sacrificed far more sample efficiency than PQN. For instance, Ape-X struggles to solve even the simplest Atari game, Pong, within 200M frames (Horgan et al., 2018). In this regard, PQN represents a significant advancement in $Q$-Learning research, offering a balanced compromise between speed, simplicity, and sample efficiency.

In Appendix D, we provide detailed data from these experiments, a comparison with Dopamine-Rainbow using the IQM score, and a comparative bar chart (Fig. 13) of the performances of algorithms in all the games. In this chart, we show that PQN reaches human-level performance in 40 of the 57 games of the ALE, underperforming mainly in the hard-exploration games, suggesting that the $\epsilon$-greedy exploration used by PQN is *too* simple to solve ALE, and indicating a clear research direction to improve the method.

### 5.3 CRAFTAX

Craftax (Matthews et al., 2024b) is an open-ended RL environment based on Crafter (Hafner, 2021) and Nethack (Küttler et al., 2020). It is a challenging environment that requires an agent to solve multiple tasks before completion. By design, Craftax is fast to run in a pure-GPU setting, but existing benchmarks are based solely on PPO. The observation size of the symbolic environment is around 8000 floats, making a pure-GPU DQN implementation with a buffer prohibitive, as it would take around 30GBs of GPU-ram. PQN can provide an off-policy $Q$-learning baseline without using GPU memory for a replay buffer. Following the Craftax paper, we evaluate for 1B steps and compared PQN to PPO using both an MLP and an RNN. The RNN results are shown in Fig. 5b. PQN is more sample efficient and with a RNN obtains a higher score of **16%** against the 15.3% of PPO-RNN. The two methods also take a similar amount of time to train. PQN offers researchers a simple, successful $Q$-learning alternative to PPO that can be run on a GPU in this challenging environment.

### 5.4 MULTI-AGENT TASKS

When dealing with multi-agent problems, any replay buffer needs to store observations for all agents, increasing the memory requirements up to hundreds of gigabytes. Additionally, RNNs are highly effective in handling the individual agents' partial observability of the environments and credit assignment, a key challenge in MARL, is typically addressed with value-based methods Sunehag et al. (2017b); Rashid et al. (2020a). Therefore, a memory-efficient, RNN-compatible and value-based method is highly desirable. We evaluate PQN combined with VDN in Hanabi (Bard et al., 2020), SMAC-SMACV2 (Ellis et al., 2024; Samvelyan et al., 2019) (in its JAX-vectorised version, Smax) (Rutherford et al., 2023), and Overcooked (Carroll et al., 2019). Smax is a faster version of SMAC, running entirely on a single GPU. Notably, when at least 20 agents are active in the environment, a replay buffer can consume all available memory on a typical 10GB GPU. PQN-VDN runs successfully on Smax without a large buffer, outperforming MAPPO and QMix. Remarkably, PQN learns a coordination policy even in the most difficult scenarios in about 10 minutes, compared

---

[1]DQN training time was optimistically estimated using the JAX-based CleanRL DQN implementation.

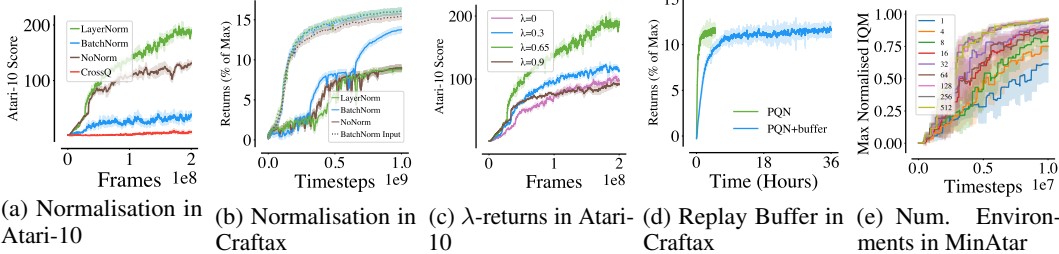

(a) Normalisation in Atari-10  (b) Normalisation in Craftax  (c) $\lambda$-returns in Atari-10  (d) Replay Buffer in Craftax  (e) Num. Environments in MinAtar

Figure 6: Ablations confirming the importance of the different components of our method.

to QMix's 1 hour (see Fig. 17). Similarly, PQN outperforms the replay-buffer-based version of VDN and PPO in Overcooked, and is significantly more sample-efficient than MAPPO in Hanabi, where it achieves an average score of 24 points.

### 5.5 ABLATIONS

To examine the effectiveness of PQN's algorithmic components, we perform the following ablations.

**Regularisation:** In Fig. 6a, we examine the impact of regularisation on performance in the Atari-10 suite. Results show that LayerNorm significantly improves performance, supporting the theoretical findings in Section 3, while BatchNorm can degrade performance when applied through the network. Additionally, applying the additional tricks from CrossQ further worsens PQN's performance.

**Input Normalisation:** In preliminary experiments, we observed that BatchNorm significantly improves PQN performances in Craftax. Figure 6b compares the performance of PQN RNN with BatchNorm, LayerNorm, and no normalisation in the two cases where BatchNorm is applied to the input before the first hidden layer or not. Without input normalisation, BatchNorm provides a substantial boost. However, PQN performs best when only the input to the first layer is batch normalised, and applying LayerNorm to the rest of the network offers a similar improvement. This suggests BatchNorm can be effective as *input normalisation*, particularly in scenarios like Craftax with large, sparse observation vectors.

**Varying $\lambda$:** In Fig. 6c we compare different values of $\lambda$ in Atari-10. We find that a value of $\lambda = 0.65$ performs the best by a significant margin. It significantly outperforms $\lambda = 0$ (which is equal to performing 1-step update with the traditional Bellman operator) confirming that the use of $\lambda$-returns represents an important design choice over one-step TD.

**Replay Buffer:** In Fig. 6d, we compare PQN from with a variant that maintains a standard sized replay buffer of 1M of experiences in GPU using Flashbax Toledo et al. (2023). This version converges to the same final performance but takes ∼6x longer to train, which is likely due to the constant need to perform random access of a buffer of around 30GBs. This reinforces our core message that a large memory buffer should be avoided in pure GPU training.

**Number of Environments:** PQN can learn even with a small number of environments but clearly benefits from collecting more experiences in parallel (Fig. 6e). As expected, PQN is also significantly faster when greater parallelisation is used, (see Fig. 10 in Appendix).

## 6 CONCLUSION

We have presented the first rigorous analysis explaining the stabilising properties of LayerNorm and $\ell^2$ regularisation in TD methods. These results allowed us to develop PQN, a simple, stable and efficient regularised $Q$-learning algorithm without the need for target networks or a large replay buffer. PQN exploits vectorised computation to achieve excellent performance across an extensive empirical evaluation with a significant boost in computational efficiency and without sacrificing sample efficiency. PQN offers a simple pipeline that is easy to implement and out-of-the-box compatible with key elements in RL, such as $\lambda$-returns and RNNs, which

Table 2: Summary of Memory Saved and Speedup of PQN Compared to Baselines. The Atari speedup is relative to the traditional DQN pipeline, which runs a single environment on the CPU while training the network on GPU. Smax and Craftax speedups are relative to baselines that also run entirely on GPU but use a replay buffer. The Hanabi speed-up is relative to an R2D2 multi-threaded implementation.

|  | **Memory Saved** | **Speedup** |
|---|---|---|
| Atari | 26gb | 50x |
| Smax | 10gb (up to hundreds) | 6x |
| Hanabi | 250gb | 4x |
| Craftax | 31gb | 6x |

are otherwise difficult to use in current $Q$-Learning implementations. Additionally, it provides a valuable baseline for multi-agent systems. By saving the memory occupied by large replay buffers, PQN paves the way for a generation of powerful but stable algorithms that exploit end-to-end GPU vectorised deep RL.

REPRODUCIBILITY STATEMENT

All our experiments can be replicated with the following repository: https://github.com/mttga/purejaxql. Proofs for all theorems and corollaries can be found in Appendix B.

ACKNOWLEDGMENTS

Mattie Fellows is funded by a generous grant from the UKRI Engineering and Physical Sciences Research Council EP/Y028481/1. Jakob Nicolaus Foerster is partially funded by the UKRI grant EP/Y028481/1 (originally selected for funding by the ERC). Jakob Nicolaus Foerster is also supported by the JPMC Research Award and the Amazon Research Award.

Matteo Gallici was partially founded by the FPI-UPC Santander Scholarship FPI-UPC_93. Ivan Masmitja is partially founded by the European Union's Horizon Europe programme under grant agreement No 101112883, as part of DIGI4ECO. This work also acknowledges the Spanish Ministerio de Ciencia, Innovacion y Universidades (BITER-ECO: PID2020- 114732RBC31), the the Spanish National Program Ramon y Cajal RYC2022-038056-I (IM) and "Severo Ochoa Centre of Excellence" accreditation (CEX2019-000928-S).

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

# A RELATED WORK

## A.1 ASYNCHRONOUS METHODS AND PARALLELISATION OF $Q$-LEARNING

Existing attempts to parallelise $Q$-learning adopt a distributed architecture, where a separate process continually trains the agent and sampling occurs in parallel threads which contain a delayed copy of its parameters (Horgan et al., 2018; Kapturowski et al., 2018; Badia et al., 2020; Hoffman et al., 2020). On the contrary, PQN samples and trains in the same process, enabling end-to-end single GPU training. While distributed methods can benefit from a separate process that continuously train the network, PQN is easier to implement, doesn't introduce time-lags between the learning agent and the exploratory policy. Moreover, PQN can be optimised to be sample efficient other than fast, while distributed system usually ignore sample efficiency.

Mnih et al. (2016) propose an asynchronous $Q$-learning, a parallelised version of $Q$-learning which performs updates of a centralised network asynchronously. Compared to PQN, asynchronous $Q$-learning still uses target networks and accumulates gradients over many timesteps to update the network. Moreover, it is a multi-threaded approach where each worker independently performs exploration and gradient updates with its own target network. This setup results in each actor being optimised independently with its own experiences and objective, introducing significant noise into the central learner that periodically unifies the gradients. Finally, the algorithm relies on collecting historical data: "We also accumulate gradients over multiple timesteps before they are applied" (Mnih et al., 2016). This undermines a key benefit of parallelised methods, which is avoiding the use of data collected under historic policies (see Section 4.1).

PQN is a synchronous method where a single actor interacts with vectorised environments, and a single gradient is computed at once using all the experiences. PQN can be seen as the synchronous version of that asynchronous Q-Learning algorithm, which has never been implemented before. Note that moving from asynchronous to synchronous, removing the target networks, and avoiding multi-step gradient accumulation drastically changes the optimisation procedure and implementation, resulting in a much simpler and more stable algorithm. To our knowledge, we are the first to unlock the potential of a parallelised deep $Q$-learning algorithm with minimal memory requirements and without target networks.

## A.2 ANALYSIS OF TD

Most prior approaches analysing TD focus on linear function approximation. Tsitsiklis & Van Roy (1997) first proved convergence of linear, on-policy TD, arguing that the projected Bellman operator in this setting is a contraction. Dalal et al. (2017) give the first finite time bounds for linear TD(0), under an i.i.d. data model similar to the one that we use here. Bhandari et al. (2018) provide bounds for linear TD in both the i.i.d. and Markov chain setting. Srikant & Ying (2019) approach the problem from the perspective of Ordinary Differential Equations (ODE) analysis, bounding the divergence of a Lyapunov function from the limiting point of the ODE that arises from the TD update scheme. Analysis of pure TD in the general nonlinear and Markov chain sampling regime is lacking.

Two papers that are most closely related to our work are: (Fellows et al., 2023) and (Yue et al., 2023).

(Yue et al., 2023) analyses the effect of LayerNorm in TD, however there are several important differences. Firstly, the paper analyzes the neural tangent kernel (NTK) of the update, which only exists in the limit of infinite width networks and does not capture the nonlinear instability that we analyse. We make no such assumption as this will never hold in practice. Instead, we use the analysis of Fellows et al. (2023) which predates (Yue et al., 2023) and provides a more general framework for studying TD with finite width *nonlinear networks*. Moreover, Fellows et al. (2023) provide key results on establishing stability of general TD using an eigenvalue analysis this is more general but are remarkably similar to Yue et al. (2023)'s SEEM framework. We extend these results to Markov chain sampling with normalised regularisation.

Yue et al. (2023) claim that LayerNorm alone can stabilise TD. Under our more general and applicable analysis, as our results show, LayerNorm without $\ell^2$ regularisation *cannot completely stabilise TD for all domains*. This is because for stability, the Jacobian eigenvalues need to be strictly negative. As Lemma 2 shows, there may still be a residual positive term that prevents this. Our empirical results in Baird's counterexample confirm this, showing that the algorithm can only be stabilised using normalisation. Existing empirical research (Lyle et al., 2023) also supports this.

### A.3 REGULARISATION IN RL

CrossQ is a recently developed off-policy algorithm based on SAC that removes target networks and introduces BatchNorm into the critic (Bhatt et al., 2024). CrossQ demonstrates impressive performance when evaluated across a suite of MuJoCo domain with high sample and computational efficiency, however no analysis is provided explaining its stability properties or what effect introducing BatchNorm has. To develop PQN, we performed a rigorous analysis of LayerNorm in TD. Here is a complete list of the differences between CrossQ and PQN:

- CrossQ is based on a soft-actor critic architecture for continuous action control. Its entropy-based actor objective optimises a stochastic policy. On the contrary, PQN consists of a single, simple value network optimised with the standard Bellman Equations, which is used to learn a deterministic policy for discrete actions.

- CrossQ is not parallelised, i.e., it interacts with a single environment at a time, while a fundamental contribution of PQN is handling parallel environments for faster training on modern hardware. Parallelisation of Q-Learning algorithms is not trivial: one cannot simply interact with multiple environments while leaving the rest of the learning pipeline unchanged, as this drastically modifies the ratio between interactions with the environments and the number of gradient updates. PQN approaches this problem by offering a sample-efficient implementation based on normalisation, Q-Lambda, and mini-batches/mini-epochs updates.

- CrossQ uses a large replay buffer containing data from historical policies to perform updates, while PQN obtains mini-batches directly from interactions with parallel environments under a single policy.

- CrossQ is not directly compatible with Q-Lambda and Recurrent Neural Networks because of the overhead introduced by the replay buffers: the use of old experiences in the update step makes computation of Q-Lambda unsafe, and the use of hidden states for the RNNs problematic. To include these methods in CrossQ one should add, e.g., Retrace and Burn-In techniques. Conversely, the theoretical absence of a replay buffer in PQN allows us to use them out of the box. Note that these are crucial in many scenarios (see Q-Lambda ablation for Atari and MLP-RNN results in Craftax).

- There is no theoretical analysis of normalisation in CrossQ and empirical evidence limited to six Mujoco continuous-action tasks. This is not sufficient to make any reasonable claims for its performance in general RL scenarios. We give a theoretical basis for our method and we compare it with baselines across 79 discrete-action tasks (2 Classic Control tasks, 4 MinAtar games, 57 Atari games, Craftax, 9 Smax tasks, 5 Overcooked, and Hanabi). The limited evaluation provided for CrossQ is concerning, and the results in Mujoco might not reflect its true capabilities. Our results in Atari demonstrate it.

- PQN is designed for complete GPU implementation and to be compatible with end-to-end compilation, which is a fundamental step for bringing Q-Learning into modern RL research (currently dominated by PPO). CrossQ does not tackle this problem, instead favouring a standard pipeline (which consists of interacting with one environment - sampling from the replay buffer - updating the network - repeat) with the addition of normalisation. This pipeline is exactly the same as that used by DQN in our Atari experiments, where we show that PQN is between 50x and 100x faster and uses 26 times less memory.

The benefits of regularisation have also been reported in other areas of the RL literature. Lyle et al. (Lyle et al., 2023; 2024) investigate plasticity loss in off-policy RL, a phenomenon where neural networks lose their ability to fit a new target functions throughout training. They propose LayerNorm (Lyle et al., 2023) and LayerNorm with $\ell^2$ regularisation (Lyle et al., 2024), as a solution to this problem, and show improved performance on the Atari Learning Environment, but they also use other methods of stabilisation, such as target networks, that we explicitly remove. In addition, they provide no formal analysis explaining stability.

### A.4 MULTI-STEP $Q$-LEARNING

The concept of n-step returns in reinforcement learning extends the traditional one-step update to consider rewards over future timesteps. The $n$-step return for a state-action pair $(s, a)$ is defined as the cumulative reward over the next $n$ steps plus the discounted value of the state reached after $n$ steps. Several variations of $n$-step $Q$-learning have been proposed to enhance learning efficiency and

stability. Peng & Williams (1994) introduced a variation known as $Q(\lambda)$, which integrates eligibility traces to account for multiple time steps while maintaining the off-policy nature of $Q$-learning. Replay buffers are difficult to combine with $Q(\lambda)$, so standard methods like DQN use a single-step TD learning. The most relevant work that aimed to use $Q(\lambda)$ with a replay buffer is Retrace (Munos et al., 2016). More recent methods have tried to reconcile $\lambda$-returns with the experience buffer Daley & Amato (2019) most notably in TD3 Kozuno et al. (2021).

### A.5 Multi-Agent Deep Q Learning

$Q$-learning methods are a popular choice for multi-agent RL (MARL), especially in the purely cooperative centralised training with decentralised execution (CTDE) setting (Foerster et al., 2018; Lowe et al., 2017). In CTDE, global information is made available at training time, but not at test time. Many of these methods develop approaches to combine individual utility functions into a joint estimate of the Q function: Son et al. (2019) introduce the individual-global-max (IGM) principle to describe when a centralised Q function can be computed from individual utility functions in a decentralised fashion; Value Decomposition Networks (VDN) (Sunehag et al., 2017a) combines individual value estimates by summing them, and QMIX (Rashid et al., 2020b) learns a hypernetwork with positive weights to ensure monotonicity. All these methods can be combined with PQN, which parallises the learning process.

IPPO (De Witt et al., 2020) and MAPPO (Yu et al., 2022) use vectorised environments, adapting a single-agent method for use in multi-agent RL. These are both on-policy actor-critic based methods based on PPO.

## B  Proofs and Derivations

### B.1  Derivation of TD Stability Results

We start by examining the TD Jacobian to separate the TD stability condition into two components. From the definition of the TD Jacobian:

$$
\begin{aligned}
J(\phi) = \nabla_\phi \delta(\phi) &= \nabla_\phi \mathbb{E}_{\varsigma \sim P_\varsigma}[\delta(\phi, \varsigma)], \\
&= \mathbb{E}_{\varsigma \sim P_\varsigma}\left[\nabla_\phi\left((r + \gamma Q_\phi(x') - Q_\phi(x))\nabla_\phi Q_\phi(x)\right)\right], \\
&= \gamma \mathbb{E}_{\varsigma \sim P_\varsigma}\left[\nabla_\phi Q_\phi(x')\nabla_\phi Q_\phi(x)^\top\right] - \mathbb{E}_{\varsigma \sim P_\varsigma}\left[\nabla_\phi Q_\phi(x)\nabla_\phi Q_\phi(x)^\top\right] \\
&\quad + \mathbb{E}_{\varsigma \sim P_\varsigma}\left[(r + \gamma Q_\phi(x') - Q_\phi(x))\nabla_\phi^2 Q_\phi(x)\right], \\
&= \gamma \mathbb{E}_{\varsigma \sim P_\varsigma}\left[\nabla_\phi Q_\phi(x')\nabla_\phi Q_\phi(x)^\top\right] - \mathbb{E}_{x \sim d^\mu}\left[\nabla_\phi Q_\phi(x)\nabla_\phi Q_\phi(x)^\top\right] \\
&\quad + \mathbb{E}_{\varsigma \sim P_\varsigma}\left[(r + \gamma Q_\phi(x') - Q_\phi(x))\nabla_\phi^2 Q_\phi(x)\right],
\end{aligned}
$$

hence, we can write the TD Jacobian condition as:

$$
\begin{aligned}
v^\top J(\phi)v &= \gamma \mathbb{E}_{\varsigma \sim P_\varsigma}\left[v^\top \nabla_\phi Q_\phi(x')\nabla_\phi Q_\phi(x)^\top v\right] - \mathbb{E}_{x \sim d^\mu}\left[v^\top \nabla_\phi Q_\phi(x)\nabla_\phi Q_\phi(x)^\top v\right] \\
&\quad + \mathbb{E}_{\varsigma \sim P_\varsigma}\left[(r + \gamma Q_\phi(x') - Q_\phi(x))\,v^\top \nabla_\phi^2 Q_\phi(x)v\right], \\
&= \gamma \mathbb{E}_{\varsigma \sim P_\varsigma}\left[v^\top \nabla_\phi Q_\phi(x')\nabla_\phi Q_\phi(x)^\top v\right] - \mathbb{E}_{x \sim d^\mu}\left[\left(v^\top \nabla_\phi Q_\phi(x)\right)^2\right] \\
&\quad + \mathbb{E}_{\varsigma \sim P_\varsigma}\left[(r + \gamma Q_\phi(x') - Q_\phi(x))\,v^\top \nabla_\phi^2 Q_\phi(x)v\right], \\
&= \mathcal{C}_{\text{OffPolicy}}(Q_\phi^k, d^\mu) + \mathcal{C}_{\text{Nonlinear}}(Q_\phi^k),
\end{aligned}
$$

yielding the two stability components introduced in Section 3.1. Next, we investigate the effect that off-policy sampling has on $\mathcal{C}_{\text{OffPolicy}}(Q_\phi^k, d^\mu)$:

$$
\mathcal{C}_{\text{OffPolicy}}(Q_\phi^k, d^\mu) = \gamma \mathbb{E}_{\varsigma \sim P_\varsigma}\left[v^\top \nabla_\phi Q_\phi(x')\nabla_\phi Q_\phi(x)^\top v\right] - \mathbb{E}_{x \sim d^\mu}\left[\left(v^\top \nabla_\phi Q_\phi(x)\right)^2\right]. \tag{10}
$$

We now apply the Cauchy-Schwarz inequality to separate the expectations in the first term:

$$
\begin{aligned}
\mathbb{E}_{\varsigma \sim P_\varsigma}\left[v^\top \nabla_\phi Q_\phi(x')\nabla_\phi Q_\phi(x)^\top v\right] &\leq \left|\mathbb{E}_{\varsigma \sim P_\varsigma}\left[v^\top \nabla_\phi Q_\phi(x')\nabla_\phi Q_\phi(x)^\top v\right]\right|, \\
&= \sqrt{\left|\mathbb{E}_{\varsigma \sim P_\varsigma}\left[v^\top \nabla_\phi Q_\phi(x')\nabla_\phi Q_\phi(x)^\top v\right]\right|^2}, \\
&\leq \sqrt{\mathbb{E}_{\varsigma \sim P_\varsigma}\left[\left(v^\top \nabla_\phi Q_\phi(x')\right)^2\right]\mathbb{E}_{\varsigma \sim P_\varsigma}\left[\left(v^\top \nabla_\phi Q_\phi(x)\right)^2\right]}, \\
&= \sqrt{\mathbb{E}_{\varsigma \sim P_\varsigma}\left[\left(v^\top \nabla_\phi Q_\phi(x')\right)^2\right]\mathbb{E}_{x \sim d^\pi}\left[\left(v^\top \nabla_\phi Q_\phi(x)\right)^2\right]}.
\end{aligned}
$$

Substituting into Eq. (10) yields:

$$\mathcal{C}_{\text{OffPolicy}}(Q_\phi^k, d^\mu) \leq \gamma \sqrt{\mathbb{E}_{\varsigma \sim P_\varsigma}\left[(v^\top \nabla_\phi Q_\phi(x'))^2\right] \mathbb{E}_{x \sim d^\pi}\left[(v^\top \nabla_\phi Q_\phi(x))^2\right]}$$
$$- \mathbb{E}_{x \sim d^\mu}\left[(v^\top \nabla_\phi Q_\phi(x))^2\right].$$

Now, as $\gamma \in [0, 1)$, to prove that $\mathcal{C}_{\text{OffPolicy}}(Q_\phi^k, d^\mu) < 0$, we require that $\mathbb{E}_{\varsigma \sim P_\varsigma}\left[(v^\top \nabla_\phi Q_\phi(x'))^2\right] \leq \mathbb{E}_{x \sim d^\pi}\left[(v^\top \nabla_\phi Q_\phi(x))^2\right]$, yielding:

$$\mathcal{C}_{\text{OffPolicy}}(Q_\phi^k, d^\mu) \leq \gamma \sqrt{\mathbb{E}_{x \sim d^\pi}\left[(v^\top \nabla_\phi Q_\phi(x))^2\right]^2} - \mathbb{E}_{x \sim d^\mu}\left[(v^\top \nabla_\phi Q_\phi(x))^2\right],$$
$$= (\gamma - 1)\mathbb{E}_{x \sim d^\mu}\left[(v^\top \nabla_\phi Q_\phi(x))^2\right],$$
$$< 0.$$

## B.2 THEOREM 1 - ANALYSING TD

We now characterise the convergence of TD in our general setting. Our proof is structured as follows: we first bound the expected norm one timestep into the future in terms of the expected norm at the current timestep: $\mathbb{E}_{\varsigma_i, -\varsigma_i}\left[\|\phi_{i+1} - \phi^\star\|^2\right] \leq \text{Constant} \cdot \mathbb{E}_{-\varsigma_i}\left[\|\phi_i - \phi^\star\|^2\right] + \text{Residual}_i$ where $\text{Residual}_i$ is a residual term that accounts for the variance of the updates and sampling from the Markov chain. This is done by expanding $\|\phi_{i+1} - \phi^\star\|^2$ and following the algebra to Ineq. 11 of Theorem 1. To bound the residual term, we then invoke Lemma 1. Bounding the variance contribution results naturally from our Lipschitz assumption. Bounding the Markov contribution follows from the definition of geometric ergodicity and our proof is similar to Bhandari et al. (2018). Crucially, this bound implies $\lim_{i \to \infty} \text{Residual}_i = 0$. We then use the fundamental theorem of calculus to show that the TD stability criterion implies Constant $< 1$ for small enough $\alpha_i$ (see Eq. (12)). This demonstrates that the TD updates are a contraction mapping with a decaying residual term, allowing us to verify convergence in the remainder of the proof.

**Theorem 1** (TD Stability). *Let Assumptions 1 and 2 hold. If the TD criterion holds then the TD updates in Eq.* (1) *converge with:*

$$\lim_{i \to \infty} \mathbb{E}\left[\|\phi_i - \phi^\star\|^2\right] = 0.$$

*Proof.* We use the notation $\mathbb{E}_{-\varsigma_i}[\cdot]$ to denote the expectation over $\{\varsigma_0, \ldots \varsigma_{i-1}\}$ and $\mathbb{E}_{\varsigma_i|\varsigma_{i-1}}[\cdot]$ to denote the expectation over $\varsigma_i$ conditioned on $\varsigma_{i-1}$. Substituting for $\phi_{i+1} = \phi_i + \alpha_i \delta(\phi_i, \varsigma_i)$ into $\mathbb{E}\left[\|\phi_{i+1} - \phi^\star\|^2\right]$ yields:

$$\mathbb{E}_{\varsigma_i, -\varsigma_i}\left[\|\phi_{i+1} - \phi^\star\|^2\right] = \mathbb{E}_{\varsigma_i, -\varsigma_i}\left[\|\phi_i + \alpha_i \delta(\phi_i, \varsigma_i) - \phi^\star\|^2\right],$$

$$= \mathbb{E}_{\varsigma_i, -\varsigma_i}\left[\|\phi_i - \phi^\star\|^2 + 2\alpha_i \delta(\phi_i, \varsigma_i)^\top(\phi_i - \phi^\star) + \alpha_i^2\|\delta(\phi_i, \varsigma_i)\|^2\right],$$

$$= \mathbb{E}_{-\varsigma_i}\left[\|\phi_i - \phi^\star\|^2 + 2\alpha_i \mathbb{E}_{\varsigma_i|-\varsigma_i}\left[\delta(\phi_i, \varsigma_i)\right]^\top(\phi_i - \phi^\star) + \alpha_i^2 \mathbb{E}_{\varsigma_i|-\varsigma_i}\left[\|\delta(\phi_i, \varsigma_i)\|^2\right]\right],$$

$$= \mathbb{E}_{-\varsigma_i}\Big[\|\phi_i - \phi^\star\|^2 + 2\alpha_i \left(\mathbb{E}_{\varsigma_i|-\varsigma_i}\left[\delta(\phi_i, \varsigma_i)\right] - \delta(\phi_i) + \delta(\phi_i)\right)^\top(\phi_i - \phi^\star)$$
$$+ \alpha_i^2 \mathbb{E}_{\varsigma_i|-\varsigma_i}\left[\|\delta(\phi_i, \varsigma_i)\|^2\right]\Big],$$

$$= \mathbb{E}_{-\varsigma_i}\Big[\|\phi_i - \phi^\star\|^2 + 2\alpha_i \delta(\phi_i)^\top(\phi_i - \phi^\star)$$
$$+ 2\alpha_i \left(\mathbb{E}_{\varsigma_i|-\varsigma_i}\left[\delta(\phi_i, \varsigma_i)\right] - \delta(\phi_i)\right)^\top(\phi_i - \phi^\star) + \alpha_i^2 \mathbb{E}_{\varsigma_i|-\varsigma_i}\left[\|\delta(\phi_i, \varsigma_i)\|^2\right]\Big],$$

$$\leq \mathbb{E}_{-\varsigma_i}\Big[\|\phi_i - \phi^\star\|^2 + 2\alpha_i \delta(\phi_i)^\top(\phi_i - \phi^\star)\Big]$$
$$+ 2\alpha_i \underbrace{\left|\mathbb{E}_{-\varsigma_i}\left[\left(\mathbb{E}_{\varsigma_i|-\varsigma_i}\left[\delta(\phi_i, \varsigma_i)\right] - \delta(\phi_i)\right)^\top(\phi_i - \phi^\star)\right]\right|}_{\text{Non i.i.d. term}} + \alpha_i^2 \underbrace{\mathbb{E}_{\varsigma_i, -\varsigma_i}\left[\|\delta(\phi_i, \varsigma_i)\|^2\right]}_{\text{Variance term}}, \quad (11)$$

where we have isolated the contribution of variance and non-i.i.d. sampling in deriving the final line. We now bound the non-i.i.d. contribution in total variation and variance term using Lemma 1:

$$
\mathbb{E}_{\varsigma_i, -\varsigma_i} \left[ \|\phi_{i+1} - \phi^\star\|^2 \right] \leq \mathbb{E}_{-\varsigma_i} \left[ \|\phi_i - \phi^\star\|^2 + 2\alpha_i \delta(\phi_i)^\top (\phi_i - \phi^\star) \right] + 2\alpha_i C_{\text{Markov}} \rho^i + \alpha_i^2 C_{\text{Var}}.
$$

Note that for i.i.d. sampling, $\mathbb{E}_{\varsigma_i | -\varsigma_i} [\delta(\phi_i, \varsigma_i)] = \mathbb{E}_{\varsigma_i \sim P_\varsigma} [\delta(\phi_i, \varsigma_i)] = \delta(\phi_i)$ and so $C_{\text{Markov}} = 0$. Next, we re-write $\delta(\phi_i)$ to contain a factor of $\phi_i - \phi^\star$. Define the line joining $\phi^\star$ to $\phi_i$ as $\ell(l) = \phi_i - l(\phi_i - \phi^\star)$. Under Assumption 2, we can apply the fundamental theorem of calculus to integrate along this line, yielding:

$$
\begin{aligned}
\delta(\phi_i) &= \delta(\phi_i) - \underbrace{\delta(\phi^\star)}_{=0}, \\
&= \delta(\phi = \ell(0)) - \delta(\phi = \ell(1)), \\
&= -\int_0^1 \partial_l \delta(\phi = \ell(l)) dl, \\
&= \int_0^1 \nabla_\phi \delta(\phi = \ell(l))(\phi_i - \phi^\star) dl, \\
&= \int_0^1 J(\phi = \ell(l)) dl (\phi_i - \phi^\star), \\
&= \tilde{J}(\phi_i - \phi^\star)
\end{aligned}
$$

where we have used the chain rule to derive the fourth line and introduced the notation $\tilde{J} := \int_0^1 J(\phi = \ell(l)) dl$. Substituting yields:

$$
\begin{aligned}
\mathbb{E}_{\varsigma_i, -\varsigma_i} \left[ \|\phi_{i+1} - \phi^\star\|^2 \right] \leq & \mathbb{E}_{-\varsigma_i} \left[ \|\phi_i - \phi^\star\|^2 + 2\alpha_i (\phi_i - \phi^\star)^\top \tilde{J} (\phi_i - \phi^\star) \right] \\
& + 2\alpha_i C_{\text{Markov}} \rho^i + \alpha_i^2 C_{\text{Var}}.
\end{aligned}
$$

Now, as the TD criterion: $v^\top J(\phi) v < 0$ holds almost everywhere, it follows that:

$$
(\phi_i - \phi^\star)^\top \underbrace{\int_0^1 J(\phi = \ell(l)) dl}_{:=\tilde{J}} (\phi_i - \phi^\star) < 0,
$$

$$
\implies (\phi_i - \phi^\star)^\top \tilde{J} (\phi_i - \phi^\star) = (\phi_i - \phi^\star)^\top \frac{1}{2} \left( \tilde{J} + \tilde{J}^\top \right) (\phi_i - \phi^\star) \leq -\lambda_{\min} \|\phi_i - \phi^\star\|^2,
$$

where $\lambda_{\min} > 0$ is the smallest (in magnitude) eigenvalue of $-\frac{1}{2}(\tilde{J} + \tilde{J}^\top)$. Substituting yields:

$$
\mathbb{E}_{\varsigma_i, -\varsigma_i} \left[ \|\phi_{i+1} - \phi^\star\|^2 \right] \leq \mathbb{E}_{-\varsigma_i} \left[ \|\phi_i - \phi^\star\|^2 \right] (1 - 2\alpha_i \lambda_{\min}) + 2\alpha_i C_{\text{Markov}} \rho^i + \alpha_i^2 C_{\text{Var}}. \quad (12)
$$

Re-arranging yields:

$$
\begin{aligned}
2\lambda_{\min} \alpha_i \mathbb{E}_{-\varsigma_i} & \left[ \|\phi_i - \phi^\star\|^2 \right] \\
& \leq \mathbb{E}_{-\varsigma_i} \left[ \|\phi_i - \phi^\star\|^2 \right] - \mathbb{E}_{\varsigma_i, -\varsigma_i} \left[ \|\phi_{i+1} - \phi^\star\|^2 \right] + 2\alpha_i C_{\text{Markov}} \rho^i + \alpha_i^2 C_{\text{Var}}.
\end{aligned}
$$

Summing over $i$ up to timestep $t$ and using the telescoping property of the series yields:

$$2\lambda_{\min} \sum_{i=0}^{t} \alpha_i \mathbb{E}_{-\varsigma_i} \left[ \|\phi_i - \phi^\star\|^2 \right]$$

$$\leq \mathbb{E}_{-\varsigma_i} \left[ \|\phi_0 - \phi^\star\|^2 \right] - \mathbb{E}_{\varsigma_i, -\varsigma_i} \left[ \|\phi_{t+1} - \phi^\star\|^2 \right] + 2C_{\text{Markov}} \sum_{i=0}^{t} \alpha_i \rho^i + C_{\text{Var}} \sum_{i=0}^{t} \alpha_i^2,$$

$$\leq \mathbb{E}_{-\varsigma_i} \left[ \|\phi_0 - \phi^\star\|^2 \right] + 2C_{\text{Markov}} \sum_{i=0}^{t} \alpha_i \rho^i + C_{\text{Var}} \sum_{i=0}^{t} \alpha_i^2,$$

$$\implies \sum_{i=0}^{t} \frac{\alpha_i}{\sum_{i'=0}^{t} \alpha_{i'}} \mathbb{E}_{-\varsigma_i} \left[ \|\phi_i - \phi^\star\|^2 \right]$$

$$\leq \frac{1}{2\lambda_{\min}} \left( \frac{\mathbb{E}_{-\varsigma_i} \left[ \|\phi_0 - \phi^\star\|^2 \right]}{\sum_{i'=0}^{t} \alpha_{i'}} + 2C_{\text{Markov}} \frac{\sum_{i=0}^{t} \alpha_i \rho^i}{\sum_{i=0}^{t} \alpha_i} + C_{\text{Var}} \frac{\sum_{i=0}^{t} \alpha_i^2}{\sum_{i'=0}^{t} \alpha_{i'}} \right), \tag{13}$$

where the penultimate bound follows from $\mathbb{E}_{\varsigma_i, -\varsigma_i} \left[ \|\phi_{t+1} - \phi^\star\|^2 \right] \geq 0$. In preparation for taking the limit $t \to \infty$, we observe that by the Cauchy-Schwarz inequality:

$$\sum_{i=0}^{t} \alpha_i \rho^i = \sum_{i=0}^{t} |\alpha_i| |\rho^i| \leq \sqrt{\sum_{i=0}^{t} \alpha_i^2 \rho^{2i}} \leq \sqrt{\sum_{i=0}^{t} \alpha_i^2 \sum_{i=0}^{t} \rho^{2i}}.$$

Now, from Assumption 1, $\lim_{t \to \infty} \sum_{i=0}^{t} \alpha_i^2 < \infty$, hence:

$$\lim_{t \to \infty} \sum_{i=0}^{t} \alpha_i \rho^i \leq \sqrt{\lim_{t \to \infty} \sum_{i=0}^{t} \alpha_i^2 \lim_{t \to \infty} \sum_{i=0}^{t} \rho^{2i}} = \mathcal{O}(1)$$

As $\lim_{t \to \infty} \sum_{i=0}^{t} \alpha_i = \infty$, this implies:

$$\lim_{t \to \infty} \frac{\sum_{i=0}^{t} \alpha_i \rho^i}{\sum_{i=0}^{t} \alpha_i} = 0.$$

We are now ready to take limits of Inq. 13, yielding:

$$\lim_{t \to \infty} \sum_{i=0}^{t} \frac{\alpha_i}{\sum_{i'=0}^{t} \alpha_{i'}} \mathbb{E}_{-\varsigma_i} \left[ \|\phi_i - \phi^\star\|^2 \right] = 0. \tag{14}$$

Eq. (14) proves our desired result:

$$\lim_{i \to \infty} \mathbb{E}_{-\varsigma_i} \left[ \|\phi_i - \phi^\star\|^2 \right] = 0.$$

To see why, assume this does not hold, that is $\lim_{i \to \infty} \mathbb{E}_{-\varsigma_i} \left[ \|\phi_i - \phi^\star\|^2 \right] \neq 0$. This implies there exists some infinite length sub-sequence $S$ such that for all $i \in S$:

$$\mathbb{E}_{-\varsigma_i} \left[ \|\phi_i - \phi^\star\|^2 \right] > 0,$$

hence, as all quantities are positive:

$$\lim_{t \to \infty} \sum_{i=0}^{t} \frac{\alpha_i}{\sum_{i'=0}^{t} \alpha_{i'}} \mathbb{E}_{-\varsigma_i} \left[ \|\phi_i - \phi^\star\|^2 \right] \geq \lim_{t \to \infty} \sum_{i \in S} \frac{\alpha_i}{\sum_{i'=0}^{t} \alpha_{i'}} \mathbb{E}_{-\varsigma_i} \left[ \|\phi_i - \phi^\star\|^2 \right] > 0,$$

which is a contradiction. $\qquad \square$

**Lemma 1.** *Let Assumption 2 hold. Then there exist constants:* $0 < C_{\text{Markov}} < \infty$, $0 < C_{\text{Var}} < \infty$ *and* $\rho \in [0, 1)$ *such that:*

$$\left| \mathbb{E}_{-\varsigma_i} \left[ \left( \mathbb{E}_{\varsigma_i | -\varsigma_i} [\delta(\phi_i, \varsigma_i)] - \delta(\phi_i) \right)^\top (\phi_i - \phi^\star) \right] \right| \leq C_{\text{Markov}} \rho^i, \quad \mathbb{E}_{\varsigma_i, -\varsigma_i} \left[ \|\delta(\phi_i, \varsigma_i)\|^2 \right] \leq C_{\text{Var}}.$$

*Proof.* For both results, we use the fact that, because $\Phi$ is compact and $\mathcal{X}$ is bounded, rewards are bounded and $\delta(\phi, \varsigma)$ is Lipschitz under Assumption 2, $\delta(\phi, \varsigma)$ is bounded almost everywhere. To prove the first bound, we denote the marginal probability distribution of the $i$-th timestep element in the Markov chain $\varsigma_i$ as $P^i$ with density:

$$p^i(\varsigma_i) = \int p(-\varsigma_i, \varsigma_i) d(-\varsigma_i).$$

Under this notation we write:

$$
\begin{aligned}
&\mathbb{E}_{-\varsigma_i} \left[ \left( \mathbb{E}_{\varsigma_i | -\varsigma_i} \left[ \delta(\phi_i, \varsigma_i) \right] - \delta(\phi_i) \right)^\top (\phi_i - \phi^\star) \right] \\
&= \mathbb{E}_{-\varsigma_i, \varsigma_i} \left[ \left( \delta(\phi_i, \varsigma_i) - \mathbb{E}_{\varsigma_i' \sim P_\varsigma} \left[ \delta(\phi_i, \varsigma_i') \right] \right)^\top (\phi_i - \phi^\star) \right], \\
&= \mathbb{E}_{-\varsigma_i, \varsigma_i} \left[ \delta(\phi_i, \varsigma_i)^\top (\phi_i - \phi^\star) - \mathbb{E}_{\varsigma_i' \sim P_\varsigma} \left[ \delta(\phi_i, \varsigma_i') \right]^\top (\phi_i - \phi^\star) \right], \\
&= \mathbb{E}_{-\varsigma_i, \varsigma_i} \left[ \delta(\phi_i, \varsigma_i)^\top (\phi_i - \phi^\star) - \mathbb{E}_{\varsigma_i' \sim P_\varsigma} \left[ \delta(\phi_i, \varsigma_i')^\top (\phi_i - \phi^\star) \right] \right], \\
&= \mathbb{E}_{\varsigma_i \sim P^i} \left[ \mathbb{E}_{-\varsigma_i \sim P^i(\varsigma_i)} \left[ \delta(\phi_i, \varsigma_i)^\top (\phi_i - \phi^\star) \right] \right] - \mathbb{E}_{\varsigma_i \sim P_\varsigma} \left[ \mathbb{E}_{-\varsigma_i \sim P^i(\varsigma_i)} \left[ \delta(\phi_i, \varsigma_i)^\top (\phi_i - \phi^\star) \right] \right],
\end{aligned}
\tag{15}
$$

where $P^i(\varsigma_i)$ is the backwards conditional distribution in the Markov chain with density:

$$p^i(-\varsigma_i | \varsigma_i) = \frac{p^i(-\varsigma_i, \varsigma_i)}{p^i(\varsigma_i)}.$$

Introducing the notation:

$$g(\varsigma_i) = \mathbb{E}_{-\varsigma_i \sim P^i(\varsigma_i)} \left[ \delta(\phi_i, \varsigma_i)^\top (\phi_i - \phi^\star) \right],$$

we write Eq. (15) as:

$$
\begin{aligned}
\mathbb{E}_{-\varsigma_i} \left[ \left( \mathbb{E}_{\varsigma_i | -\varsigma_i} \left[ \delta(\phi_i, \varsigma_i) \right] - \delta(\phi_i) \right)^\top (\phi_i - \phi^\star) \right] &= \mathbb{E}_{\varsigma_i \sim P^i} \left[ g(\varsigma_i) \right] - \mathbb{E}_{\varsigma_i \sim P_\varsigma} \left[ g(\varsigma_i) \right], \\
&= \mathbb{E}_{\varsigma_0} \left[ \mathbb{E}_{\varsigma_i \sim P^i(\varsigma_0)} \left[ g(\varsigma_i) \right] - \mathbb{E}_{\varsigma_i \sim P_\varsigma} \left[ g(\varsigma_i) \right] \right], \\
&= \mathbb{E}_{\varsigma_0} \left[ g_{\max} \left( \mathbb{E}_{\varsigma_i \sim P^i(\varsigma_0)} \left[ \frac{g(\varsigma_i)}{g_{\max}} \right] - \mathbb{E}_{\varsigma_i \sim P_\varsigma} \left[ \frac{g(\varsigma_i)}{g_{\max}} \right] \right) \right],
\end{aligned}
\tag{16}
$$

where $g_{\max} := \max_\varsigma |g(\varsigma)| < \infty$ almost everywhere, which follows from the fact that $\delta(\phi, \varsigma)$ is bounded almost everywhere, implying $g(\varsigma)$ is also bounded almost everywhere. Now, as $\frac{g(\cdot)}{g_{\max}} : \mathcal{X} \times \mathbb{R} \times \mathcal{X} \to [-1, 1]$, we can bound Eq. (16) in total variation using Roberts & Rosenthal (2004)[Proposition 3b]:

$$
\begin{aligned}
\left| \mathbb{E}_{\varsigma_0} \left[ g_{\max} \left( \mathbb{E}_{\varsigma_i \sim P^i(\varsigma_0)} \left[ \frac{g(\varsigma_i)}{g_{\max}} \right] - \mathbb{E}_{\varsigma_i \sim P_\varsigma} \left[ \frac{g(\varsigma_i)}{g_{\max}} \right] \right) \right] \right| &\leq \\
\mathbb{E}_{\varsigma_0} \left[ \left| g_{\max} \left( \mathbb{E}_{\varsigma_i \sim P^i(\varsigma_0)} \left[ \frac{g(\varsigma_i)}{g_{\max}} \right] - \mathbb{E}_{\varsigma_i \sim P_\varsigma} \left[ \frac{g(\varsigma_i)}{g_{\max}} \right] \right) \right| \right] & \\
= 2 g_{\max} \mathbb{E}_{\varsigma_0} \left[ \frac{1}{2} \left| \mathbb{E}_{\varsigma_i \sim P^i(\varsigma_0)} \left[ \frac{g(\varsigma_i)}{g_{\max}} \right] - \mathbb{E}_{\varsigma_i \sim P_\varsigma} \left[ \frac{g(\varsigma_i)}{g_{\max}} \right] \right| \right], & \\
\leq 2 g_{\max} \mathbb{E}_{\varsigma_0} \left[ \frac{1}{2} \sup_{f : \mathcal{X} \times \mathbb{R} \times \mathcal{X} \to [-1, 1]} \left| \mathbb{E}_{\varsigma_i \sim P^i(\varsigma_0)} \left[ f(\varsigma_i) \right] - \mathbb{E}_{\varsigma_i \sim P_\varsigma} \left[ f(\varsigma_i) \right] \right| \right], & \\
= 2 g_{\max} \mathbb{E}_{\varsigma_0} \left[ \text{TV}(P^i(\varsigma_0) \| P_\varsigma) \right], &
\end{aligned}
\tag{17}
$$

where $\text{TV}(P^i(\varsigma_0) \| P_\varsigma)$ is the total variational distance between the marginal distribution $P^i(\varsigma_0)$ (conditioned on initial observations) and the steady state distribution $P_\varsigma$. Now, as the Markov chain is geometricaly ergodic, by definition there exists some function $M(\varsigma_0)$ and constant $\rho \in [0, 1)$ such that:

$$\text{TV}(P^i(\varsigma_0) \| P_\varsigma) \leq M(\varsigma_0) \rho^i,$$

almost surely (see Roberts & Rosenthal (2004)[Section 3.4]), hence substituting into Eq. (17) yields our desired result:

$$
\begin{aligned}
\left| \mathbb{E}_{-\varsigma_i} \left[ \left( \mathbb{E}_{\varsigma_i | -\varsigma_i} \left[ \delta(\phi_i, \varsigma_i) \right] - \delta(\phi_i) \right)^\top (\phi_i - \phi^\star) \right] \right| &\leq 2 g_{\max} \mathbb{E}_{\varsigma_0} \left[ \mathrm{TV}(P^i(\varsigma_0) \| P_\varsigma) \right], \\
&\leq 2 g_{\max} \mathbb{E}_{\varsigma_0} \left[ M(\varsigma_0) \rho^i \right], \\
&= 2 g_{\max} \mathbb{E}_{\varsigma_0} \left[ M(\varsigma_0) \right] \rho^i, \\
&= C_{\mathrm{Markov}} \rho^i,
\end{aligned}
$$

where $C_{\mathrm{Markov}} := 2 g_{\max} \mathbb{E}_{\varsigma_0} \left[ M(\varsigma_0) \right] < \infty$. Our second bound follows from the fact that $\delta(\phi, \varsigma)$ is bounded almost everywhere. This implies there exists some $C_{\mathrm{Var}} > 0$ such that $\| \delta(\phi, \varsigma) \|^2 \leq C_{\mathrm{Var}}$ almost everywhere, hence:

$$
\mathbb{E}_{\varsigma_i, -\varsigma_i} \left[ \| \delta(\phi_i, \varsigma_i) \|^2 \right] \leq C_{\mathrm{Var}}.
$$

$\square$

### B.3 THEOREM 2 - STABILISING TD WITH LAYERNORM AND $\ell^2$-REGULARISATION

**Notation:** For all proofs in this section, we introduce the following simplifying notations:

$$
\begin{aligned}
f_M(x) &:= \sigma_{\mathrm{Pre}} \circ M x, \\
\mathrm{LayerNorm}_i^k[f](x) &:= \frac{1}{\sqrt{k}} \cdot \frac{f_i(x) - \hat{\mu}[f](x)}{\hat{\sigma}[f](x)},
\end{aligned}
$$

where $\hat{\mu}[f](x)$ and $\hat{\sigma}[f](x)$ are the element-wise empirical mean and standard deviation of the output $f(x)$:

$$
\hat{\mu}[f](x) := \frac{1}{k} \sum_{i=0}^{k-1} f_i(x), \quad \hat{\sigma}[f](x) := \sqrt{\frac{1}{k} \sum_{i=0}^{k-1} (f_i(x) - \hat{\mu}[f](x))^2 + \epsilon},
$$

Finally, we write $M$ in terms of its row vectors:

$$
M = \begin{bmatrix} - & m_0^T & - \\ - & m_1^\top & - \\ & \vdots & \\ - & m_{k-1}^\top & - \end{bmatrix}.
$$

and split the test vector into the corresponding $k + 1$ sub-vectors:

$$
v^\top = [v_w^T, v_{m_0}^\top, v_{m_1}^\top, \cdots v_{m_{k-1}}^\top],
$$

where $v_w$ is a vector with the same dimension as the final weight vector $w$ and each $v_{m_i} \in \mathbb{R}^n$ has the same dimension as $x$. We will make use of the following three key properties of LayerNorm:

**Proposition 1.** *Let $f : \mathcal{X} \to \mathbb{R}^k$ be a vector-valued function such that all components $f_i$ are bounded, then:*

$$
\begin{aligned}
\| \mathrm{LayerNorm}^k[f(x)] \| &\leq 1, \\
\partial_{f_i} \mathrm{LayerNorm}_j^k[f(x)] &= \mathcal{O}\left( k^{-\frac{1}{2}} \left( \mathbb{1}(i = j) + \frac{1}{k} \right) \right), \\
\partial_{f_s} \partial_{f_t} \mathrm{LayerNorm}_j^k[f(x)] &= \mathcal{O}\left( k^{-\frac{3}{2}} \left( \mathbb{1}(t = j) + \mathbb{1}(t = s) + \mathbb{1}(j = s) + \frac{1}{k} \right) \right).
\end{aligned}
$$

*Proof.* Our first result follows directly from the definition of LayerNorm:

$$\|\text{LayerNorm}^k[f(x)]\| = \frac{1}{\sqrt{k}} \frac{\|f(x) - \hat{\mu}[f](x)\|}{\hat{\sigma}[f](x)},$$

$$= \frac{\sqrt{\frac{1}{k} \sum_{i=0}^{k-1} (f_i(x) - \hat{\mu}[f](x))^2}}{\hat{\sigma}[f](x)},$$

$$\leq \frac{\sqrt{\frac{1}{k} \sum_{i=0}^{k-1} (f_i(x) - \hat{\mu}[f](x))^2 + \epsilon}}{\hat{\sigma}[f](x)},$$

$$= \frac{\hat{\sigma}[f](x)}{\hat{\sigma}[f](x)},$$

$$= 1,$$

as required. For our second result, we take partial derivatives with respect to the $i$th input channel to the LayerNorm:

$$\partial_{f_i} \text{LayerNorm}_j^k[f(x)] = \frac{1}{\sqrt{k}} \left( \frac{\mathbb{1}(i=j) - \frac{1}{k}}{\hat{\sigma}[f](x)} - \frac{f_j(x) - \hat{\mu}[f](x)}{\hat{\sigma}[f](x)^2} \partial_{f_i} \hat{\sigma}[f](x) \right),$$

$$= \frac{1}{\sqrt{k}} \left( \frac{\mathbb{1}(i=j) - \frac{1}{k}}{\hat{\sigma}[f](x)} - \sqrt{k} \frac{\text{LayerNorm}_j^k[f(x)]}{\hat{\sigma}[f](x)} \partial_{f_i} \hat{\sigma}[f](x) \right).$$

Finding the derivative of the empirical variance yields:

$$\partial_{f_i} \hat{\sigma}[f](x) = \partial_{f_i} \sqrt{\frac{1}{k} \sum_{i=0}^{k-1} (f_i(x) - \hat{\mu}[f](x))^2 + \epsilon},$$

$$= \frac{1}{2} \left( \frac{1}{k} \sum_{i=0}^{k-1} (f_i(x) - \hat{\mu}[f](x))^2 + \epsilon \right)^{-\frac{1}{2}} \partial_{f_i} \left( \frac{1}{k} \sum_{i=0}^{k-1} (f_i(x) - \hat{\mu}[f](x))^2 + \epsilon \right),$$

$$= \frac{1}{k\hat{\sigma}[f](x)} \sum_{l=0}^{k-1} (f_l(x) - \hat{\mu}[f](x)) \left( \mathbb{1}(i=l) - \frac{1}{k} \right),$$

$$= \frac{1}{k\hat{\sigma}[f](x)} \left( \sum_{l=0}^{k-1} (f_l(x) - \hat{\mu}[f](x)) \mathbb{1}(i=l) - \frac{1}{k} \sum_{l=0}^{k-1} f_l(x) + \hat{\mu}[f](x) \sum_{l=0}^{k-1} \frac{1}{k} \right),$$

$$= \frac{1}{k\hat{\sigma}[f](x)} \left( f_i(x) - \hat{\mu}[f](x) - \underbrace{\frac{1}{k} \sum_{l=0}^{k-1} f_l(x)}_{=\hat{\mu}[f](x)} + \hat{\mu}[f](x) \right),$$

$$= \frac{f_i(x) - \hat{\mu}[f](x)}{k\hat{\sigma}[f](x)},$$

$$= \frac{1}{\sqrt{k}} \text{LayerNorm}_i^k[f(x)],$$

hence:

$$\partial_{f_i} \text{LayerNorm}_j^k[f(x)]$$

$$= \frac{1}{\sqrt{k}\hat{\sigma}[f](x)} \left( \mathbb{1}(i=j) - \frac{1}{k} - \text{LayerNorm}_i^k[f(x)] \text{LayerNorm}_j^k[f(x)] \right), \qquad (18)$$

$$= \mathcal{O} \left( \frac{1}{\sqrt{k}} \left( \mathbb{1}(i=j) + \frac{1}{k} \right) \right),$$

where we use the fact that $\text{LayerNorm}_j^k[f(x)] = \mathcal{O}\left(\frac{1}{\sqrt{k}}\right)$ to derive the final line. To prove our third result, we start with the first order partial derivative using Eq. (18):

$$\partial_{f_t} \text{LayerNorm}_j^k[f(x)] = \frac{1}{\sqrt{k}\hat{\sigma}[f](x)} \left( \mathbb{1}(t=j) - \frac{1}{k} - \text{LayerNorm}_t^k[f(x)] \text{LayerNorm}_j^k[f(x)] \right),$$

Taking partial derivatives with respect to $f_s$ yields:

$$\partial_{f_s}\partial_{f_t}\text{LayerNorm}_j^k[f(x)]$$

$$= -\frac{\partial_{f_s}\hat{\sigma}[f](x)}{\sqrt{k}\hat{\sigma}[f](x)^2}\left(\mathbb{1}(t=j) - \frac{1}{k} - \text{LayerNorm}_t^k[f(x)]\text{LayerNorm}_j^k[f(x)]\right)$$

$$- k^{-\frac{1}{2}}\cdot\frac{\partial_{f_s}\text{LayerNorm}_t^k[f(x)]\cdot\text{LayerNorm}_j^k[f(x)] + \text{LayerNorm}_t^k[f(x)]\partial_{f_s}\text{LayerNorm}_j^k[f(x)]}{\hat{\sigma}[f](x)},$$

$$= \mathcal{O}\left(k^{-\frac{3}{2}}\left(\mathbb{1}(t=j)+\frac{1}{k}\right)\right) + \mathcal{O}\left(k^{-\frac{3}{2}}\left(\mathbb{1}(t=s)+\frac{1}{k}\right)\right) + \mathcal{O}\left(k^{-\frac{3}{2}}\left(\mathbb{1}(j=s)+\frac{1}{k}\right)\right),$$

$$= \mathcal{O}\left(k^{-\frac{3}{2}}\left(\mathbb{1}(t=j) + \mathbb{1}(t=s) + \mathbb{1}(j=s) + \frac{1}{k}\right)\right),$$

as required. $\qquad\square$

We are now ready to prove our main result. Most of the work is done by proving Lemma 2: once the bounds in Lemma 2 have been established, the result follows by subtracting the regularisation term from the off policy and nonlinear components of the TD stability condition. We split the proof of Lemma 2 into two parts. Firstly, we bound the off-policy contribution by splitting it further into components that affect the final layer weights and the other matrix weights. By doing so, we find a residual term remains that is only affected by the final layer weights (Lemma 3). Secondly, we bound the non-linear contribution in Lemma 4 by isolating the second order derivative of the function approximator. What remains is to show this term decays as $1/\sqrt{k}$, which we prove in Lemma 5. Our proof of Lemma 5 is similar to Liu et al. (2020).

**Theorem 2.** *Let Assumption 2 apply. Using the* LayerNorm *regularised TD update $\delta_{reg}^k(\phi,\varsigma)$ in Eq. (9), there exists some finite $k'$ such that the TD stability criterion holds for all $k > k'$*

*Proof.* From the definition of the expected regularised TD error vector:

$$\delta_{\text{reg}}^k(\phi) = \mathbb{E}_{\varsigma\sim P_\varsigma}\left[\left(r + \gamma Q_\phi^k(x') - Q_\phi^k(x)\right)\nabla_\phi Q_\phi^k(x)\right]$$

$$- \left(\eta\left(\frac{\gamma L_{\text{Post}}}{2}\right)^2\begin{bmatrix}w\\0\end{bmatrix} + (\eta-1)\begin{bmatrix}0\\\text{Vec}(M)\end{bmatrix}\right),$$

$$\implies v^\top\nabla_\phi\delta_{\text{reg}}^k(\phi)v = \mathbb{E}_{\varsigma\sim P_\varsigma}\left[\left(r + \gamma Q_\phi^k(x') - Q_\phi^k(x)\right)v^\top\nabla_\phi^2 Q_\phi^k(x)v\right]$$

$$+ \mathbb{E}_{\varsigma\sim P_\varsigma}\left[v^\top(\gamma\nabla_\phi Q_\phi^k(x') - \nabla_\phi Q_\phi^k(x))\nabla_\phi Q_\phi^k(x)^\top v\right]$$

$$- \eta\left(\frac{\gamma L_{\text{Post}}}{2}\right)^2\|v_w\|^2 - (\eta-1)\|v_{-m}\|^2,$$

$$= \mathcal{C}_{\text{OffPolicy}}(Q_\phi^k, d^\mu) + \mathcal{C}_{\text{Nonlinear}}(Q_\phi^k)$$

$$- \eta\left(\frac{\gamma L_{\text{Post}}}{2}\right)^2\|v_w\|^2 - (\eta-1)\|v_{-m}\|^2.$$

Applying Lemma 2 and taking the limit $k\to\infty$ yields:

$$\lim_{k\to\infty}v^\top\nabla_\phi\delta_{\text{reg}}^k(\phi)v = \left(\frac{\gamma L_{\text{Post}}}{2}\right)^2(1-\eta)\|v_w\|^2 + (1-\eta)\|v_M\|^2 < 0,$$

almost everywhere, which follows from the fact $\eta > 1$, hence, by the definition of the limit, there must exist some finite $k'$ such that for all $k > k'$:

$$v^\top\nabla_\phi\delta_{\text{reg}}^k(\phi)v < 0,$$

almost everywhere, as required. $\qquad\square$

**Lemma 2.** *Let Assumption 2 apply. Let $v_w$ be the first $k$ components of the test vector $v = [v_w^\top, v_M^\top]^\top$, associated with final layer parameters $w$, and $v_M$ be the remaining components, associated with the matrix $M$ parameters. Using the* LayerNorm *Q-function defined in Eq. (5):*

Off-Policy Bound: $\qquad\mathcal{C}_{\text{OffPolicy}}(Q_\phi^k, d^\mu) \leq \|v_w\cdot\gamma L_{Post}/2\|^2 + \mathcal{O}\left(\|v_M\|^2/k\right),$

Nonlinear Bound: $\qquad\mathcal{C}_{\text{Nonlinear}}(Q_\phi^k) = \mathcal{O}\left(\|v\|^2/\sqrt{k}\right),$

*almost surely for any test vector and any state-action transition pair $x, x'\in\mathcal{X}$.*

*Proof.* By definition of the off-policy and nonlinear contribution terms:

$$\mathcal{C}_{\text{OffPolicy}}(Q_\phi^k, d^\mu) := \gamma \mathbb{E}_{\varsigma \sim P_\varsigma} \left[ v^\top \nabla_\phi Q_\phi^k(x') v^\top \nabla_\phi Q_\phi^k(x) \right] - \mathbb{E}_{x \sim d^\mu} \left[ \left( v^\top \nabla_\phi Q_\phi^k(x) \right)^2 \right],$$

$$\mathcal{C}_{\text{Nonlinear}}(Q_\phi^k) := \mathbb{E}_{\varsigma \sim P_\varsigma} \left[ (r + \gamma Q_\phi^k(x') - Q_\phi^k(x)) v^\top \nabla_\phi^2 Q_\phi^k(x) v \right].$$

Applying Lemma 3 and Lemma 4 yields:

$$\mathcal{C}_{\text{OffPolicy}}(Q_\phi^k, d^\mu) = \mathbb{E}_{\varsigma \sim P_\varsigma} \left[ \gamma v^\top \nabla_\phi Q_\phi^k(x') v^\top \nabla_\phi Q_\phi^k(x) - \left( v^\top \nabla_\phi Q_\phi^k(x) \right)^2 \right],$$

$$\leq \mathbb{E}_{\varsigma \sim P_\varsigma} \left[ \left( \frac{\gamma L_{\text{Post}} \|v_w\|}{2} \right)^2 + \mathcal{O}\left( \frac{\|v_M\|^2}{k} \right) \right],$$

$$= \left( \frac{\gamma L_{\text{Post}} \|v_w\|}{2} \right)^2 + \mathcal{O}\left( \frac{\|v_M\|^2}{k} \right),$$

$$\mathcal{C}_{\text{Nonlinear}}(Q_\phi^k) := \mathbb{E}_{\varsigma \sim P_\varsigma} \left[ (r + \gamma Q_\phi^k(x') - Q_\phi^k(x)) v^\top \nabla_\phi^2 Q_\phi^k(x) v \right],$$

$$\leq \mathbb{E}_{\varsigma \sim P_\varsigma} \left[ \left| (r + \gamma Q_\phi^k(x') - Q_\phi^k(x)) v^\top \nabla_\phi^2 Q_\phi^k(x) v \right| \right],$$

$$= \mathcal{O}\left( \frac{\|v\|^2}{\sqrt{k}} \right),$$

as required. $\square$

**Lemma 3** (Mitigating Off-policy Instability). *Under Assumption 2, using the* LayerNorm *critic in Eq. (5):*

$$\gamma v^\top \nabla_\phi Q_\phi^k(x') \nabla_\phi Q_\phi^k(x)^\top v - (v^\top \nabla_\phi Q_\phi^k(x))^2 \leq \left( \frac{\gamma L_{Post} \|v_w\|}{2} \right)^2 + \mathcal{O}\left( \frac{\|v_M\|^2}{k} \right), \qquad (19)$$

*almost surely for any test vector and any state-action transition pair $x, x' \in \mathcal{X}$.*

*Proof.* Using the notation introduced at the start of Appendix B.3, we start by splitting the left hand side of Eq. (19) into two terms, one determining the stability of the final layer weights and one for the matrix vectors:

$$\gamma v^\top \nabla_\phi Q_\phi^k(x') \nabla_\phi Q_\phi^k(x)^\top v - (v^\top \nabla_\phi Q_\phi^k(x))^2$$
$$= \gamma v_w^\top \nabla_w Q_\phi^k(x') \nabla_w Q_\phi^k(x)^\top v_w - (v_w^\top \nabla_w Q_\phi^k(x))^2$$
$$+ \sum_{i=0}^{k-1} \left( \gamma v_{m_i}^\top \nabla_{m_i} Q_\phi^k(x') \nabla_{m_i} Q_\phi^k(x)^\top v_{m_i} - v_{m_i}^\top \nabla_{m_i} Q_\phi^k(x) \nabla_{m_i} Q_\phi^k(x)^\top v_{m_i} \right). \qquad (20)$$

We first focus on the term determining stability of the final layer weights. Taking derivatives of the critic with respect to the final layer weights $w$ yields:

$$\nabla_w Q_\phi^k(x') = \sigma_{\text{Post}} \circ \text{LayerNorm}^k [f_M(x')],$$

hence:

$$\|\nabla_w Q_\phi^k(x')\| = \|\sigma_{\text{Post}} \circ \text{LayerNorm}^k [f_M(x')]\|,$$
$$= \|\sigma_{\text{Post}} \circ \text{LayerNorm}^k [f_M(x')] - \underbrace{\sigma_{\text{Post}}(0)}_{=0}\|,$$
$$\leq L_{\text{Post}} \|\text{LayerNorm}^k [f_M(x')] - 0\|,$$
$$= L_{\text{Post}} \|\text{LayerNorm}^k [f_M(x')]\|,$$
$$\leq L_{\text{Post}}, \qquad (21)$$

where we have used the fact that $\sigma_{\text{Post}}(\cdot)$ is $L_{\text{Post}}$-Lipschitz to derive the third line and applied $\|\text{LayerNorm}^k [f_M(x')]\| \leq 1$ from Proposition 1 to derive the final line. We then bound $v_w^\top \nabla_w Q_\phi^k(x') \nabla_w Q_\phi^k(x)^\top v_w$ as:

$$v_w^\top \nabla_w Q_\phi^k(x') \nabla_w Q_\phi^k(x)^\top v_w \leq \|v_w\| \|\nabla_w Q_\phi^k(x')\| |\nabla_w Q_\phi^k(x)^\top v_w|,$$
$$\leq L_{\text{Post}} \|v_w\| |\nabla_w Q_\phi^k(x)^\top v_w|.$$

Defining $\epsilon := |\nabla_w Q_\phi^k(x)^\top v_w|$ yields:

$$\gamma v_w^\top \nabla_w Q_\phi^k(x') \nabla_w Q_\phi^k(x)^\top v_w - (v_w^\top \nabla_w Q_\phi^k(x))^2 \leq \gamma \|v_w\| |\nabla_w Q_\phi^k(x)^\top v_w| - |\nabla_w Q_\phi^k(x)^\top v_w|^2,$$
$$= \gamma L_{\text{Post}} \|v_w\| \epsilon - \epsilon^2,$$
$$\leq \max_\epsilon \left( \gamma L_{\text{Post}} \|v_w\| \epsilon - \epsilon^2 \right).$$

Our desired result follows from the fact that the function $\gamma L_{\text{Post}} \|v_w\| \epsilon - \epsilon^2$ is maximised at $\epsilon = \frac{\gamma L_{\text{Post}} \|v_w\|}{2}$, hence:

$$\gamma v_w^\top \nabla_w Q_\phi^k(x') \nabla_w Q_\phi^k(x)^\top v_w - (v_w^\top \nabla_w Q_\phi^k(x))^2 \leq \frac{\gamma^2 L_{\text{Post}}^2 \|v_w\|^2}{2} - \left( \frac{\gamma L_{\text{Post}} \|v_w\|}{2} \right)^2$$
$$= \left( \frac{\gamma L_{\text{Post}} \|v_w\|}{2} \right)^2$$

Substituting into Eq. (20) yields:

$$\gamma v^\top \nabla_\phi Q_\phi^k(x') \nabla_\phi Q_\phi^k(x)^\top v - (v^\top \nabla_\phi Q_\phi^k(x))^2 \leq \left( \frac{\gamma L_{\text{Post}} \|v_w\|}{2} \right)^2$$
$$+ \sum_{i=0}^{k-1} \left( \gamma v_{m_i}^\top \nabla_{m_i} Q_\phi^k(x') \nabla_{m_i} Q_\phi^k(x)^\top v_{m_i} - v_{m_i}^\top \nabla_{m_i} Q_\phi^k(x) \nabla_{m_i} Q_\phi^k(x)^\top v_{m_i} \right). \qquad (22)$$

We now bound the remaining terms (i.e. those that characterise stability of the matrix row vectors) by taking derivatives of the critic with respect to each matrix row vector: $m_i$ :

$$\nabla_{m_i} Q_\phi^k(x) = \nabla_{m_i} w^\top \sigma_{\text{Post}} \circ \text{LayerNorm}^k \left[ f_M(x) \right],$$
$$= \sum_{j=0}^{k-1} w_j \nabla_{m_i} \sigma_{\text{Post}} (\text{LayerNorm}_j^k \left[ f_M(x) \right]).$$

Applying the chain rule to find an expression for the derivative:

$$\nabla_{m_i} Q_\phi^k(x) = \sum_{j=0}^{k-1} w_j \sigma'_{\text{Post}} (\text{LayerNorm}_j^k \left[ f_M(x) \right]) \nabla_{m_i} \text{LayerNorm}_j^k \left[ f_M(x) \right],$$
$$= \sum_{j=0}^{k-1} w_j \sigma'_{\text{Post}} (\text{LayerNorm}_j^k \left[ f_M(x) \right]) \partial_{f_i} \text{LayerNorm}_j^k \left[ f_M(x) \right] \sigma'_{\text{Pre}} (m_i^\top x) x,$$

where $\sigma'_{\text{Pre}}$ and $\sigma'_{\text{Post}}$ denote the derivatives of the activation functions, which are bounded almost surely from the Lipschitz assumption, hence:

$$\sigma'_{\text{Pre}}(m_i^\top x), \; \sigma'_{\text{Post}} \left( \text{LayerNorm}_j^k \left[ f_M(x) \right] \right) = \mathcal{O}(1).$$

Using this, we bound $\left| \nabla_{m_i} Q_\phi^k(x)^\top v_{m_i} \right|$ as:

$$\left| \nabla_{m_i} Q_\phi^k(x)^\top v_{m_i} \right| \leq \sum_{j=0}^{k-1} \mathcal{O}(1) w_j \partial_{f_i} \text{LayerNorm}_j^k \left[ f_M(x) \right] v_{m_i}^\top x.$$

Now, as each element $f_{M,i}(x)$ is Lipschitz and defined over a bounded set of parameters $m_i$ and inputs $\mathcal{X}$, it follows that $f_{M,i}(x)$ must be a bounded function. We can thus apply the derivative bound to $\partial_{f_i} \text{LayerNorm}_j^k [f_M(x)]$ from Proposition 1, yielding:

$$\left| \nabla_{m_i} Q_\phi^k(x)^\top v_{m_i} \right| \leq \left( \sum_{j=0}^{k-1} \mathcal{O} \left( \frac{\mathbb{1}(i=j) - \frac{1}{k}}{\sqrt{k}} \right) w_j \right) v_{m_i}^\top x,$$
$$= \left( \sum_{j=0}^{k-1} \mathcal{O} \left( \frac{\mathbb{1}(i=j) - \frac{1}{k}}{\sqrt{k}} \right) \right) v_{m_i}^\top x,$$
$$= \mathcal{O} \left( \frac{1}{\sqrt{k}} \right) v_{m_i}^\top x,$$

where we have used the fact that $w_j = \mathcal{O}(1)$ in deriving the second line. Finally, we use this result to bound each $\nabla_{m_i} Q_\phi^k(x')^\top v_{m_i}$ and $\nabla_{m_i} Q_\phi^k(x)^\top v_{m_i}$ term in Eq. (22):

$$\gamma v^\top \nabla_\phi Q_\phi^k(x') \nabla_\phi Q_\phi^k(x)^\top v - (v^\top \nabla_\phi Q_\phi^k(x))^2 \leq \left( \frac{\gamma L_{\text{Post}} \|v_w\|}{2} \right)^2$$

$$+ \sum_{i=0}^{k-1} \left( \gamma \left| v_{m_i}^\top \nabla_{m_i} Q_\phi^k(x') \right| \left| \nabla_{m_i} Q_\phi^k(x)^\top v_{m_i} \right| + \left| v_{m_i}^\top \nabla_{m_i} Q_\phi^k(x) \right|^2 \right),$$

$$\leq \left( \frac{\gamma L_{\text{Post}} \|v_w\|}{2} \right)^2 + \mathcal{O}\left( \frac{1}{k} \right) \sum_{i=0}^{k-1} \left( \gamma \left| v_{m_i}^\top x' \right| \left| v_{m_i}^\top x \right| + \left| v_{m_i}^\top x \right|^2 \right),$$

$$\leq \left( \frac{\gamma L_{\text{Post}} \|v_w\|}{2} \right)^2 + \mathcal{O}\left( \frac{1}{k} \right) \left( \sum_{i=0}^{k-1} \|v_{m_i}\|^2 \right) \left( \gamma \|x'\| \|x\| + \|x\|^2 \right).$$

Now, it follows from the definition of the euclidean norm:

$$\sum_{i=0}^{k-1} \|v_{m_i}\|^2 = \|v_M\|^2,$$

and by the definition of the state-action space of the MDP in Section 2.1:

$$\|x\|, \ \|x'\| = \mathcal{O}(1),$$

hence:

$$\gamma v^\top \nabla_\phi Q_\phi^k(x') \nabla_\phi Q_\phi^k(x)^\top v - (v^\top \nabla_\phi Q_\phi^k(x))^2 \leq \left( \frac{\gamma L_{\text{Post}} \|v_w\|}{2} \right)^2 + \mathcal{O}\left( \frac{\|v_M\|^2}{k} \right),$$

as required. $\qquad\square$

**Lemma 4** (Mitigating Nonlinear Instability). *Under Assumption 2, using the* LayerNorm *Q-function defined in Eq. (5):*

$$\left| \left( r + \gamma Q_\phi^k(x') - Q_\phi^k(x) \right) v^\top \nabla_\phi^2 Q_\phi^k(x) v \right| = \mathcal{O}\left( \frac{\|v\|^2}{\sqrt{k}} \right),$$

*almost surely for any test vector and any state-action transition pair $x, x' \in \mathcal{X}$.*

*Proof.* We start by bounding the TD error, second order derivative and test vector separately:

$$\left| \left( r + \gamma Q_\phi^k(x') - Q_\phi^k(x) \right) v^\top \nabla_\phi^2 Q_\phi^k(x) v \right| \leq \left| \left( r + \gamma Q_\phi^k(x') - Q_\phi^k(x) \right) \right| \left\| \nabla_\phi^2 Q_\phi^k(x) \right\| \|v\|^2,$$

By the definition of the LayerNorm $Q$-function:

$$|Q_\phi^k(x)| \leq \|w\| \left\| \sigma_{\text{Post}} \circ \text{LayerNorm}^k \left[ f_M(x) \right] \right\|,$$
$$\leq \|w\| L_{\text{Post}}.$$

where the final line follows from Eq. (21). As the reward is bounded by definition and $\|w\|$ is bounded under Assumption 2, we can bound the TD error vector as:

$$\left| \left( r + \gamma Q_\phi^k(x') - Q_\phi^k(x) \right) \right| = \mathcal{O}(1),$$

hence:

$$\left| \left( r + \gamma Q_\phi^k(x') - Q_\phi^k(x) \right) v^\top \nabla_\phi^2 Q_\phi^k(x) v \right| = \left\| \nabla_\phi^2 Q_\phi^k(x) \right\| \|v\|^2.$$

Our result follows immediately by using Lemma 5 to bound the second order derivative:

$$\left| \left( r + \gamma Q_\phi^k(x') - Q_\phi^k(x) \right) v^\top \nabla_\phi^2 Q_\phi^k(x) v \right| = \mathcal{O}\left( \frac{\|v\|^2}{\sqrt{k}} \right).$$

$\qquad\square$

**Lemma 5.** *Let Assumption 2 hold.* $\|\nabla_\phi^2 Q_\phi^k(x)\| = \mathcal{O}\left(\frac{1}{\sqrt{k}}\right).$

*Proof.* Using the notation introduced at the start of Appendix B.3, we denote the partial derivative with respect to the $i, j$-th matrix element as: $\partial_{m_{i,j}} \text{LayerNorm}_l^k[f_M(x)]$. Using the chain rule, we find the partial derivatives with respect to each element as:

$$\partial_{m_{i,j}} \text{LayerNorm}_l^k\left[f_M(x)\right] = \partial_{f_i} \text{LayerNorm}_l^k\left[f_M(x)\right] \partial_{m_{i,j}} f_i,$$
$$= \partial_{f_i} \text{LayerNorm}_l^k\left[f_M(x)\right] \sigma'_{\text{pre}}(m_i^\top x) x_j,$$

where $\sigma'_{\text{Pre}}$ denotes the derivatives of the activation function, which is bounded almost surely from the Lipschitz assumption, hence applying Proposition 1 it follows:

$$\partial_{m_{i,j}} \text{LayerNorm}_l^k\left[f_M(x)\right] = \mathcal{O}\left(\partial_{f_i} \text{LayerNorm}_l^k\left[f_M(x)\right]\right),$$
$$= \mathcal{O}\left(k^{-\frac{1}{2}}\left(\mathbb{1}(l=i) + \frac{1}{k}\right)\right).$$

We find a similar result for the second order derivative:

$$\partial_{m_{s,t}} \partial_{m_{i,j}} \text{LayerNorm}_l^k\left[f_M(x)\right] = \partial_{f_i} \text{LayerNorm}_l^k\left[f_M(x)\right] \sigma''_{\text{pre}}(m_i^\top x) x_j x_t \mathbb{1}(i=s)$$
$$+ \partial_{f_s} \partial_{f_i} \text{LayerNorm}_l^k\left[f_M(x)\right] \partial_{m_{s,t}} f_s,$$
$$= \partial_{f_i} \text{LayerNorm}_l^k\left[f_M(x)\right] \sigma''_{\text{pre}}(m_i^\top x) x_j x_t \mathbb{1}(i=s)$$
$$+ \partial_{f_s} \partial_{f_i} \text{LayerNorm}_l^k\left[f_M(x)\right] \sigma'_{\text{pre}}(m_s^\top x) x_t,$$

where $\sigma''_{\text{pre}}$ denotes the second order derivative, which is bounded by assumption, hence:

$$\partial_{m_{s,t}} \partial_{m_{i,j}} \text{LayerNorm}_l^k\left[f_M(x)\right]$$
$$= \mathcal{O}\left(\partial_{f_i} \text{LayerNorm}_l^k\left[f_M(x)\right]\right) \mathbb{1}(i=s) + \mathcal{O}\left(\partial_{f_s} \partial_{f_i} \text{LayerNorm}_l^k\left[f_M(x)\right]\right),$$
$$= \mathcal{O}\left(\mathbb{1}(i=s)k^{-\frac{1}{2}}\left(\mathbb{1}(l=i) + \frac{1}{k}\right)\right) + \mathcal{O}\left(k^{-\frac{3}{2}}\left(\mathbb{1}(l=i) + \mathbb{1}(i=s) + \mathbb{1}(l=s) + \frac{1}{k}\right)\right),$$

We now use these results find the partial derivatives of the LayerNorm $Q$-function. Starting with $\partial_{w_u} \partial_{m_{i,j}} Q_\phi^k(x)$:

$$\partial_{w_u} \partial_{m_{i,j}} Q_\phi^k(x) = \partial_{m_{s,t}} \sum_{l=0}^{k-1} w_l \partial_{m_{i,j}} \sigma_{\text{Post}}(\text{LayerNorm}_l^k[f_M(x)]),$$

$$= \partial_{w_u} \sum_{l=0}^{k-1} w_l \sigma'_{\text{Post}}(\text{LayerNorm}_l^k[f_M(x)]) \partial_{m_{i,j}} \text{LayerNorm}_l^k[f_M(x)],$$

$$= \partial_{w_u} \sum_{l=0}^{k-1} w_l \mathcal{O}\left(k^{-\frac{1}{2}}\right)\left(\mathbb{1}(l=i) + \mathcal{O}\left(\frac{1}{k}\right)\right),$$

$$= \mathcal{O}\left(k^{-\frac{1}{2}}\right)\left(\mathbb{1}(u=i) + \mathcal{O}\left(\frac{1}{k}\right)\right), \tag{23}$$

and now for $\partial_{m_{s,t}}\partial_{m_{i,j}}Q_\phi^k(x)$:

$$\partial_{m_{s,t}}\partial_{m_{i,j}}Q_\phi^k(x) = \partial_{m_{s,t}}\sum_{l=0}^{k-1}w_l\sigma'_{\text{Post}}(\text{LayerNorm}_l^k[f_M(x)])\partial_{m_{i,j}}\text{LayerNorm}_l^k[f_M(x)],$$

$$= \sum_{l=0}^{k-1}w_l\big(\sigma'_{\text{Post}}(\text{LayerNorm}_l^k[f_M(x)])\partial_{m_{s,t}}\partial_{m_{i,j}}\text{LayerNorm}_l^k[f_M(x)]$$

$$+ \sigma''_{\text{Post}}(\text{LayerNorm}_l^k[f_M(x)])\partial_{m_{s,t}}\text{LayerNorm}_l^k[f_M(x)]\partial_{m_{i,j}}\text{LayerNorm}_l^k[f_M(x)]\big),$$

$$= \sum_{l=0}^{k-1}w_l\Bigg(\mathcal{O}\left(\mathbb{1}(i=s)k^{-\frac{1}{2}}\left(\mathbb{1}(l=i)+\frac{1}{k}\right)\right) +$$

$$+ \mathcal{O}\left(k^{-\frac{3}{2}}\left(\mathbb{1}(l=i)+\mathbb{1}(i=s)+\mathbb{1}(l=s)+\frac{1}{k}\right)\right)$$

$$+ \mathcal{O}\left(\frac{1}{k}\right)\left(\mathbb{1}(l=s)+\mathcal{O}\left(\frac{1}{k}\right)\right)\left(\mathbb{1}(l=i)+\mathcal{O}\left(\frac{1}{k}\right)\right)\Bigg),$$

$$= \mathcal{O}\left(\mathbb{1}(i=s)k^{-\frac{1}{2}}\right)+\mathcal{O}\left(k^{-\frac{1}{2}}\right)\left(\mathbb{1}(i=s)+\mathcal{O}\left(\frac{1}{k}\right)\right)$$

$$+ \sum_{l=0}^{k-1}w_l\mathcal{O}\left(\frac{1}{k}\right)\left(\mathbb{1}(l=s)+\mathcal{O}\left(\frac{1}{k}\right)\right)\left(\mathbb{1}(l=i)+\mathcal{O}\left(\frac{1}{k}\right)\right),$$

$$= \mathcal{O}\left(k^{-\frac{1}{2}}\right)\left(\mathbb{1}(i=s)+\mathcal{O}\left(\frac{1}{k}\right)\right)+\mathcal{O}\left(\frac{1}{k}\right)\mathbb{1}(i=s)+\mathcal{O}\left(\frac{1}{k^2}\right),$$

$$= \mathcal{O}\left(k^{-\frac{1}{2}}\right)\left(\mathbb{1}(i=s)+\mathcal{O}\left(\frac{1}{k}\right)\right), \tag{24}$$

where $\sigma''_{\text{Post}}(x)$ denotes the second order derivative and we have used the fact that $\sigma'_{\text{Post}}(\cdot)$ and $\sigma''_{\text{Post}}(\cdot)$ are bounded by assumption.

Now, from the definition of the Matrix 2-norm:

$$\|\nabla_\phi^2 Q_\phi^k(x)\| = \sup_v \frac{v^\top \nabla_\phi^2 Q_\phi^k(x)v}{v^\top v},$$

for any test vector. As the $Q$-function is linear in $w$, we can ignore second order derivatives with respect to elements of $w$ as their value is zero. The matrix norm can then be written in terms of the partial derivatives of $Q_\phi^k(x)$ as:

$$\|\nabla_\phi^2 Q_\phi^k(x)\| = \sup_v \frac{1}{v^\top v}\Bigg(\sum_{u=0}^{k-1}\sum_{i=0}^{k-1}\sum_{j=0}^{d-1}v_{m_{i,j}}\partial_{w_u}\partial_{m_{i,j}}Q_\phi^k(x)v_{w_u}$$

$$+ \sum_{i=0}^{k-1}\sum_{j=0}^{d-1}\sum_{s=0}^{k-1}\sum_{t=0}^{d-1}v_{m_{i,j}}\partial_{m_{s,t}}\partial_{m_{i,j}}Q_\phi^k(x)v_{m_{s,t}}\Bigg).$$

We now bound the partial derivatives using:

$$\partial_{w_u}\partial_{m_{i,j}}Q_\phi^k(x) = \mathcal{O}\left(k^{-\frac{1}{2}}\right)\left(\mathbb{1}(u=i)+\mathcal{O}\left(\frac{1}{k}\right)\right),$$

$$\partial_{m_{s,t}}\partial_{m_{i,j}}Q_\phi^k(x) = \mathcal{O}\left(k^{-\frac{1}{2}}\right)\left(\mathbb{1}(i=s)+\mathcal{O}\left(\frac{1}{k}\right)\right),$$

from Eq. (23) and Eq. (24), yielding:

$$
\begin{aligned}
\|\nabla_\phi^2 Q_\phi^k(x)\| =& \mathcal{O}\left(k^{-\frac{1}{2}}\right) \sup_v \frac{1}{v^\top v} \left( \sum_{u=0}^{k-1} \sum_{i=0}^{k-1} \sum_{j=0}^{d-1} v_{m_{i,j}} v_{w_u} \left( \mathbb{1}(u=i) + \mathcal{O}\left(\frac{1}{k}\right) \right) \right. \\
& \left. + \sum_{i=0}^{k-1} \sum_{j=0}^{d-1} \sum_{s=0}^{k-1} \sum_{t=0}^{d-1} v_{m_{i,j}} v_{m_{s,t}} \left( \mathbb{1}(i=s) + \mathcal{O}\left(\frac{1}{k}\right) \right) \right), \\
=& \mathcal{O}\left(k^{-\frac{1}{2}}\right) \sup_v \frac{1}{v^\top v} \left( \sum_{u=0}^{k-1} v_{w_u} \sum_{j=0}^{d-1} \sum_{i=0}^{k-1} v_{m_{i,j}} \left( \mathbb{1}(u=i) + \mathcal{O}\left(\frac{1}{k}\right) \right) \right. \\
& \left. + \sum_{i=0}^{k-1} \sum_{j=0}^{d-1} v_{m_{i,j}} \sum_{s=0}^{k-1} \sum_{t=0}^{d-1} v_{m_{s,t}} \left( \mathbb{1}(i=s) + \mathcal{O}\left(\frac{1}{k}\right) \right) \right), \\
=& \mathcal{O}\left(k^{-\frac{1}{2}}\right) \sup_v \frac{1}{v^\top v} \left( \sum_{u=0}^{k-1} v_{w_u} \mathcal{O}(1) + \sum_{i=0}^{k-1} \sum_{j=0}^{d-1} v_{m_{i,j}} \mathcal{O}(1) \right), \\
=& \mathcal{O}\left(k^{-\frac{1}{2}}\right) \sup_v \frac{1}{v^\top v} \mathcal{O}\left( \sum_{u=0}^{k-1} v_{w_u}^2 + \sum_{i=0}^{k-1} \sum_{j=0}^{d-1} v_{m_{i,j}}^2 \right).
\end{aligned}
$$

Using the definition $v^\top v := \sum_{u=0}^{k-1} v_{w_u}^2 + \sum_{i=0}^{k-1} \sum_{j=0}^{d-1} v_{m_{i,j}}^2$ yields our desired result:

$$
\begin{aligned}
\|\nabla_\phi^2 Q_\phi^k(x)\| =& \mathcal{O}\left(k^{-\frac{1}{2}}\right) \sup_v \frac{1}{v^\top v} \mathcal{O}\left(v^\top v\right), \\
=& \mathcal{O}\left(k^{-\frac{1}{2}}\right) \sup_v \mathcal{O}\left(1\right), \\
=& \mathcal{O}\left(k^{-\frac{1}{2}}\right).
\end{aligned}
$$

$\square$

### B.4 DERIVATION OF RECURSIVE $\lambda$-RETURNS.

The original proof can be found in Daley & Amato (2019, Appendix D), which we repeat and adapt here for convenience. We wish to write $R_t^\lambda$ as a function of $R_{t+1}^\lambda$. First, note the general recursive relationship between $n$-step returns:

$$R_k^{(n)} = r_k + \gamma R_{k+1}^{(n-1)} \tag{25}$$

Let $N = T - t$. Starting with the definition of the $\lambda$-return,

$$
\begin{aligned}
R_t^\lambda &= \left[ (1-\lambda) \sum_{n=1}^{N-1} \lambda^{n-1} R_t^{(n)} + \lambda^{N-1} R_t^{(N)} \right] \\
&= (1-\lambda) R_t^{(1)} + \left[ (1-\lambda) \sum_{n=2}^{N-1} \lambda^{n-1} R_t^{(n)} + \lambda^{N-1} R_t^{(N)} \right] \\
&= (1-\lambda) R_t^{(1)} + \left[ (1-\lambda) \sum_{n=2}^{N-1} \lambda^{n-1} \left( r_t + \gamma R_{t+1}^{(n-1)} \right) + \lambda^{N-1} \left( r_t + \gamma R_{t+1}^{(N-1)} \right) \right] \\
&= (1-\lambda) R_t^{(1)} + \lambda r_t + \gamma\lambda \left[ (1-\lambda) \sum_{n=2}^{N-1} \lambda^{n-2} R_{t+1}^{(n-1)} + \lambda^{N-2} R_{t+1}^{(N-1)} \right] \\
&= (1-\lambda) R_t^{(1)} + \lambda r_t + \gamma\lambda \left[ (1-\lambda) \sum_{n'=1}^{N-2} \lambda^{n'-1} R_{t+1}^{(n')} + \lambda^{N-2} R_{t+1}^{(N-1)} \right] \\
&= (1-\lambda) R_t^{(1)} + \lambda r_t + \gamma\lambda R_{t+1}^\lambda \\
&= R_t^{(1)} - \lambda R_t^{(1)} + \lambda r_t + \gamma\lambda R_{t+1}^\lambda \\
&= r_t + \gamma \max_{a'} Q(s', a') - \lambda \left( r_t + \gamma \max_{a'} Q(s', a') \right) + \lambda r_t + \gamma\lambda R_{t+1}^\lambda \\
&= r_t + \gamma \max_{a'} Q(s', a') + \gamma\lambda R_{t+1}^\lambda - \gamma\lambda \max_{a'} Q(s', a') \\
&= r_t + \gamma \left[ \lambda R_{t+1}^\lambda + (1-\lambda) \max_{a'} Q(s', a') \right],
\end{aligned}
$$

where we used the recursive relationship for $R_t^{(n)}$ in Equation (25) and the substitution $R_t^{(1)} = r_t + \gamma \max_{a'} Q(s', a')$. Finally, we note that in our implementation, we replace the true value function with a function approximator.

## C EXPERIMENTAL SETUP

All experimental results are shown as mean of 10 seeds, except in Atari Learning Environment (ALE) where we followed a common practice of reporting 3 seeds. They were performed on a single NVIDIA A40 by jit-compiling the entire pipeline with Jax in the GPU, except for the Atari experiments where the environments run on an AMD 7513 32-Core Processor. Hyperparameters for all experiments can be found in Appendix E. We used the algorithm proposed in Algorithm 1. All experiments used Rectified Adam optimiser Liu et al. (2019). We didn't find any improvements in scores by using RAdam instead of Adam, but we found it more robust in respect to the epsilon parameter, simplifying the tuning of the optimiser.

**Baird's Counterexample** For these experiments, we use the code provided as solutions to the problems of (Sutton & Barto, 2018b) [2]. We use a single-layer neural network with a hidden size of 16 neurons, with normalisation between the hidden layer and the output layer. To not include additional parameters and completely adhere to theory, we don't learn affine tranformation parameters in these experiments, which rescale the normalised output by a factor $\gamma$ and add a bias $\beta$. However, in more complex experiments we do learn these parameters.

---

[2] https://github.com/vojtamolda/reinforcement-learning-an-introduction/tree/main

**DeepSea**    For these experiments, we utilised a simplified version of Bootstrapped-DQN (Osband et al., 2016), featuring an ensemble of 20 randomly initialised policies, each represented by a two-layered MLP with middle-layer normalisation. We did not employ target networks and updated all policies in parallel by sampling from a shared replay buffer. We adhered to the same parameters for Bootstrapped-DQN as presented in Osband et al. (2019).

**MinAtar**    We used the vectorised version of MinAtar (Young & Tian, 2019) present in Gymnax and tested PQN against PPO in the 4 available tasks: Asterix, SpaceInvaders, Freeway and Breakout. PQN and PPO use both a Convolutional Network with 16 filters with a 3-sized kernel (same as reported in the original MinAtar paper) followed by a 128-sized feed-forward layer. Results in MinAtar are reported in Fig. 9. Hyperparameters were tuned for both PQN and PPO.

**Atari**    We use the vectorised version of ALE provided by Envpool for a preliminary evaluation of our method. Given that our main baseline is the CleanRL (Huang et al., 2022b) implementation of PPO (which also uses Envpool and Jax), we used its environment and neural network configuration. This configuration is also used in the results reported in the original Rainbow paper, allowing us to obtain additional baseline scores from there. Aitchison et al. (Aitchison et al., 2023) recently found that the scores obtained by algorithms in 5 of the Atari games have a high correlation with the scores obtained in the entire suite, and that 10 games can predict the final score with an error lower than 10%. This is due to the high level of correlation between many of the Atari games. The results we present for PQN are obtained by rolling out a greedy-policy in 8 separate parallel environments during training, which is more effective than stopping training to evaluate on entire episodes, since in Atari they can last hundreds of thousands of frames. We did not compare with distributed methods like Ape-X and R2D2 because they use an enormous time-budget (5 days of training per game) and frames (almost 40 Bilions), which are outside our computational budget. We comment that these methods typically ignore concerns of sample efficiency. For example Ape-X (Horgan et al., 2018) takes more than 100M frames to solve Pong, the easiest game of the ALE, which can be solved in few million steps by traditional methods and PQN.

**Craftax**    We follow the same implementation details indicated in the original Craftax paper Matthews et al. (2024a). Our RNN implementation is the same as the MLP one, with an addtional LSTM layer before the last layer.

**Hanabi**    We used the Jax implementation of environments present in JaxMARL. Our model doesn't use RNNs in this task. From all the elements present in the R2D2-VDN presented in Hu et al. (2021), we only used the duelling architecture Wang et al. (2016). Presented results of PQN are average across 100k test games.

**Smax**    We used the same RNN architecture of QMix present in JaxMARL, with the only difference that we don't use a replay buffer, with added normalisation and $Q(\lambda)$. We evaluated with all the standard SMAX maps excluding the ones relative to more than 20 agents, because they could not be run with traditional QMix due to memory limitations.

**Overcooked**    We used the same CNN architecture of VDN present in JaxMARL, with the only difference that we don't use a replay buffer, with added normalisation and $Q(\lambda)$.

# D    FURTHER RESULTS

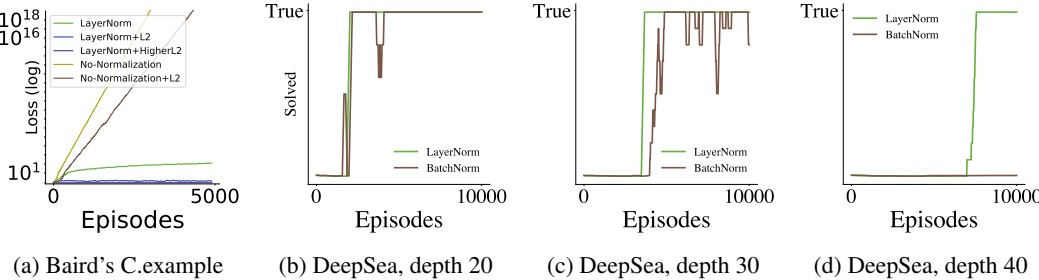

(a) Baird's C.example    (b) DeepSea, depth 20    (c) DeepSea, depth 30    (d) DeepSea, depth 40

Figure 7: Results from theoretical analysis in Baird's Counterexample and DeepSea.

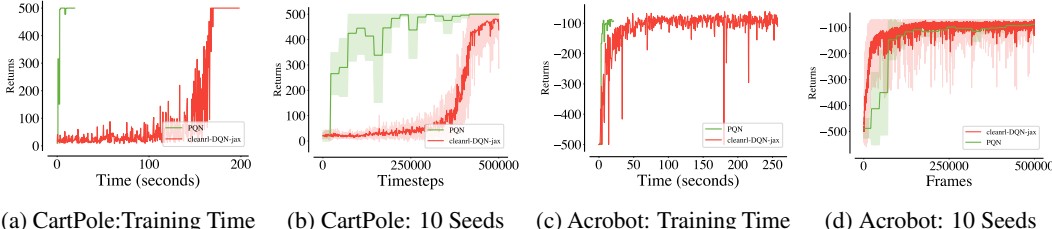

(a) CartPole:Training Time    (b) CartPole: 10 Seeds    (c) Acrobot: Training Time    (d) Acrobot: 10 Seeds

Figure 8: Results in classic control tasks. The goal of this comparison is to show the time boost of PQN relative to a traditional DQN agent running a single environment in the cpu. PQN is compiled to run entirely on gpu, achieving a 10x speed-up compared to the standard DQN pipeline.

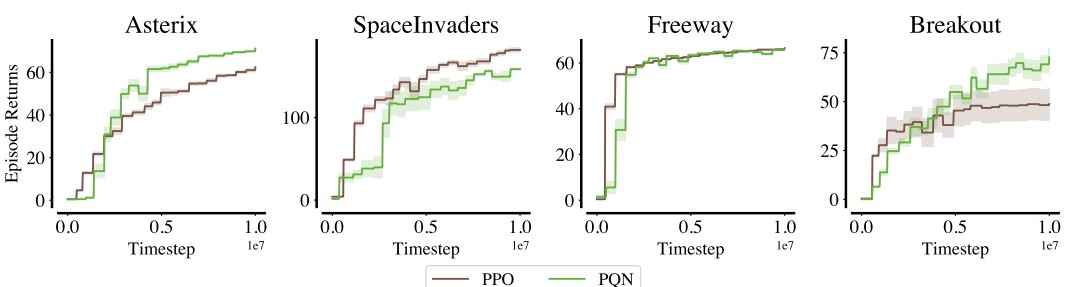

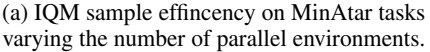

Figure 9: Results in MinAtar

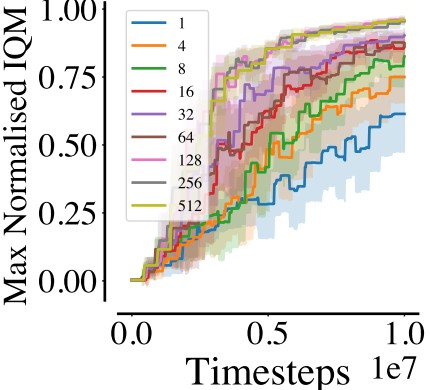

(a) IQM sample effincency on MinAtar tasks varying the number of parallel environments.

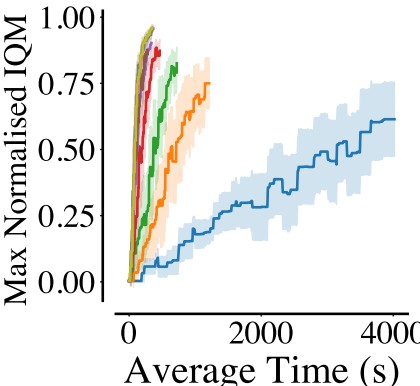

(b) IQM time effincency on MinAtar tasks varying the number of parallel environments.

Figure 10: Ablation study varying the number of parallel environments in Minatar. PQN can learn even with a small number of environments but clearly benefits from collecting more experiences in parallel. PQN is also significantly more time-efficient when more environments are used in parallel (time is considered for running 10 seeds in parallel). For a fair comparison, we adjusted the number of minibatches and epochs so that PQN performs the same number of gradient steps with the same batch size (or, where not possible, with an adjusted learning rate) for every number of parallel environments considered.

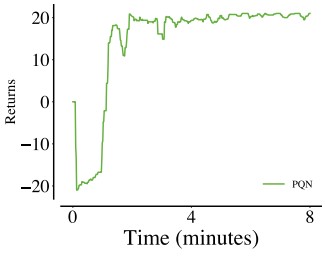
(a) Atari-Pong: PQN training

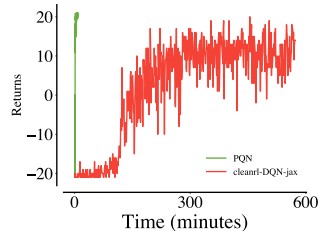
(b) Atari-Pong: Time Comparison

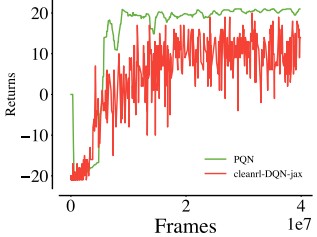
(c) Atari-Pong: Performances

Figure 11: Comparison between training a Q-Learning agent in Atari-Pong with PQN and the CleanRL implementation of DQN. PQN can solve the game by reaching a score of 20 in less than 4 minutes, while DQN requires almost 6 hours. As shown in the plot on the right, this doesn't result in a loss of sample efficiency, as traditional distributed systems like Ape-X need more than 100 million frames to solve this simple game.

Table 3: Scores in ALE.

| Method (Frames) | Time (hours) | Gradient Steps | Atari-10 Score | Atari-57 Median | Atari-57 Mean | Atari-57 >Human |
|---|---|---|---|---|---|---|
| PPO (200M) | 2.5 | **780k** | 165 | | | |
| PQN (200M) | **1** | **780k** | 191 | | | |
| PQN (400M) | 2 | 1.4M | 243 | **245** | 1440 | 40 |
| Rainbow (200M) | 100 | 12.5M | **239** | 230 | **1461** | **43** |

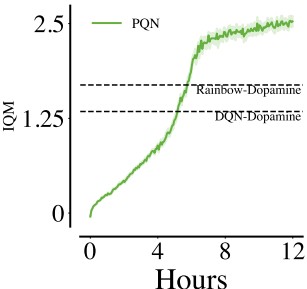

Figure 12: IQM computed over 3 seeds when training PQN with the ALE configuration proposed by Dopamine Castro et al. (2018). This configuration incorporates sticky actions and doesn't set the done flag when an agent loses a life. With this setup, PQN can still outperform Rainbow, but it requires significantly more compute time (almost 5 hours), corresponding to 800 million frames, indicating a loss of sample efficiency. Sample efficiency might be recovered in this configuration by using a larger network or tuning the hyperparameters, but we leave this as future work. PQN is still much faster to train than a Dopamine agent, which requires multiple days depending on the hardware.

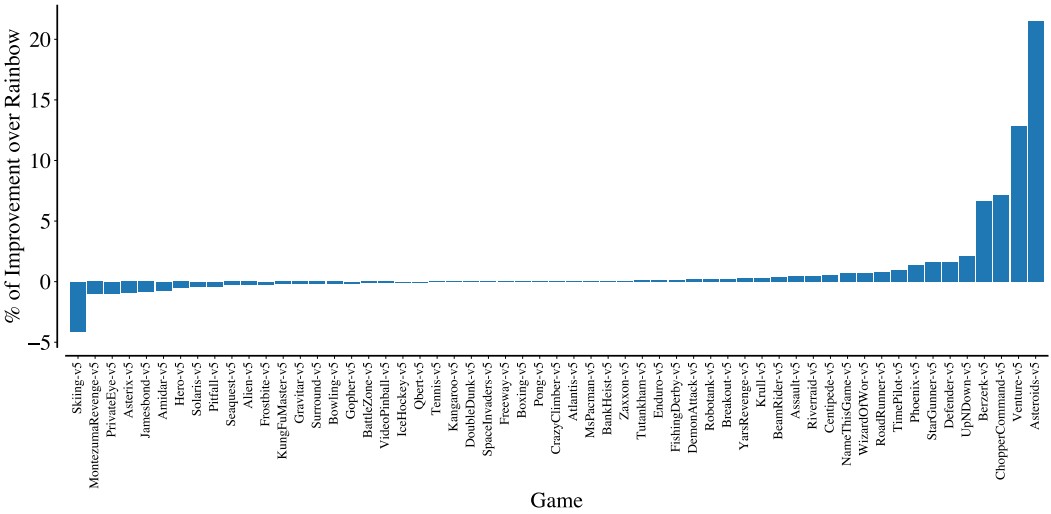

Figure 13: Improvement of PQN over Rainbow. Results refer to PQN trained for 400M frames, i.e. 2 hours of GPU time.

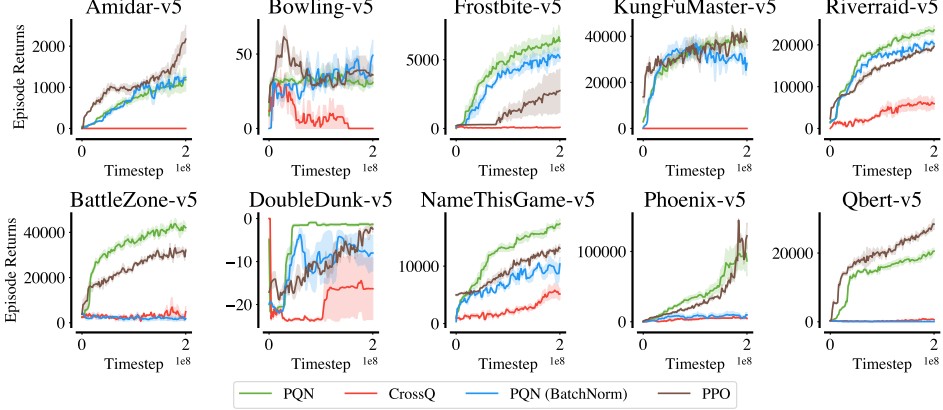

Figure 14: Atari 10 Results

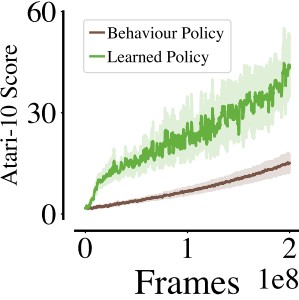

Figure 15: PQN learns a policy from an almost random policy. To further test PQN's off-policiness, we conducted an experiment on Atari10 games using a highly random policy for collecting data, gradually shifting from 100% random to 70% random during training. As expected, the resulting policy was less effective compared to one with more exploitation. However, the key finding is that PQN can still learn a policy even when off-policiness is extremely high — i.e., when data is collected almost randomly from the environment, without following the learning policy.

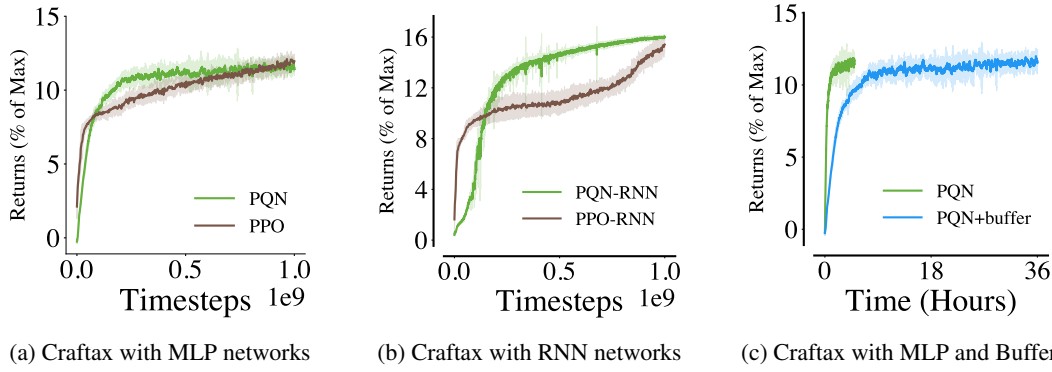

Figure 16: Left: comparison between PPO and PQN in Craftax. Center: comparison with RNN versions of the two algorithms. Right: time to learn for 1e9 timesteps keeping a replay buffer in GPU.

Figure 17: Results in Smax

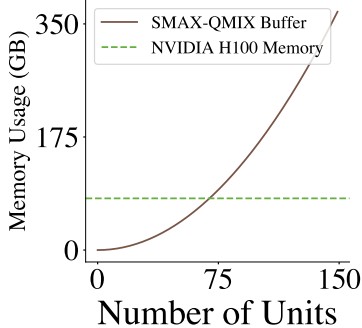

Figure 18: The buffer size scales quadratically in respect to the number of agents in SMAX.

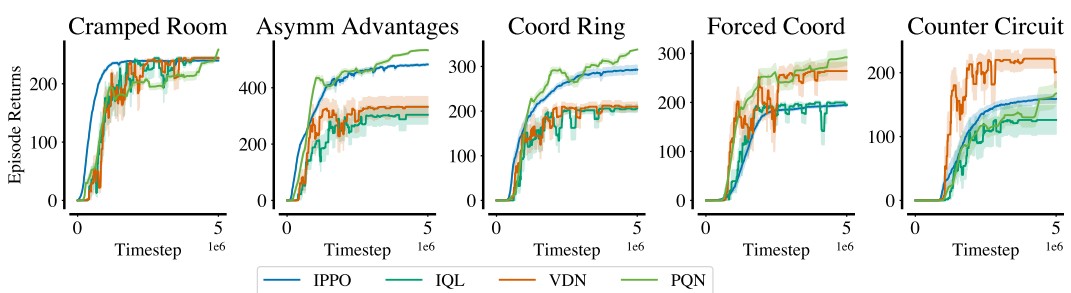

Figure 19: Results in Overcooked

# E  HYPERPARAMETERS

Table 4: Craftax RNN Hyperparameters

| Parameter | Value |
|---|---|
| NUM_ENVs | 1024 |
| NUM_STEPS | 128 |
| EPS_START | 1.0 |
| EPS_FINISH | 0.005 |
| EPS_DECAY | 0.1 |
| NUM_MINIBATCHES | 4 |
| NUM_EPOCHS | 4 |
| NORM_INPUT | True |
| NORM_TYPE | "batch_norm" |
| HIDDEN_SIZE | 512 |
| NUM_LAYERS | 1 |
| NUM_RNN_LAYERS | 1 |
| ADD_LAST_ACTION | True |
| LR | 0.0003 |
| MAX_GRAD_NORM | 0.5 |
| LR_LINEAR_DECAY | True |
| REW_SCALE | 1.0 |
| GAMMA | 0.99 |
| LAMBDA | 0.5 |

Table 5: Atari Hyperparameters

| Parameter | Value |
|-----------|-------|
| NUM_ENVs | 128 |
| NUM_STEPS | 32 |
| EPS_START | 1.0 |
| EPS_FINISH | 0.001 |
| EPS_DECAY | 0.1 |
| NUM_EPOCHS | 2 |
| NUM_MINIBATCHES | 32 |
| NORM_INPUT | False |
| NORM_TYPE | "layer_norm" |
| LR | 0.00025 |
| MAX_GRAD_NORM | 10 |
| LR_LINEAR_DECAY | False |
| GAMMA | 0.99 |
| LAMBDA | 0.65 |

Table 6: SMAX Hyperparameters

| Parameter | Value |
|-----------|-------|
| NUM_ENVs | 128 |
| MEMORY_WINDOW | 4 |
| NUM_STEPS | 128 |
| HIDDEN_SIZE | 512 |
| NUM_LAYERS | 2 |
| NORM_INPUT | True |
| NORM_TYPE | "batch_norm" |
| EPS_START | 1.0 |
| EPS_FINISH | 0.01 |
| EPS_DECAY | 0.1 |
| MAX_GRAD_NORM | 1 |
| NUM_MINIBATCHES | 16 |
| NUM_EPOCHS | 4 |
| LR | 0.00025 |
| LR_LINEAR_DECAY | True |
| GAMMA | 0.99 |
| LAMBDA | 0.85 |

Table 7: Overcooked Hyperparameters

| Parameter | Value |
|---|---|
| NUM_ENVs | 64 |
| NUM_STEPS | 16 |
| HIDDEN_SIZE | 512 |
| NUM_LAYERS | 2 |
| NORM_INPUT | False |
| NORM_TYPE | "layer_norm" |
| EPS_START | 1.0 |
| EPS_FINISH | 0.2 |
| EPS_DECAY | 0.2 |
| MAX_GRAD_NORM | 10 |
| NUM_MINIBATCHES | 16 |
| NUM_EPOCHS | 4 |
| LR | 0.000075 |
| LR_LINEAR_DECAY | True |
| GAMMA | 0.99 |
| LAMBDA | 0.5 |

Table 8: Hanabi Hyperparameters

| Parameter | Value |
|---|---|
| NUM_ENVS | 1024 |
| NUM_STEPS | 1 |
| TOTAL_TIMESTEPS | 1e10 |
| HIDDEN_SIZE | 512 |
| N_LAYERS | 3 |
| NORM_TYPE | layer_norm |
| DUELING | True |
| EPS_START | 0.01 |
| EPS_FINISH | 0.001 |
| EPS_DECAY | 0.1 |
| MAX_GRAD_NORM | 0.5 |
| NUM_MINIBATCHES | 1 |
| NUM_EPOCHS | 1 |
| LR | 0.0003 |
| LR_LINEAR_DECAY | False |
| GAMMA | 0.99 |

| Game | Rainbow | PQN |
|---|---|---|
| Alien | **9491.70** | 6970.42 |
| Amidar | **5131.20** | 1408.15 |
| Assault | 14198.50 | **20089.42** |
| Asterix | **428200.30** | 38708.98 |
| Asteroids | 2712.80 | **45573.75** |
| Atlantis | 826659.50 | **845520.83** |
| BankHeist | 1358.00 | **1431.25** |
| BattleZone | **62010.00** | 54791.67 |
| BeamRider | 16850.20 | **23338.83** |
| Berzerk | 2545.60 | **18542.20** |
| Bowling | **30.00** | 28.71 |
| Boxing | 99.60 | **99.63** |
| Breakout | 417.50 | **515.08** |
| Centipede | 8167.30 | **11347.98** |
| ChopperCommand | 16654.00 | **129962.50** |
| CrazyClimber | 168788.50 | **171579.17** |
| Defender | 55105.00 | **140741.67** |
| DemonAttack | 111185.20 | **133075.21** |
| DoubleDunk | **-0.30** | -0.92 |
| Enduro | 2125.90 | **2349.17** |
| FishingDerby | 31.30 | **46.17** |
| Freeway | **34.00** | 33.75 |
| Frostbite | **9590.50** | 7313.54 |
| Gopher | **70354.60** | 60259.17 |
| Gravitar | **1419.30** | 1158.33 |
| Hero | **55887.40** | 26099.17 |
| IceHockey | **1.10** | 0.17 |
| Jamesbond | **20000.00** | 3254.17 |
| Kangaroo | **14637.50** | 14116.67 |
| Krull | 8741.50 | **10853.33** |
| KungFuMaster | **52181.00** | 41033.33 |
| MontezumaRevenge | **384.00** | 0.00 |
| MsPacman | 5380.40 | **5567.50** |
| NameThisGame | 13136.00 | **20603.33** |
| Phoenix | 108528.60 | **252173.33** |
| Pitfall | **0.00** | -89.21 |
| Pong | **20.90** | 20.92 |
| PrivateEye | **4234.00** | 100.00 |
| Qbert | **33817.50** | 31716.67 |
| Riverraid | 20000.00 | **28764.27** |
| RoadRunner | 62041.00 | **109742.71** |
| Robotank | 61.40 | **73.96** |
| Seaquest | **15898.90** | 11345.00 |
| Skiing | **-12957.80** | -29975.31 |
| Solaris | **3560.30** | 2607.50 |
| SpaceInvaders | **18789.00** | 18450.83 |
| StarGunner | 127029.00 | **331300.00** |
| Surround | **9.70** | 5.88 |
| Tennis | **0.00** | -1.04 |
| TimePilot | 12926.00 | **21950.00** |
| Tutankham | 241.00 | **264.71** |
| UpNDown | 100000.00 | **308327.92** |
| Venture | 5.50 | **76.04** |
| VideoPinball | **533936.50** | 489716.33 |
| WizardOfWor | 17862.50 | **30192.71** |
| YarsRevenge | **102557.00** | 129463.79 |
| Zaxxon | 22209.50 | **23537.50** |

Table 9: ALE Scores: Rainbow vs PQN (400M frames)

