# OpenReview forum: "Simplifying Deep Temporal Difference Learning"
_ICLR.cc/2025/Conference — ICLR 2025 Spotlight_

### Official Review · Reviewer_KANn · 2024-10-31

**Soundness:** 3
**Presentation:** 3
**Contribution:** 4
**Rating:** 8
**Confidence:** 3

**Summary:**

This paper introduces Parallelised Q-Network (PQN), a streamlined deep online Q-Learning algorithm. PQN comprises two key components: a TD Learning objective without a target network, which instead applies layer normalization and L2 regularization, and a parallelized sampling approach that avoids the use of a replay buffer by leveraging vectorized environments. The authors provide theoretical analysis to support the claim that regularization can stabilize TD Learning in the absence of target networks. PQN demonstrates competitive performance across a wide range of environments, achieving results in significantly less wall-clock time.

**Strengths:**

1. PQN is straightforward and easy to implement. It removes the need for a target network and simplifies TD Learning.
2. The paper includes theoretical analysis showing that regularization can help keep TD Learning stable without using target networks.
3. PQN achieves higher computational efficiency compared with baseline methods, with minimal impact in sample efficiency.

**Weaknesses:**

1. It’s somewhat counterintuitive that PQN maintains sample efficiency while training only on online samples without a replay buffer. Additional explanation would help readers understand this aspect better.
2. The removal of the target network in TD Learning and the parallelized sampling are independent components of the algorithm, yet their individual contributions to overall performance are unclear. More controlled experiments, like the one in Figure 6.d, would clarify the impact of each component.
3. The parallelized sampling approach depends on vectorized environments, which feels more like an engineering choice than a novel contribution and is not feasible in many real-world applications. A significant portion of the wall-clock savings seems to come from this aspect. If my understanding is correct, DQN could also potentially eliminate the replay buffer and use a similar parallelized sampling approach. It would be informative to see how DQN performs under this setup.

**Questions:**

See above

---

> ### Author Response · Authors · 2024-11-21
> **Rebuttal to Reviewer KANn**
>
> We thank the reviewer for pointing out areas where we can improve our paper.
>
> 1. One of the strengths of PQN is indeed its sample efficiency combined with fast parallelised training. The performance of PQN is the result of several factors:
>     - Removing the target networks reduces the lag in learning.
>     - Adding normalisation improves stability and accelerates convergence (see Figure 6.a).
>     - All the benefits of parallelisation discussed in Section 4.1.
>     - To further increase sample efficiency, the use of minibatches and miniepochs allows training with experiences sampled online multiple times (see our PQN($\lambda$) algorithm).
>
>     We realise that we did not sufficiently discuss the benefits of removing the target networks or the additional improvements in sample efficiency achievable through minibatches and miniepochs. We will add a discussion on these points in a future draft, with the aim of providing greater clarity.
>
> 2. We did conduct an ablation study to test the impact of layer normalisation on the method (Figure 6.a). We are currently working on an additional ablation to assess the effect of using multiple parallel environments with PQN, although this ablation may be challenging to perform rigorously (see the final point of our comment, *“Unfair or unclear empirical evaluation,”* to reviewer LgPr).
>
> 3. Many RL applications are trained in simulators, where there is no reason not to utilise parallelisation (see more on this in our comment, *“PQN solves a problem different from the baseline DQN, but this was never discussed,”* to reviewer LgPr). Moreover, our contribution is not solely based on parallelisation but also includes a deep theoretical analysis of the role of network normalisation in RL. Removing the replay buffer to perform parallelised sampling would make DQN very similar to PQN. However, PQN additionally employs layer normalisation to stabilise training instead of target networks, thereby improving stability and reducing the computational complexity of the algorithm.

---

> ### Author Response · Authors · 2024-12-02
>
> As all of the reviewer's original concerns have been addressed in the updated draft (including new ablations) following our response below and our detailed response to Reviewer LgPr, the authors would really appreciate it if the reviewer could either acknowledge this and raise their score accordingly for the paper to be accepted if satisfied or let the authors know if they have further concerns.

---

> > ### Comment · Reviewer_KANn · 2024-12-03
> >
> > Thank you for providing additional experiments and making revisions. I have reviewed the revised version, and I believe it is much stronger than before, thanks in large part to the thoughtful suggestions by Reviewer LgPr. In particular, I find the following two modifications greatly enhance the quality of the paper:
> >
> > 1. The addition of the claim that PQN addresses a parallel-world problem distinct from the original problem DQN aimed to solve. This improves the clarity of the paper and avoids confusion.
> > 2. The ablation studies on the number of environments are highly valuable and informative. These studies demonstrate that PQN can learn effectively in a single environment, highlighting the impact of removing the target network. Additionally, they show that PQN benefits substantially from increased parallelism.
> >
> > In addition, I agree that the theoretical part is a valuable contribution to the area.
> >
> > As all my concerns have been addressed, I have updated my score to reflect these improvements

---

> ### Author Response · Authors · 2024-12-03
>
> Thanks so much, the authors really appreciate the feedback and the chance to improve the paper from it.

---

### Official Review · Reviewer_Jf9o · 2024-11-01

**Soundness:** 3
**Presentation:** 4
**Contribution:** 4
**Rating:** 8
**Confidence:** 2

**Summary:**

This paper analyses stability in temporal difference (TD) methods. The main contributions are theoretical proofs that (i) TD instability can be established using a Jacobian evaluated on the unit circle; and (ii) using the Layernorm regularisation technique can ensure stability. This then leads the authors to propose a deep Q-learning algorithm called PQN which is comptetive with the PPO approach to reinforcement learning.

**Strengths:**

The paper is well written, well formatted and quite readable. The authors present the essence of their results very well and show that stability of TD algorithms reduces to checking that a Jacobian is negative definite on the unit circle. This is a nice succinct and somehow intuitive result. Given the complexity of the proof, it was good to see the summary presented so concisely.

Other results are then presented after this, including some insight into the causes of instability and then the approach using the layernorm to obtain stabilisation. The presentation here was not as clear as the above, but still acceptable and still concise.

The authors claim that their parallelised version of Q learning is motivated well and this seems to be backed up by experiments.

The overall implications of the authors' results are very significant: they have discovered and captured the root cause for TD instability, they have proposed an improvement which guarantees instability and they have showed that their new PQN performs exceptionally well on some examples.

**Weaknesses:**

A criticism is that the proof of the main theoretical results is long (there are 20 pages of additional material) and I would say not particularly well organised. Before the authors give the proofs, in my view, it would be good for them to outline the main steps. I found the proofs hard to follow and as one goes through the proofs, there is a feeling of being somewhat adrift. In other words, the summary in the main paper is good; the actual proofs in the appendix are less well clear.

**Questions:**

One question I have is why does the Jacobian have to be negative definite on the unit Circle? Why is simple negative definiteness not enough? Since any vector can be written as || u || = c || v || where c is a positive constant, it seems that simply negative definiteness of the Jacobian is required? It would be helpful to the reader if the authors could give more justification and/or insight into the reason negative definiteness on unit circle is required, or the authors may wish to reconsider their results and see whether the restriction can be removed.

---

> ### Author Response · Authors · 2024-11-21
> **Rebuttal to Reviewer Jf9o**
>
> We thank the reviewer for carefully reading our work and proofs. Their feedback is really constructive and will help improve the paper further.
>
> We have included more details on the theory in the main body of our paper. We will also provide more detail in the Appendix proofs in an updated draft once Reviewer LgPr responds to our rebuttal, we agree it can be a lot to take in on first read.
>
> ## Answer to question:
>
> We choose a unit test vector $v$ as the property $\lVert v \rVert^2=1$ makes our proofs cleaner and aligns nicely with the definition of the Matrix 2 norm, especially as we take limits of infinite network widths. As the reviewer has identified, negative definiteness on a unit circle implies negative definiteness in the whole of $\mathbb{R}^n\setminus \{0\}$: $v^\top J v<0 \implies c^2 v^\top J v<0\implies(cv)^\top J (cv)<0$ for any $c>0$, so this condition is no less general.

---

> ### Comment · Reviewer_Jf9o · 2024-11-27
>
> Thanks for the response. I of course agree that $v^T J v < 0$ for all $\| v\|=1$ is equivalent to $v^T J v$ for all $v \in \mathbb{R}^n / 0$: this was my point in my review. I can also see that restricting $\| v \| =1$ is useful in the *proof*. However, in the *statement* of the result, why not simply state that $J$ must be negative definite?

---

> > ### Author Response · Authors · 2024-11-27
> >
> > We see. We didn't use negative definite because it only applies to symmetric matrices, so would be incorrect and misleading. Unlike a Hessian, the Jacobian has no such guarantee of this. The equivalent definition for non-symmetric matrices is negative quadratic form. We have updated the draft to make this clear.

---

> > > ### Comment · Reviewer_Jf9o · 2024-12-02
> > >
> > > There are two schools of thought on what makes a negative definite matrix. One school says a negative definite matrix is symmetric; one does not insist on this. Since all real matrices can be decomposed as $J = J_s + J_{ss}$ where $J_s$ is symmetric and $J_{ss}$ is skew-symmetric, from the quadratic form argument, it is clear that $J_{ss}$ has no effect on definiteness. So, if you were from the first school of thought, you could simply say $J_s=\frac{1}{2}(J+J^T)$ should be negative definite, or simply that $J+J^T<0$. I am pointing this out just because, in my view, it is easier to read $J+J^T<0$ than the quadratic form condition, and it is of course easy to check.

---

> > > > ### Author Response · Authors · 2024-12-02
> > > >
> > > > We appreciate the reviewer's response and we use the format suggested in our proof directly. Our core concern was we find papers using the term positive definite without proper qualification a bit sloppy. As long as there is precision in it's meaning we don't have particularly strong opinions on how the condition is defined and are happy to change to the reviewer's suggested format using the transpose.

---

### Official Review · Reviewer_gFms · 2024-11-04

**Soundness:** 4
**Presentation:** 3
**Contribution:** 3
**Rating:** 8
**Confidence:** 3

**Summary:**

This paper proposes simplifications to multiple components of the Deep Q-Network (DQN)/TD learning method to enable more efficient training on a single GPU, offering a potential DQN baseline for future research. The modifications include:
- Eliminating target network tricks: The authors theoretically demonstrate that combining layer normalization with L2 regularization leads to convergent temporal difference (TD) learning. They then conducted experiments to validate this empirically by removing the target network update tricks.
- Removing replay buffer for experience replay: The paper identifies the replay buffer as a memory bottleneck that limits single-GPU training. While directly removing the replay buffer impacts sample efficiency, the paper demonstrates that when combined with vectorized environments, the GPU-based training method achieves better wall-clock time efficiency.
- Batch-wise rollout using vectorized environments: The paper implements DQN training in a batch-wise manner, leveraging GPU parallelization (after removing the replay buffer). Instead of single actor rollout, the paper uses vectorized environments to generate rollouts in batch.

To validate their approach, the authors conduct comprehensive experiments on both theoretical/proof-of-concept environments (Baird's counterexample) and standard benchmarks (Atari and Crafter). Their results show that the proposed Parallelized Q-Network (PQN) achieves comparable performance to well-known baselines like PPO and Rainbow. Through ablation studies, they further demonstrate the importance of network normalization and justify the removal of the replay buffer.

**Strengths:**

- This paper is well-written and easy to follow, with clear presentation of both theoretical derivations and experimental results.
- The paper's motivation is clear and compelling enough to me: it mainly provides a simplified Q-learning baseline that effectively leverages GPU parallelization and vectorized environments.
- The proposed experimental evaluation is relatively comprehensive: it covers multiple domains including proof-of-concept environments, standard single-agent benchmarks such as Atari and Crafter, and multi-agent scenarios. They also covers variants of the Q learning methods to support the claim better in general.

**Weaknesses:**

There is no major weakness of this paper, but feel free to check the question section for minor questions.

**Questions:**

- Including Baird's counterexample results in the main text would strengthen the paper in my opinion, by providing a clearer connection between the theoretical analysis and experimental validation.
- (Minor) The PPO baseline comparison in Figure 3 could be more consistent, though I understand the thinking to save compute. The paper could either compare both methods using 4e8 training frames in Figure 3(a), or include PPO results directly in Figure 3(b) across all Atari environments. Either approach would more effectively demonstrate PQN's competitiveness against established policy gradient methods, in terms of sample efficiency.

---

> ### Author Response · Authors · 2024-11-21
> **Rebuttal to Reviewer gFms**
>
> We thank the reviewer for their thoughtful and response and the time taken to review our work carefully. If space permits, depending upon Reviewer LgPr's response to our rebuttal, we will include Baird's counterexample (which we have extended to stronger $\ell_2$-regularisation) in the main body of the paper.

---

> > ### Author Response · Authors · 2024-11-25
> >
> > FYI, we have added Baird's counterexample to the new draft of the paper. Once again, thanks for the help improving our work.

---

### Official Review · Reviewer_LgPr · 2024-11-04

**Soundness:** 3
**Presentation:** 3
**Contribution:** 3
**Rating:** 6
**Confidence:** 3

**Summary:**

Modern deep-reinforcement learning resorts to techniques such as replay buffers and target networks to provide stability with nonlinear off-policy learning. However, learning becomes unstable without a replay buffer or target networks and can diverge. Recently, several works suggested using layer normalization or layer normalization in addition to l2 regularization to remedy this learning instability issue. This paper theoretically studies layer normalization’s role and identifies how layer normalization helps with stability and convergence. The paper also proposed a new method called PQN that uses layer normalization and parallelized environments to stabilize learning. The authors show the effectiveness of their method through a series of experiments on different domains of environments.

**Strengths:**

The work on simplifying deep reinforcement learning and removing techniques that might not be necessary, like replay buffer and target networks, is undoubtedly fundamental to deep reinforcement research. This has vast implications for rethinking the widely existing deep RL approaches and can help in other important directions, like scaling RL with the number of parameters/samples. This paper provides a unique view that challenges existing beliefs on the importance of replay buffers and target networks. The approach is effective and efficient since it can be implemented parallelized on GPU, which outperforms other baselines with respect to wall clock time.

**Weaknesses:**

- The theoretical part of the manuscript is largely incoherent.
  - The current manuscript scatters many things in the theory parts. It lacks a proper flow of ideas when describing the theoretical results and their implications, which makes it difficult to follow. Currently, it reads as bullet points, listing findings quickly without proper linking between subsequent findings or results.
  - For example, the current theorems and lemmas are not well integrated with the text before and after them. They read as detached components, making reading unnecessarily harder.
  - Notations can be improved. I suggest following Sutton & Barto (2018).
  - Two things that are instrumental for the results based on Jacobian analysis: inequality 3 and inequality 4. More discussion is needed to understand these two conditions and their implications.
  - Off-policy instability and nonlinear instability require their own theorem statements and separate proofs (even if previous works have shown them). In addition, the TD stability criterion needs a theorem statement about contraction mapping.
  - The connection for why l2 is needed was not clear. It was directly introduced after layer norm without proper linking.
- PQN solves a problem different from the baseline DQN, but this was never discussed
  - The authors emphasize that PQN does not use a replay buffer or target network as an advantage over other methods, which is great. However, a similar emphasis is needed for the fact that PQN requires parallel environments and probably may fail if a single environment were used (the setting of the other baselines). Additionally, PQN solves another orthogonal problem (parallelized worlds) to the original RL problem (single world).
  - Figure 1 is inaccurate. The replay buffer is part of the agent, not an external component. This needs to be fixed.
  - The difference between Distributed DQN and PQN is unclear from Figure 1, although both solve the same problem (parallelized worlds), especially the point on synchronism and GPU is not clear.
- The paper has several inaccuracies.
  - The authors claim that their PQN is based on Peng’s Q($\lambda$), but the actual algorithm does not use eligibility traces. Instead, the authors use Q-learning with $\lambda$-return. This needs to be corrected. Additionally, the equation (line 326) needs to be derived from the first principle to make the paper accessible to the unfamiliar reader.
  - Algorithm 1 is Q-learning with one-step targets. The authors mention that they use $\lambda$-return target, so Algorithm 1 needs to be replaced with the algorithm actually used (Algorithm 2 in Appendix C).
  - The authors claim they stabilize learning Baird’s counterexample and use Figure 7a to demonstrate that. However, in Figure 7a, the error increases from the starting point until it plateaus. I don’t think meaningful learning has happened since the error has increased instead of decreased. I see that with layer norm or layer norm + L2 regularization, the error doesn’t increase without bound but at the same time, the problem is not solved either.
- Unfair or unclear empirical evaluation
  - The authors compare algorithms that work with parallel environments against ones that do not, which requires careful experiments to compare them.
  - In Figure 3, the authors need to write rainbow (200M) or DQN (200M) to understand what those horizontal lines represent clearly.
  - When the authors say that PQN was trained for 400M frames, does that include the parallel environments? For example, if 128 parallel environments are used (according to Table 5), does this mean 3.125M frames were collected from each environment, resulting in a total of 400M frames, or does it mean that 400M frames were collected from each environment, resulting in 128x400M=51200M overall frames? The first option gives a fair comparison, but the second option is biased towards PQN since significantly more experience is used. I would like the authors to clarify this point.
  - The authors used only 3 independent runs for Atari, relying on precedence.  This is a too low number to provide any statistical significance. Even if something was accepted before, it does not mean it is correct. I highly suggest the authors increase the number of independent runs to at least 10. This should be possible since both PQN and PPO are efficient (small clock time) and easy to run, according to the paper's claims.
  - The authors mentioned that DQN/Rainbow uses 50M updates compared to 700k updates for PQN. I think it is unclear how these numbers are obtained, especially for PQN.
  - In Figure 6a, PQN still learns well without divergence when no layer normalization is used. What is the reason? Why has no divergence happened?
  - Since the authors compare against DQN and Rainbow (methods that use a single environment), an ablation where a different number of environments are considered (e.g., n=1 and n=10) to understand the provided stability coming from parallelized environments. I think parallel environments make the gradient signal more reliable compared to the single environment case, which is more prone to noisy gradients. This may be instrumental for PQN; thus, an ablation is needed.

&nbsp;

**Minor issues:**

- line 9 in Algorithm 1 and line 10 in Algorithm 2 are incorrect. The negative sign should be a plus. Additionally, I think the use of $\texttt{StopGrad}$ can be eliminated with the TD error vector you gave in Eq. 2

&nbsp;
&nbsp;

Overall, I believe the ideas from this paper could serve as a good contribution to the community, but the current manuscript is not ready for publication and needs significant improvement. I'm willing to increase my score, given that the authors improved their manuscript's quality by 1) rewriting the theoretical part to make it coherent and more rigorous, 2) Fixing the inaccuracies, and 3) improving the empirical evaluation quality.

**Questions:**

- Is PPO using the same number of parallel environments as PQN in all experiments? I couldn’t find this information in the paper. Could you share this information and add it to the paper’s revision?
- In line 428, the authors refer to a histogram in Appendix E, but there is no histogram. Do they mean the bar plot in Figure 12?
- Typically, layer norm is not applied to the post-activation but instead to the pre-activation. This is an important distinction. The theory still works with the preactivation layer norm as long as you don’t use activation functions that scale up the inputs.

---

> ### Author Response · Authors · 2024-11-21
> **Rebuttal to Reviewer LgPr**
>
> We thank the reviewer for their constructive review and appreciate that they see the strong contribution of our work. We feel we have addressed most of their concerns in the updated draft and appreciate the opportunity to improve and make our paper intelligible to as wide an audience as possible. With remaining issues, we would like to raise a few points of clarification in separate responses. As the reviewer has raised several points, we will address them here in few separate comments. Regarding the questions that the reviewer asked:
>
> 1. Yes PPO is using the same parallel environments of PQN. We've added this to the updated draft.
> 2. Yes we refer to Figure 12. Thank you for pointing out the typo, we've fixed it.
> 3. We are slightly struggling to parse the final part of this question. Do you mean what happens if an activation is placed after the final LayerNorm? If this is so, let's say the activation has Lipschitz constant $L$, then all theory remains the same except the residual term in Eq. 9 becomes $\left\lVert v_w \cdot\frac{L \gamma  }{2}\right\rVert^2$. We would then need to scale the $\ell_2$ regularisation term by $L$ to compensate for this. We put the activation before the LayerNorm as this is typical for RL-specific methods like CrossQ, but can include a discussion of this in the Appendix if necessary.

---

> > ### Comment · Reviewer_LgPr · 2024-11-24
> > **Thank you for answering my questions**
> >
> > For the last answer, yes, I would appreciate it if this point could be discussed in the main paper or the appendix. I disagree with the precedence argument, as the paper has to be self-contained without relying too much on other papers. The theory holds regardless of the place of the activation with respect to the layer normalization, which strengthens the theoretical part of the paper.

---

> ### Author Response · Authors · 2024-11-21
> **The theoretical part of the manuscript is largely incoherent.**
>
> 1. The reviewer claims notation would be improved following Sutton and Barto. Most of our notation follows this, however when it differs it is to ensure precision and intelligibility. Firstly, Sutton and Barto do not write expectations with respect to the underlying distribution, they just use $E$ rather than $E_{x\sim P_X}$ like our notation. Whilst this may suffice for a less technical paper, the authors find it sloppy and extremely frustrating when this notation is used as the distribution is essential to determining stability of TD. Secondly, Sutton and Barto use the notation $Q(S_t,A_t)$ for $Q$-functions. We use $Q(x)$ where $x=(s,a)$ is clearly defined in the preliminaries and at several points in the paper. Switching to $Q(S_t,A_t)$ and similar notation would cause most equations to overflow onto several lines and would hinder the intelligibility of the work. Similar work proving TD convergence [1] (also authored by Sutton) uses even sparser notation, i.e. just $V^\pi$ and $V_\theta$. With this in mind, could the reviewer highlight any additional specific notation that they would wish us to change?
>
>
> 2. The reviewer asks for the off-policy and nonlinear parts to have their own proofs. These do have their own separate proofs that exist clearly within separate sections in the Appendix (Lemma 3 (Mitigating Off-policy Instability) and Lemma 4 (Mitigating Nonlinear Instability)). These results are summarised on separate lines in Lemma 1 of the main body. In the new draft, we have made this even clearer, labelling them off-policy bound and nonlinear bound. Aside from this, we are not sure what the reviewer is asking for here as separating the two lines into two lemmas in the main body would not add anything except take up unnecesary space in the main body. Please could the reviewer clarify what they mean?
>
> We have added more discussion of Inqs. 3 and 4, provided more intuition of adding $\ell_2$ regularisation and a geometric interpretation of our Jacobian analysis in the updated draft, and would appreciate if the reviewer could indicate they are satisfied with this. We thank the reviewer, we feel the paper's theoretical contribution is now very simple to understand.
>
>
> [1] [Sutton et al., Fast Gradient-Descent Methods for Temporal-Difference Learning
> with Linear Function Approximation, ICML 2009](https://icml.cc/Conferences/2009/papers/546.pdf)

---

> ### Author Response · Authors · 2024-11-21
> **PQN solves a problem different from the baseline DQN, but this was never discussed**
>
> - We agree that PQN is designed to interact with an environment by taking multiple actions and observing multiple states and rewards in parallel, which may not be feasible in some scenarios. However, as the reviewer points out, it does so by building on a much simpler algorithm; removing target networks and replay through regularised networks and reverting back to the original Q-learning algorithm, with the benefit of theoretical guarentees. We are thus unsure what the review means by *"probably may fail if a single environment were used."*  There is a wealth of empirical evidence to suggest that regularising Q-Learning [2][3][4] or actor-critic methods [5] (especially without a target network [2][5]) in single-interaction environments improves sample efficiency, computational efficiency and performance. As we state in our paper, whilst providing strong empirical evidence, this prior work offers no theoretical analysis. This is why our theoretical contribution is necessary.  Repeating these experiments offers nothing in the way of contribution as we'd be repeating what has been done many times before. We will highlight this empirical evidence even further in an updated draft.
> -  Moreover, we wish to clarify that many RL applications are trained in simulators rather than in the real world. In these situations, there is no reason not to exploit parallelisation. Our method is the first approach that unlocks the potential of these methods through its stability, allowing for a truely online off-policy algorithm with improved sample efficieny and performance. With this in mind, DQN is a baseline for PQN because both try to apply Q-Learning to the RL problem in a sample efficient way. DQN uses a large replay buffer. PQN uses parallelised interactions. We compared them using sample efficiency as an evaluation metric. We are happy to discuss these points further in an updated daft.
> - We understand the confusion in Fig. 1. We changed "Agent" for "Q-Network" and kept them separately as they can be considered two different parts of the agent (PQN keeps only the Q-Network).
> - The difference between Distrubuted DQN and PQN is that the former uses multiple copies of a neural network (for instance via multi-thread or multi-machine actors) to collect experiences while continuously traning the network in a separate, parallel process (i.e. having a learner module and multiple actors modules running concurrently). PQN instead is a sequential process which involves collecting vectorised experiences, using them to train a single network. In other words, a single process running in batches. We updated the figure caption accordingly.
>
> [2] [Nauman et al, Bigger, Regularized, Optimistic: scaling for compute and sample-efficient continuous control. 2024](https://arxiv.org/pdf/2405.16158)
>
> [3] [Understanding Plasticity in Neural Networks. ICML 2023](https://arxiv.org/pdf/2303.01486)
>
> [4] [Disentangling the causes of plasticity loss in neural network. 2024](https://arxiv.org/pdf/2402.18762)
>
> [5] [CrossQ: Batch Normalization in Deep Reinforcement Learning for Greater Sample Efficiency and Simplicity. ICLR 2024](https://arxiv.org/pdf/1902.05605)

---

> > ### Comment · Reviewer_LgPr · 2024-11-24
> > **My response to authors' rebuttal (1/2)**
> >
> > I thank the authors for their detailed rebuttal. Here is my response to the authors' rebuttal. I respond here to all the points except for the theoretical part, and the changes made in the new revision since these would need a look at the paper after the authors mark the changes with a unique color.
> > \
> > \
> > **PQN solves a problem different from the baseline DQN, but this was never discussed**
> >
> > To reiterate, I’m not trying to undermine the value of the algorithm. I think there is a large number of researchers that would benefit from this for training in simulation. My point is that it’s not emphasized enough that PQN does not solve the original RL problem (single world) but instead solves another orthogonal problem (parallel worlds). It is presented in such a way that PQN is a better alternative to DQN, which is not true. To make such a claim, you need to examine what happens to PQN when the number of environments is 1. If the authors are willing to make the necessary effort to make the point about single world vs parallel worlds problems clear in the introduction, my concern will be addressed.
> >
> > **The paper has several inaccuracies**
> >
> > - The authors have a misunderstanding about the difference between eligibility traces (backward view) and $\lambda$-return (forward view). They are not equivalent to each other except in a limited sense (in linear function approximation or tabular settings, they lead to the same updates). Under nonlinearity, they are not equivalent. Furthermore, even under linear function approximation or tabular settings, they achieve the same updates but in completely different ways, resulting in online algorithms with eligibility and offline algorithms with $\lambda$-returns. In [6], the authors discuss the tabular setting where the equivalence between the forward and backward view can be shown, which is not applicable for PQN since neural networks are considered. Finally, PQN uses $\lambda$-returns, so we cannot write PQN($\lambda$) because it makes it look like it uses eligibility traces like TD($\lambda$). I suggest renaming it as “PQN with $\lambda$-returns".
> > - I thank the authors for providing the derivation for the recursive $\lambda$-return formula. The last missing point is to emphasize that you replace the true action values with their estimated action values.
> > - After the authors moved Algorithm 2 to the main paper, I see that the $\lambda$-returns now are computed in a different way, can the authors explain why they made that change? Which way of computation does PQN use?
> > - I appreciate that the authors moved the actual algorithm used to the main paper.
> > - The point I’m trying to make here about Baird’s counterexample is that it’s not clear from the figure that the error is reduced to zero. It seems like the algorithm only solves the divergence issue but still is not able to reduce the error compared to other algorithms [7] (Figure 11.6). I expect the authors to make this point clear in the paper. My concern would be addressed if they either acknowledge in the paper that the error doesn’t get reduced completely or zoom in on the figure to show that it reduces to zero.
> >
> > **Unfair or unclear empirical evaluation**
> >
> > - I thank the authors for assuring me that the overall number of frames is fixed for both PQN and the other baselines.
> > - I appreciate that the authors are planning to provide more independent runs to make the results more statistically significant.
> > - I think by looking at the algorithm again, I *no longer agree* with the claim that no buffer is used. To compute such $\lambda$-returns, PQN has to maintain a buffer to store $n\times m$ transitions where $n$ is the number of environments and $m$ is the number of steps to run each environment before making the update. I recognize that this buffer is smaller than the one used in DQN, but it’s still a buffer nonetheless ($128\times 32= 4096$ in Atari and $1024\times128=131072$ in Craftax). This point has to be clearly mentioned in the main paper to avoid confusion about the main claims of the paper.
> > - I strongly believe the ablation is necessary for the current paper. The number of environments is a hyperparameter of PQN, and the reader would like to know what happens when we set it to a very small value.

---

> > > ### Author Response · Authors · 2024-11-25
> > > **Response to Reviewer LgPr**
> > >
> > > - Sure, we will rename the algorithm PQN with $\lambda$-returns and will emphasize that we replace the true action values with their estimated action values.
> > > - The returns are calculated exactly as in [6]. Does the reviewer agree that this is the case in the current draft? We have used more concise notation (as requested) to show the update is an expectation over agent interactions and a minibatch (hence the summations), but don't beleive we have changed anything else and apologise for any typos if so.
> > > - Re Baird's: Value error tending to zero and convergence to fixed points are two seperate issues. This is well known [8]. We make no claim about reducing value error to zero. Prior work characterising stability of TD [9][10][11][12][13] does not touch upon it and we don't understand why it is relevant for our paper, however we will add: 'It is well known that convergence to fixed point does not imply a value error of zero [8]'
> > > - We will add a discussion about memory requirements when introducing $\lambda$-returns.
> > > - The ablations are nearly ready and will be found in the next draft.
> > > - If the reviewer is now satisfied, we will work to get the updated draft ready ASAP.
> > >
> > >
> > > [8] [Kolter, The Fixed Points of Off-Policy TD, 2011](https://zicokolter.com/publications/kolter2011fixed.pdf)
> > >
> > > [9] [Bhandari et al., A Finite Time Analysis of Temporal Difference Learning With Linear Function Approximation](https://arxiv.org/pdf/1806.02450)
> > >
> > > [10] [Narayanan and Szepesvári, Finite Time Bounds for Temporal Difference Learning with Function Approximation: Problems with some “state-of-the-art” result 2018](https://sites.ualberta.ca/~szepesva/papers/TD-issues17.pdf)
> > >
> > > [11] [Analysis of Temporal-Diffference Learning with Function Approximation](https://proceedings.neurips.cc/paper_files/paper/1996/file/e00406144c1e7e35240afed70f34166a-Paper.pdf)
> > >
> > > [12] [Dalal et al., Finite Sample Analyses for TD(0) with Function Approximation, 2107](https://arxiv.org/pdf/1704.01161)
> > >
> > > [13] [Fellows et al., Why Target Networks Stabilise Temporal Difference Methods, 2023](https://arxiv.org/pdf/2302.12537)

---

> > > > ### Comment · Reviewer_LgPr · 2024-11-25
> > > > **Response to the authors**
> > > >
> > > > I thank the authors for the response and for their effort to address the concerns. Here is my reply:
> > > >
> > > > - Re Baird's: adding that sentence addressed my concern about this experiment.
> > > >
> > > > - Adding the discussion about the memory requirement would address my concern if the authors agree also to drop the claims about removing the need for replay buffers from their manuscript. Instead, the claim can be reducing the memory requirements when compared to DQN.
> > > >
> > > > - Are the authors willing to provide a discussion emphasizing the point about solving a different problem orthogonal to the original RL problem? If so, then that would address my concern.
> > > >
> > > > - Please let me know when the ablation is ready.

---

> > > > > ### Author Response · Authors · 2024-11-25
> > > > >
> > > > > The ablation is ready, we are just finishing the updated draft.
> > > > >
> > > > > With regards to removing the need for replay buffers, there's a profound theoretical point that we can strip the algorithm of target networks and storage of historic data (including replay buffers) and it is stable. This is such a powerful part of our theoretical results.  So on a purely theoretical level, we would like to make this point clear.
> > > > >
> > > > > On a practical level, in all versions of PQN, this means we don't use data that comes from historic exploration policies like a true replay buffer would. The authors believe that collecting data under historic policies is a defining feature of a replay buffer. DQN can't work without this. Again, we feel this is a really important point to make, which is why we introduced Fig. 3.
> > > > >
> > > > > Do you agree with these points and would you accept a draft where we say: 1 our theory allows us to remove replay buffers; and 2: when introducing our algorithm, make it clear that our approach requires memory to store transitions from the current policy, but not a replay buffer which stores transitions from historic policies.

---

> > > > > > ### Comment · Reviewer_LgPr · 2024-11-25
> > > > > > **Response to the authors**
> > > > > >
> > > > > > - I agree that on a purely theoretical level, you can make the claim that no replay buffer or target network is necessary for the proof to work. But on an empirical level, you used a buffer with PQN, so you need to be transparent about that. I understand that you don't use the buffer the same way used in DQN. Your approach is more like PPO's buffer. Nonetheless, it is still a buffer and the paper should make this very clear. I don't see any reason for calling it something else. This would only confuse the reader.
> > > > > >
> > > > > > - You missed one of my questions so I'm posting it again: Are the authors willing to provide a discussion emphasizing the point about solving a different problem orthogonal to the original RL problem?

---

> > > > > > > ### Author Response · Authors · 2024-11-25
> > > > > > >
> > > > > > > - We will provide a clear discussion when introducing the algorithm that it requires a buffer
> > > > > > >
> > > > > > > - Sure, we will do this.

---

> > > > > > > ### Author Response · Authors · 2024-11-25
> > > > > > > **Draft updated, a summary of changes**
> > > > > > >
> > > > > > > We have uploaded a new draft of the paper. In this version:
> > > > > > >
> > > > > > > - We have added a preliminary ablation study varying the number of environments across all the tasks in MinAtar. The ablation demonstrates that PQN can learn a policy even with a single environment, while also highlighting the benefits of parallel environments in terms of sample and time efficiency. Performance improvement is significant even using few parallel environments. We are willing to extend this ablation to Atari-10 if the reviewer believes it is necessary; however, this would require significant time and resources, as experiments with fewer environments are highly time-intensive (we estimate more than one month of compute time).
> > > > > > > - We have emphasised that PQN is designed to exploit situations where multiple actions can be taken in an environment at once, that is the parallel world problem, firstly in the introduction (line 54) and then when introducing the algorithm (line 363).
> > > > > > > - To avoid confusion, we have followed the reviewer’s suggestion and referred to PQN with $\lambda$-returns as the algorithm that uses $\lambda$ targets throughout the paper (Section 4 onwards)
> > > > > > > - We have clarified that PQN’s sample efficiency can be improved by using minibatches and miniepochs. (Line: 355)
> > > > > > > - We have tried to make Algorithm 1 more clear.
> > > > > > > - We have extended our analysis to deal with a final activation after the LayerNorm operator. (Line 249, Lemma 2 and Eq.9, following through in the Appendix)
> > > > > > > - We have clarified that when using $\lambda$-returns, a small buffer may be necessary depending on the specific implementation (Line 330) and refer to this at several points in the paper. When referring to PQN with $\lambda$-returns, we claim it replaces the need for a large replay buffer.
> > > > > > > - We have added that we replace the true value function with the approximate value function when deriving $\lambda$-returns (Line 1729)
> > > > > > >
> > > > > > > We thank the reviewers for identifying unclear points in our paper and for helping us improve our work in a collaborative manner. It is certaintly a better paper and we appreciate the time taken. If they are satisfied, we hope they can raise their score so our work can reach a large audience.

---

> > > > > > > > ### Comment · Reviewer_LgPr · 2024-11-25
> > > > > > > > **Thank you for revising the paper**
> > > > > > > >
> > > > > > > > Thank you for the new draft. To make the process efficient so that I can respond quickly, can you please mark the changes with a different text color?

---

> > > > > > > > > ### Author Response · Authors · 2024-11-26
> > > > > > > > >
> > > > > > > > > We're really sorry but we have been working flat out to get this ready for the original deadline and are so exhausted that this is not possible until tomorrow due to the time it will take to track all the changes and colour them. We have provided line references instead in the meantime.

---

> > > > > > > > > > ### Comment · Reviewer_LgPr · 2024-11-26
> > > > > > > > > >
> > > > > > > > > > Of course! Take your time. Also, thanks for letting me know that the deadline got extended.

---

> > > > > > > > > > > ### Author Response · Authors · 2024-11-26
> > > > > > > > > > > **Coloured Version Uploaded**
> > > > > > > > > > >
> > > > > > > > > > > The updated version with major changes made in cyan has been uploaded.

---

> > > ### Public Comment · ~Antonin_Raffin1 · 2024-11-25
> > > **Ablation Study Needed**
> > >
> > > Dear authors,
> > >
> > > > I strongly believe the ablation is necessary for the current paper.
> > >
> > > I agree with reviewer LgPr that this ablation study is very important, it would allow practitioners to know if they can use your algorithm or not.
> > > In the sense that PQN works because of parallel environments, but not all environments can be massively parallelized.
> > >
> > > The question is, how many parallel environments do you need to be able to use PQN?
> > > And what hyper-parameters should be adjusted when using less envs?
> > > (correct me if I'm wrong, but the current version of the paper does not answer this question).

---

> > > > ### Author Response · Authors · 2024-11-25
> > > >
> > > > Please see the latest draft and our response to the reviewer (Draft updated, a summary of changes) about these new ablations

---

> ### Author Response · Authors · 2024-11-21
> **The paper has several inaccuracies.**
>
> -  Our implementation of $\lambda$-returns follows [6], where it is proved this formulation correspond to Peng's $Q(\lambda)$: "Note that the $\lambda$-return presented here unconditionally conducts backups using the maximizing action for each n-step return, regardless of which actions were actually selected by the behavioral policy μ. This is equivalent to Peng’s Q(λ)". We have added the derivation of equation 326 Q($\lambda$) in the Appendix.
> -  We have added Algorithm 2 in the main text. Subjectively, the authors really don't like this as we find it is messy and feel it ruins the exposition of the paper. We provided the simplest form of PQN (Algorithm 1) in the main body of the text as it is a powerful and clean expositional tool that helps readers understand our approach. Algorithm 2 is simply a generalisation of this based on $Q(\lambda)$ and we feel its details don't contribute anything to the reader's understanding. Many papers (including [6]) opt to detail algorithmic extensions they may use in the Appendix as providing these details add nothing to the paper's exposition. However, we will keep Algorithm 2 in the main body if the reviewer still disagrees with us about this.
> -  Our experiments follow evaluations of other convergent TD algorithms such as GTD, GTD2 and TDC (see specifically Section 11.7 and Figure 11.5 of Sutton and Barto [7]). Baird's counterexample is a provably divergent domain used to show that off-policy TD approaches diverge, that is their parameters grow without bound. Demonstrating that this is not the case, that is parameter values converge to some fixed value, is the purpose of the experiment which confirms the theoretical results. Plateauing of parameter values and thus value error is expected: like in our experiment, Figure 11.5 of [6] shows TDC learn non-zero weights and a value error that plateaus. Our results are thus no different to existing established methods.
> -  Finally, it is important to note from our theory that adding $\ell_2$ regularisation of magnitude $\left(\frac{\gamma}{2}\right)^2$ is just enough to ensure the TD stabilty criterion holds. We can think of this as being the edge of guaranteed stability. Without $\ell_2$ regularisation, parameters diverge in Baird's counterexample whereas with regularisation of magnitude $\left(\frac{\gamma}{2}\right)^2$, parameters converge, albeit to a plateaued value. We find it remarkable that the theory aligns so well with the empirical results. Strenghtening $\ell_2$ regularisation beyond this thus further regularises the problem, allowing us to reduce the value error to a greater extent. We have since tested and confirmed this empirically, results can be found in the updated draft.
>
>
> [6] [Daley, Amato, Reconciling λ-Returns with Experience Replay. 2019.](https://arxiv.org/abs/1810.09967)
>
> [7] [Sutton and Barto, Reinforcement Learning: An Introduction. 2018](http://incompleteideas.net/book/RLbook2020.pdf#page=279.21)

---

> ### Author Response · Authors · 2024-11-21
> **Unfair or unclear empirical evaluation**
>
> - We compare PQN mostly with PPO, which uses the same parallel environments of PQN. The comparison with non-parallelised environments is in order to compare sample efficency of PQN. Non-parallelised environments are notably much more sample efficient, and many times distributed algorithms are compared with non-parallelised only considering time as reference. On the contrary, we compare them using the number of frames, i.e. the sample efficency, where non-paralleised environments have an advantage.
> - Our calculation includes parallel environments. 400M of frames is the overall number of frames collected in parallel.
> - We are collecting more seeds for Atari right now and will update the figures when ready.
> - DQN/Rainbow update the network every 4 frames: 200M frames/4 = 50M updates. PQN updates the network two times every 4 frames but collects 128 frames in parallel: ((200M frames/4)/128)*2 = ~70k updates
> - We are open to conducting such an ablation; however, we are concerned that if done naively, it could be misleading. The issue lies in the interdependence between the number of parallel environments and other learning parameters. If we fix a budget for environment frames, collecting experiences faster generally leads to fewer network updates with larger batch sizes. This, in turn, necessitates adjusting hyperparameters such as the learning rate and the number/size of minibatches. A fair ablation would therefore require fine-tuning these hyperparameters for each number of parallel environments considered. We plan to perform such a fair ablation in at least some simpler environments, but we believe that a comprehensive analysis of this depth would be more suitable for a separate experimental study.

---

> > ### Comment · Reviewer_LgPr · 2024-11-23
> > **Thank you for your response**
> >
> > Thank you for your response and for revising the paper. I'll be more available from now until the deadline for a back-and-forth discussion. To make it more efficient, I would appreciate it if the authors could use a different color to distinguish the modification made from the original text and allow me to respond quickly.

---

> ### Author Response · Authors · 2024-11-27
>
> We would like to ask if the reviewer's concerns have been addressed as there is little time now to provide any more drafts before the deadline.

---

> ### Author Response · Authors · 2024-11-28
> **Colour draft was uploaded two days ago. Have all concerns been addressed?**
>
> The authors are keen to hear back from the reviewer as to whether the latest draft (which was uploaded in colour as requested) has  satisfied their concerns. There is precious little time now until the deadline and we would like to know if there are any other points that need addressing. If they are satisfied, we hope they can raise their score as promised so our work can reach a large audience.

---

> ### Comment · Reviewer_LgPr · 2024-11-29
>
> I thank the authors for their efforts to address my concerns. The provided ablation and the author's promise to provide more runs for the Atari experiments address my concern about the empirical evaluation. Thus, I raised my score accordingly. I'm willing to increase the score further as soon as my other concerns are addressed which can be done by answering my questions and by agreeing make the necessary modifications.
> \
> \
> I'm checking the draft at the moment. I'll leave my feedback to the parts I check as soon as I'm finish reading them. Here is my feedback on the algorithm:
> \
> \
> There are some remaining issues in Algorithm 1:
> - Using $r_t^i \sim P_R(s_t^i ,a_t^i),  s_{t+1}^i \sim P_S(s_t^i ,a_t^i)$ can be misleading for the unfamiliar reader. Those distributions are not accessible for the agent. I suggest using Sutton & Barto (2018) style of writing this line "Take $a_t^i$, observe $s_{t+1}^i$ and $r_{t+1}^i$, $\forall i \in \\{0,\dots,T-1\\}$.
> - The place of the $s_0\sim P_0$ is problematic. It needs to be right after "for each episode do". Also, it's missing the $i$ index for each environment.
> - The notation $\\{0 : I−1\\}$ is unclear. I suggest replacing that with $\\{0,...,I−1\\}$
> - I suggest the indexing starts with $1$ instead of $0$ for better readability.
> - typo: mini-epochs should be epochs
> - The notation $\\{t-T:t\\}$ is very ambiguous to refer to a buffer. I think the clearest way is to define a buffer $\mathcal{B}$ the is initialized to $\emptyset$ at the beginning of the algorithm. The buffer can store transitions $(s_{t}^i, a_{t}^i, r_{t+1}^i, s_{t+1}^i), \forall i \in \\{1,\dots, T \\}$ at each time step and emptied after the updating phase is complete.
> - What exactly is $\pi_{\text{explore}}$? I suppose you're using $\epsilon$-greedy policy. If so, then this needs to be a clear about that and say $\pi_{\text{$\epsilon$-greedy}}$.
> - The algorithm uses $x_t^i$ without a definition. I understand that it's defined in the paper, but I would be very useful for the algorithm to be self-contained such that people can understand it standalone.
>
> Related to the algorithm:
> - The recursive formulation for $\lambda$-return should have $r_{t+1}$ not $r_t$ (see line 328). The subscript in the derivation also needs to corrected as well. Additionally, I suggest using the letter $G$ instead of $R$ to avoid any confusion between return and reward. Also, lines 116 and 119, they have $r_{t}$ where it should be $r_{t+1}$.
> - In line 329, $R_T$ is not defined; is it a typo? Also, the subscript in the $R_T^{\lambda}$ equation indices are not consistent with the previous line if we replaced $t$ with $T$. Lastly, it's not clear how we can end up with this equation. Can the authors provide an explanation?

---

> ### Author Response · Authors · 2024-11-29
>
> - Choice of notation in Algorithm 1: We used the notation to align with [6] as well as being concise enough to fit into the page limit with all the extra information, ablations and explanations. As we have said, we can't update the draft anymore as the deadline has now passed. If it means the paper is accepted, we are more than happy to change to notation to what the reviewer has suggested.
>
> - Exploration policy: $\pi_\textrm{explore}$ is a general exploration policy that differs from the target policy. If could be $\epsilon$-greedy, decaying $\epsilon$-greedy or it could be a more sophisticated exploration policy (based on UCB or Bayesian uncertainty estimates). The point is, for the sake of generality, alignment with theory, future research and to highlight the off-policy nature of our approach, it is very important that this is kept as general possible. We say that for our experiments we used $\epsilon$-greedy and will highlight this further when it is introduced by writing: '$\pi_\textrm{Exploration}$ is an exploration policy. We use $\epsilon$-greedy exploration in this paper, but our theory is general enough to allow for the use of more sophisticated exploration policies in future work'.
>
> - Regards to subscripts, we are not sure that the reviewer is correct here. To see this, take $\lambda=0$, which should recover the 1-step TD update, $R_t=r_t+\gamma \max Q_\phi(s_{t+1},a')$  however under the reviewer's notation this would be  $R_t=r_{t+1}+\gamma \max Q_\phi(s_{t+1},a')$.
>
> - Yes. this is a typo, it should read $R^\lambda_t$. The term should read $R^\lambda_T=\max_{a'}Q_\phi(s_{T},a')$ which is the first starting point in the iterate process before progressing backwards. This is in line with Algorithm 1 of [6]. We will make this clearer in the final version writing:
>
> The exploration policy $\pi_\textrm{Explore}$  is rolled out for a small trajectory of size $T$: $(s_i,a_i,r_i,s_{i+1}\dots s_{i+T})$. Starting with $R_{i+T}^\lambda=\max_{a'} Q_\phi(s_{i+T}, a')$ the targets are computed recursively back in time from $ R_{i+T-1}^\lambda$ to $ R_i^\lambda$ using: $R_{t}^{\lambda} =r_t + \gamma \left[ \lambda R_{t+1}^{\lambda} + (1 - \lambda) \max_{a'} Q_\phi(s_{t+1}, a') \right]$ or $R_{t}^\lambda=r_t$ if $s_{t}$ is a terminal state.
>
>  Apologies, we were really exhausted trying to get the draft ready for the original deadline. We hope there is understanding here

---

> ### Author Response · Authors · 2024-12-02
> **Last day for reviewer-author discussion**
>
> As this is the last day of reviewer-author discussions, we were wondering if there was anything else the reviewer is concerned about before raising their score as promised.

---

> ### Comment · Reviewer_LgPr · 2024-12-02
>
> Thank you for the reminder. Sorry for not getting back to you sooner. I thank the authors for their reply. It is great to hear that the authors are willing to make the suggested changes if the paper gets accepted. Here is my response to some of the points:
>
> - Re: exploration policy, I understand that you can have policies other than $\epsilon$-greedy. My point here is to have at least "e.g.,  $\epsilon$-greedy" as part of the algorithm to guide the reader what this exploration policy might mean.
>
> - I suggested using $r_{t+1}$ instead of $r_t$, following Sutton and Barto (2018) since it's intuitive that the reward is given in the next step similar to the next state. It's not intuitive to have the reward and next state given in two consecutive time steps. I understand that according to your notations the definition is consistent, but I was hoping you can consider the trajectory to be $\tau_t \doteq (s_0, a_0, r_1, s_1, a_1, r_2, s_2, ..., s_{t-1}, a_{t-1}, r_t, s_t)$.
>
> - In line 365, the authors still say "without any replay buffer". This needs to be fixed. Also, the appendix needs to be checked for same claims (e.g., we don’t use a replay buffer in line 1815)
>
> - There is still mentioning to $Q(\lambda)$ in lines 536, 490, and 514. Also, the appendix needs to be checked for similar claims (e.g., with added normalisation and Q(λ) in line 1815)
>
> - In line 062 there is a claim about PQN performing online updates. I think the use of the word online is not meaningful for parallel environments or with $\lambda$-return computations.
>
> ---
>
> I checked the theory part once again. I think there is still room for improvement, so I encourage the authors to strive to improve accessibility for a wider range of audience. For example, a primer on Lyapunov stability needs to be added to section 2 since your analysis heavily depends on it.
>
> In my own experience, the most confusing part was the introduction of off-policy and nonlinear components without any derivations. Now after the authors have provided the derivation in Appendix B1, the logical flow became clear and easier to follow.
>
> There are some minor errors in the theory part:
> - In line 214, you have $\delta(\phi_t) (\phi_t-\phi^\star)<0$. Are you missing a transpose to have the dot product?  something like $\delta(\phi_t)^\top (\phi_t-\phi^\star)<0$?
> - In assumption 2 (line 146), the TD error vector $\delta$ takes $x$ and $\phi$ as arguments. Should this be ς and $\phi$ instead? Also in line 147, should it be "Lipschitz in $\phi$,ς".
> - In line 272, the authors have a limit with an index but the function doesn't have that index, so it needs to be fixed (e.g., making $Q_\phi^{\text{Layer}}$ be an explicit function of $k$.
>
> ---
>
> To summarize, here are my last concerns. They are easy to fix in the final paper if the authors agreed to. Please let me know if you are willing to make these modifications if the paper gets accepted.
>
> - Remove remaining claims about not using a replay buffer from the paper and using a more clear language instead to communicate this information. For example, "PQN uses a relatively small buffer" can replace "PQN eliminates the use of large replay buffers".
>
> - Provide a more comprehensive discussion that the learning paradigm of PQN is heavily inspired by PPO learning paradigm in the sense you use collect data in a buffer then go over the collected data in multiple epochs and mini-batch updates then empty the buffer.
>
> - Fixing Algorithm 1 as suggested.
>
> - Fixing the minor errors in the theory part.

---

> > ### Author Response · Authors · 2024-12-02
> >
> > ### Response to points:
> >
> > - Sure, we'll put "e.g. $\epsilon$-greedy" in the algorithm
> > - We're not sure we agree on reward notation. We find thinking about reward as a result of taking an action in the current state to be much more intuitive: as an example, if we reduce down to the special case of a bandit setting, having $r_{t+1}$ makes little sense when it is a result of an action at time $t$. For the reasons given earlier in the rebuttal, the authors really don't like the notation of Sutton and Barto as we find it confusing and sloppy and its mathematical imprecision has personally led to mistakes in derivations when replicating it. We recognise this is subjective and if it makes the difference in the reviewer's eyes between, say, a score of 6 or an 8, we will change it.
> > - Refs to replay buffer: will change
> > - Refs to Q$(\lambda)$: will change
> > - Ref to online, will change
> >
> > ### Theory:
> > - We can add a primer on Lyapunov stability
> > - We are glad the theory regarding nonlinear and off-policy stability contributions is clearer
> > - Minor points: yep, these are typos. Thanks for spotting, will change.
> > - We can make $Q^\textrm{Layer}_\phi$ explicitly depend on $k$.
> >
> > ### Other points:
> > Agree to all. Thanks for your continuing help and for reading the paper carefully.

---

> > > ### Comment · Reviewer_LgPr · 2024-12-02
> > >
> > > I think as long as your definition is clear and consistent, you can keep using the notations you like. You may also remind the reader in page 7 that it's $r_t$ not $r_{t+1}$ because of the trajectory definition in page 2.
> > >
> > > Thank you again for the hard work during the discussion period. I think this discussion was very fruitful and I think it will strengthen the paper. My concerns are now addressed. I increased my score accordingly to reflect the improvements.

---

> > > > ### Author Response · Authors · 2024-12-02
> > > >
> > > > Great, that's good to hear. We also appreciate the discussion too.
> > > >
> > > > If there's any change of a higher score, the authors really would appreciated it. We feel like even the theoretical contribution alone solves a longstanding open problem in RL - researchers can now formally use general TD methods with nonlinear function approximation or off-policy sampling without worrying about divergence - which we believe is very important if algorithms are to be used safely in practice. The lack of formal stability guarantees for popular methods is something the authors found quite surprising and unsatisfactory, which motivated our project. In addition, in the time since submitting, our algorithm has already been adopted by the community due to its improved sample and computational efficiency and ease of implementation. As such, we would love to get our paper read by a wide an audience as possible.
> > > >
> > > > If there is anything remaining that is preventing this, we would be keen to address it.

---

### Public Comment · ~Nico_Bohlinger1 · 2024-11-23
**Continuous action spaces and compatibility with SAC, DDPG, TD3**

PQN is a great step to lift the boundary between off-policy and on-policy RL algorithms and research. The authors show how the removal of replay buffers and target networks and the addition of training batches collected online by leveraging parallelized environments simplify and improve upon DQN. All the suggested changes can also directly be applied to off-policy algorithms for continuous action spaces, like SAC, DDPG, TD3, etc. I want to suggest that the authors could use at least one of these algorithms to test the general applicability of their changes and benchmark on the typical continuous action space environments, e.g. Gym MuJoCo. This is important to determine the usefulness of the proposed changes for the whole field and also to make a fairer comparison with the often mentioned PPO and CrossQ.
I would be happy to discuss the applicability of PQN's changes to algorithms for continuous action spaces.

---

> ### Author Response · Authors · 2024-11-26
>
> We thank you for your comment.
>
> Our algorithmic focus for this paper was to develop a modern $Q$-learning approach. It is well known that $Q$-learning based algorithms are not suitable for continuous action spaces because the use of $\max_{a} Q(s',a)$.
>
> We remark that we have provided a general and powerful theoretical analysis of TD (which applies to continuous domains), proved convergence of TD using LayerNorm + $\ell_2$ regularisation, thereby solving one of the most important open questions in RL - that is whether there exist powerful, simple nonlinear and/or off-policy TD algorithms that are provably convergent. This can stand alone as a significant theoretical contribution. In addition, we have developed a state-of-the-art $Q$-learning based algorithm that unlocks the potential of parallelised sampling. We have tested with baselines across 79 discrete-action tasks (2 Classic Control tasks, 4 MinAtar games, 57 Atari games, Craftax, 9 Smax tasks, 5 Overcooked, and Hanabi) and provided an extensive ablation study.
>
> In contrast, the original CrossQ paper, which was an excellent piece of work and was rightly awarded a spotlight position at ICLR 2024, provided no theoretical analysis and was evaluated in only six Mujoco continuous-action tasks. For this reason, it is clear that developing a continuous actor-critic algorithm lies well beyond the scope of a conference paper, although we do hope to explore such avenues in a journal paper.

---

### Public Comment · ~Mark_Towers1 · 2024-12-02

I'm very impressed by this paper and wish it would be published as I believe that the RL community can learn a lot from the extensive theoretical and empirical results presented.

However, I believe the authors consistently incorrectly cite the number of network updates for Rainbow as 50 million (Table 3 and Line 442, for example). The confusion arises from the difference in frames and steps. Rainbow uses a frame skip of four (for 50 million steps) and then does a gradient step every four steps (not frames), meaning they run 12.5 million updates, not the listed 50 million.
Additionally, the authors say PQN uses 700k network updates in Table 3 and L442, but I believe the value is actually ~780k, an error of >10%. Like Rainbow in Atari, PQN uses a frame skip of four with 128 environments and 32 rollout steps for roughly 12k batch updates (50 million / 128 environment / 32-step rollout). For these batches, PQN has two epochs of 32 mini-batches, making the number of network updates ~780k. Please correct me if you believe me wrong; otherwise, could the paper be updated with these corrected values?

An additional detail is that on reviewing the open-souced code for the implementation by the author (thank you this is extremely helpful for future work), I found the `pqn_atari` implementation config had `episodic_life=True`. Could the authors clarify if this parameter was used for the experiments in Section 5.2 and Figure 4? This is weakly referenced in Figure 14 that this was disabled so I presume it is enabled in Figure 4 but was not included in the Atari Hyperparmeter table (Table 5). If so, I strongly believe that the authors should note this when discussing their Atari results, as this has an impact on an agent's training for Atari to ensure fair comparisons in Figure 4 and for future work. The impact of this parameter is even noted by EnvPool in their [documentation](https://envpool.readthedocs.io/en/latest/env/atari.html) to "improve value estimation". The parameter changes Atari's behaviour to provide termination (done) signals when an agent loses a life, not just at the end of an episode, improving value estimation. As a result, if the authors are using this for Figure 4 but it is not used (to my knowledge) in the prior work (Rainbow, Prioritised DDQN and DDQN), this potentially provides an unfair comparison. Therefore, a discussion or note on this parameter seems necessary, in my opinion, beyond a caption in Appendix D of the paper.

A side note is that Figure 14 should cite [Machado et al., 2018](https://arxiv.org/abs/1709.06009) rather than Casto et al., 2018 to my understanding.

I would also request that the authors publish PQN performance for 200 million (alongside 400 million), as this is the standard benchmark in the field and would allow future readers to better compare data with other papers (Tables 3 and 9).

---

### Public Comment · ~Brett_Daley1 · 2025-04-21
**Paper contains copy and pasted material from Daley & Amato (2019)**

I am the first author of [Reconciling λ-returns with Experience Replay](https://arxiv.org/pdf/1810.09967), published at NeurIPS 2019, which came up during the reviewer discussion [6].

I was not involved in the review process.
It only recently came to my attention that the recursive λ-return derivation in Appendix B.4 reproduces a large portion of our own derivation (Appendix D of our paper) without acknowledgment, including
- 8 equations (same notation),
- 4 English sentences (identical wording).

[See the attached screenshots for a comparison.](https://drive.google.com/drive/folders/172odF77RDlxLABC6br66F0_YBcXc48ms?usp=sharing)

Copying and pasting without attribution is plagiarism.
This could be fixed by either
- Explicitly stating that a substantial portion of the derivation has been copied from Daley & Amato (2019), or
- Removing the derivation altogether and referring readers to Appendix D of Daley & Amato (2019).

Note that although it appears you slightly changed the λ-return formula used in your paper, we discuss both formulas in our work (see footnote 4 of our paper).
Both formulas have been well known since the 1990s.
However, the derivation in our paper is original and must be acknowledged if it is used.

Please resolve this soon for both the camera-ready and arXiv versions of the paper.

---

> ### Public Comment · ~Mattie_Fellows1 · 2025-04-21
>
> Thank you for raising this, we have immediately updated the Arxiv version with the statement `The original derivation can be found in Daley & Amato (2019, Appendix D), which we repeat and adapt here for convenience' at the start of Appendix B.4. We also request that the camera-ready revision be reopened as soon as possible be made so we can make the same change.
>
> Our intention was never to present the derivation as original; we clearly cite Daley & Amato (2019) in the main body when referring to the algorithm and derivation. In addition, the derivation was not included in the submission - as you can see from the discussion with the reviewer below, they requested we include it for completeness. We apologise again for this oversight, and will rectify this as soon as the opportunity on open review arises. EDIT: Paper has been updated, thanks for drawing attention to this.

---

### Meta-Review · Area_Chair_tMcP · 2024-12-20

**Metareview:**

This paper makes two contributions. First, a proof that TD learning converges when the network uses layer normalization and weight-decay. This is demonstrated in experiments that show that one need not use a target value network or a “replay buffer” (there was a lot of back and forth on what constitutes a replay buffer with the reviewers). Second, the authors use vectorized environments for batched-rollouts to speed up training. We would like to thank the reviewers and the authors for a healthy discussion that has led to improvements in the manuscript during the review process.

I recommend that this paper be accepted. I encourage the authors to tie up some of the loose ends (e.g., public comments on this forum, as well as comments by Reviewer LgPr) in the camera-ready manuscript.

**Additional Comments On Reviewer Discussion:**

Reviewer LgPr: there was a lot of discussion on the theory (improving the narrative of the the mathematics, some inaccurate claims made by the authors regarding the replay buffer, inconsistencies in reporting the number of gradient updates for other methods in the existing literature, why parallelism, etc.) The authors have done an excellent job of conversing with the reviewer, convincing them of certain points, made due modifications where necessary.

Reviewer gFms and Reviewer Jf9o: very positive review and not much to discuss.

Reviewer KANn wanted ablation studies which were added in the updated manuscript.

---

### Decision · Program_Chairs · 2025-01-22

Accept (Spotlight)